# Conservation Laws for Modern Neural Architectures

Viet-Hoang Tran [*1]  Vinh Khanh Bui [*2]  Tan Lai Ngoc [3]  Nam Nguyen [4]  Tuan Dam [4]  Tan M. Nguyen [1]

## Abstract

Understanding gradient descent dynamics is key to explaining the success of over-parameterized models, where implicit bias manifests through conservation laws in gradient flow. While such laws are well understood for linear and ReLU networks, they remain largely unexplored for modern architectures. This work develops a unified framework to characterize conservation laws for contemporary models, including feedforward networks with GELU, SiLU, and SwiGLU activations, multihead attention with sinusoidal and rotary positional encodings, and Mixture-of-Experts architectures under diverse gating designs. Our theoretical findings are supported by experiments that validate the predicted invariants.

## 1. Introduction

Conservation laws–quantities invariant along optimization trajectories–provide a principled lens on neural network training dynamics. By exposing geometric constraints inherent to both the architecture and the learning algorithm, these invariants shed light on the implicit bias of training, revealing properties that persist from initialization to convergence (Saxe et al., 2013; Bah et al., 2022; Tarmoun et al., 2021; Min et al., 2021). They also play a central role in theoretical studies of convergence, stability, and optimization (Du et al., 2018; Arora et al., 2018; Chizat & Bach, 2020; Ji & Telgarsky, 2018), and have inspired optimization methods that regulate these invariants to accelerate training (Saul, 2023; Stock et al., 2019).

**Characterizing Conservation Laws.** While many conserved quantities are known (Kunin et al., 2021; Abbe et al., 2023; Zhang et al., 2025), rigorous *completeness* results–showing that these exhaust all possible conservation laws–are still rare. Marcotte et al. (2023) provided a framework to

---
*Equal contribution  [1]National University of Singapore  [2]Center for AI Research, VinUniversity  [3]Independent Researcher  [4]Hanoi University of Science and Technology, Hanoi, Vietnam. Correspondence to: Viet-Hoang Tran <hoang.tranviet@u.nus.edu>.

*Proceedings of the 43rd International Conference on Machine Learning*, Seoul, South Korea. PMLR 306, 2026. Copyright 2026 by the author(s).

fully characterize smooth conservation laws in shallow linear or ReLU networks under additional assumptions on the loss and data. Marcotte et al. (2024) extended this approach to non-Euclidean gradient flows (e.g., ICNN, NMF) and momentum dynamics, revealing new invariants and establishing completeness. More recently, Marcotte et al. (2025) extended this framework to ResNets (He et al., 2016) and Transformers (Vaswani et al., 2017). However, their analysis of attention was limited to the single-head setting. Although several conserved quantities were identified for multi-head attention, the authors emphasized that a complete characterization in the multi-head case remains an open problem. This suggests that conservation laws for more sophisticated neural architectures are still far from being fully understood.

**Feedforward Networks.** Modern feedforward networks have moved well beyond classical activations such as sigmoid, tanh, and even ReLU. Although ReLU (Nair & Hinton, 2010) was long favored for alleviating vanishing gradients, its non-smoothness and dead-neuron behavior have driven the adoption of smoother alternatives. GELU (Hendrycks, 2016) became standard in Transformers such as BERT (Devlin et al., 2019), while SiLU/Swish (Ramachandran et al., 2017; Elfwing et al., 2018) often outperforms ReLU in large models. More recently, gated activations have emerged as the dominant choice: extending GLUs (Dauphin et al., 2017), SwiGLU blocks (Shazeer, 2020) significantly improve Transformer scaling and are now widely used in modern LLMs such as LLaMA (Touvron et al., 2023).

**Attention Mechanism and Positional Encoding.** The attention mechanism was originally proposed to allow selective focus on relevant input portions in sequence-to-sequence architectures (Luong et al., 2015), before Vaswani et al. (2017) positioned multi-head attention (MHA) as the cornerstone of Transformers, enabling models to capture diverse interaction patterns through parallel attention subspaces. Given that attention operations are inherently permutation-equivariant, positional encodings (PE) are necessary to inject sequential order information (Vaswani et al., 2017). The original Transformer employed sinusoidal positional encodings (Vaswani et al., 2017), computing position-dependent representations via trigonometric functions at different frequencies, which became widely adopted in seminal architectures such as Transformer-XL (Dai et al., 2019), and DETR (Zhu et al., 2020). A notable recent advance-

ment is *Rotary Positional Embedding* (RoPE) (Su et al., 2024), whose widespread adoption across leading contemporary models (Touvron et al., 2023; Chowdhery et al., 2023; Nijkamp et al., 2023; DeepSeek-AI, 2024; 2025; OpenAI, 2025; Bai et al., 2025; Yang et al., 2025) highlights its strong scalability and robustness in practice.

**Mixture-of-Experts.** Mixture-of-Experts (MoE) architectures were originally introduced to enable ensemble learning through specialized subnetworks (Jacobs et al., 1991; Jordan & Jacobs, 1994). Early Dense MoE models combine all experts via a gating network, but their computational cost becomes prohibitive at scale. This motivated Sparse MoE (SMoE) (Shazeer et al., 2017), which activates only a small subset of experts per token via Top-$k$ routing, greatly reducing inference cost while maintaining model capacity. The gating mechanism, traditionally implemented using softmax functions to produce normalized expert weights (Shazeer et al., 2017; Fedus et al., 2021), has been complemented by alternative designs including sigmoid-based gating (Lewis et al., 2021; Dai et al., 2022) that offers greater routing flexibility and load balancing properties. Recent work has further refined these approaches through normalized sigmoid gating (Liu et al., 2024). MoE architectures power numerous state-of-the-art LLMs (Fedus et al., 2021; Hui et al., 2024; Jiang et al., 2024; OpenAI, 2025; Liu et al., 2024; Yang et al., 2025), validating their effectiveness as a scalable paradigm for modern neural architectures.

**Contribution.** Motivated by the problem of characterizing conservation laws in training dynamics, this paper develops a unified theoretical framework and establishes complete characterizations of conservation laws for modern architectures widely used in deep learning. Our results are supported by rigorous proofs and cover key building blocks of contemporary models. The paper is organized as follows:

1. Section 2 reviews the conservation-law characterization framework developed in Marcotte et al. (2023; 2025). We also discuss the rationale behind their assumptions and explain how these conditions facilitate a complete characterization of conserved quantities.

2. Section 3 develops our perspective on the problem of characterizing conservation laws and outline a general solution strategy for obtaining a complete description of conserved quantities for a given model. This discussion leads to the conclusion that identifying conservation laws can be formulated as solving an associated system of partial differential equations. We also include toy examples to build intuition before guiding the reader toward the more technical developments in later sections.

3. Section 4 presents our main results on the complete characterization of conservation laws. We begin with feedforward networks equipped with GeLU, SiLU, and SwiGLU activations in Section 4.1. Next, in Section 4.2, we establish our characterization for multihead attention, resolving the open problem posed by Marcotte et al. (2025), and further extend the analysis to settings incorporating positional encodings. Finally, in Section 4.3, we study Mixture-of-Experts architectures under various gating designs, including dense and sparse regimes as well as sigmoid-based gating mechanisms.

4. Section 5 provides a high-level sketch of the main ideas underlying the proofs of our results, highlighting the key techniques, limitations, and broader implications.

5. Section 6 reports experiments validating our theoretical findings on conservation laws across all considered architectures, using both full-batch and SGD training on small and large scale multimodal datasets.

Theoretical proofs and additional experimental details are provided in the Appendix.

## 2. Previous Work on Conservation Laws

We adopt the problem setting of Marcotte et al. (2023; 2025). For a differentiable map $F\colon \mathbb{R}^m \to \mathbb{R}^n$, and for a vector-valued input component $A \in \mathbb{R}^{n'}$, we denote by $\nabla_A F \in \mathbb{R}^{n \times n'}$ the gradient of $F$ with respect to $A$. If $a \in \mathbb{R}$ is a scalar variable of $F$, we similarly write $\partial_a F = \nabla_a F \in \mathbb{R}^n$ for the corresponding partial derivative.

**Formal Definition of Conservation Laws.** Consider a model $f(\cdot; \theta)\colon \mathcal{X} \to \mathcal{Y}$ parameterized by $\theta \in \Theta = \mathbb{R}^D$, where $D$ denotes the total number of parameters. In this work, we primarily focus on regression settings in which $\mathcal{X} = \mathbb{R}^{d_{\text{in}}}$ and $\mathcal{Y} = \mathbb{R}^{d_{\text{out}}}$. Given a dataset $\mathcal{D} = \{(x_i, y_i)\}_{i \in [N]} \subset \mathcal{X} \times \mathcal{Y}$ and a loss function $\ell\colon \mathcal{Y} \times \mathcal{Y} \to \mathbb{R}_{\geq 0}$, training the model amounts to minimizing the cost

$$L_{\mathcal{D}}(\theta) = 1/N \cdot \sum_{i=1}^{N} \ell\big(f(x_i; \theta), y_i\big),$$

with respect to $\theta \in \Theta$. In practice, the training dynamics are often modeled by the Euclidean gradient flow

$$\dot{\theta}(t) = -\nabla L_{\mathcal{D}}\big(\theta(t)\big), \quad \theta(0) \in \Theta. \qquad (1)$$

A function $h\colon \Theta \to \mathbb{R}$ is called a *conservation law* for $f$ if it remains invariant along training trajectories. More precisely, for every solution $\theta(\cdot)$ of the ordinary differential equation (1) with any initialization $\theta(0) \in \Theta$, one has

$$h(\theta(t)) = h(\theta(0)), \quad \text{for all } t \geq 0.$$

**Remark 2.1.** In practice, training often includes weight decay. However, Marcotte et al. (2025) showed that conserved functions correspond with and without weight decay. Thus, we omit weight decay and focus on the flow ODE (1).

**Intractability.** While the definition of a conservation law $h$ is conceptually clear, characterizing *all* such functions is highly challenging. The main difficulty is that the training objective depends on an unknown dataset $\mathcal{D}$, making it impossible to formulate $L_{\mathcal{D}}$ in a universal way and rendering the problem intractable in full generality. To address this issue, Marcotte et al. (2023) proposed a weaker yet more tractable notion of conservation laws. Specifically, they define $h$ to be a conservation law for $f$ if $h(\theta(t)) = h(\theta(0))$ for every solution $\theta(\cdot)$ of the ODE (1), for any initialization, and for *any dataset* $\mathcal{D} \subset \mathcal{X} \times \mathcal{Y}$. They further restrict attention to functions $h \in \mathcal{C}^1(\Theta, \mathbb{R})$, which leads to the following orthogonality characterization for $\mathcal{C}^1$ conservation laws. Assume that $\ell(z, y)$ is $\mathcal{C}^2$-differentiable in $z$ for each $y \in \mathcal{Y}$. Then $h$ is a conservation law for $f$ if and only if $\nabla_\theta h(\theta) \perp \mathcal{W}_\theta^{f,\ell}$ for all $\theta \in \Theta$, where

$$\mathcal{W}_\theta^{f,\ell} := \operatorname*{span}_{(x,y) \in \mathcal{X}_\theta \times \mathcal{Y}} \left\{ \nabla_\theta f(x; \theta) \cdot \nabla_z \ell\big(f(x; \theta), y\big) \right\} \subseteq \mathbb{R}^D.$$

Note that $\nabla_\theta f(x; \theta) \in \mathbb{R}^{D \times d_{\text{out}}}$ and $\nabla_z \ell(z, y) \in \mathbb{R}^{d_{\text{out}}}$, so their product lies in $\mathbb{R}^D$. $\mathcal{X}_\theta$ denotes the set of $x \in \mathbb{R}^{d_{\text{in}}}$ such that $f(x; \cdot)$ is $\mathcal{C}^2$-differentiable in a neighborhood of $\theta$.

Following Marcotte et al. (2025), we impose a structural assumption on the loss $\ell$. For each $z \in \mathbb{R}^{d_{\text{out}}}$, define

$$\mathcal{V}_\ell(z) := \operatorname{span}_y \nabla_z \ell(z, y) \subseteq \mathbb{R}^{d_{\text{out}}}.$$

They assume that $\mathcal{V}_\ell(z)$ is independent of $z$, so that one may simply write $\mathcal{V}_\ell(z) = \mathcal{V}_\ell$ for all $z$. Under this assumption, it follows that for every $\theta \in \Theta$, one has

$$\mathcal{W}_\theta^{f,\ell} = \operatorname*{span}_{x \in \mathcal{X}_\theta, \; v \in \mathcal{V}_\ell} \{\nabla_\theta f(x; \theta) \cdot v\} \subseteq \mathbb{R}^D.$$

**Remark 2.2.** Although assumptions such as $\mathcal{C}^1/\mathcal{C}^2$ regularity and the independence of $\mathcal{V}_\ell(z)$ are mild in practice, requiring conservation laws to hold for *any dataset* may appear overly restrictive. However, allowing dataset dependence introduces the challenge of imposing meaningful constraints without rendering the problem ill-posed or trivial. Dataset-independence is therefore best viewed as a principled design choice, isolating the implicit bias induced solely by the architecture and optimization dynamics.

## 3. Characterization of Conservation Laws

Under the setting in Section 2, Marcotte et al. (2023; 2025) pursued their analysis through a Lie-theoretic framework. However, in most of their analysis, $\mathcal{V}_\ell$ is taken to be $\mathbb{R}^{d_{\text{out}}}$, as this assumption holds for a broad class of commonly used regression losses. Throughout the remainder of this paper, we adopt the assumptions introduced in Section 2, and in particular we also assume that $\mathcal{V}_\ell = \mathbb{R}^{d_{\text{out}}}$. However, our approach differs in spirit. Rather than relying on Lie-theoretic machinery, we return to first principles and ask:

*What does it actually mean to characterize all conservation laws for a given model?*

One way to interpret this question is as follows. Characterizing all conservation laws amounts to identifying all $\mathcal{C}^1$ functions $h \colon \Theta \to \mathbb{R}$ such that

$$\nabla_\theta h(\theta) \perp \operatorname*{span}_{x \in \mathcal{X}_\theta, \; v \in \mathbb{R}^{d_{\text{out}}}} \{\nabla_\theta f(x; \theta) \cdot v\}.$$

Equivalently, for every $\theta \in \Theta$, $x \in \mathcal{X}_\theta$, and $v \in \mathbb{R}^{d_{\text{out}}}$,

$$\langle \nabla_\theta h(\theta), \nabla_\theta f(x; \theta) \cdot v \rangle = 0,$$

where $\langle \cdot, \cdot \rangle$ denotes the dot product. Writing $f = (f_1, \ldots, f_{d_{\text{out}}})$, the condition above becomes: For all $i \in [d_{\text{out}}]$, $\theta \in \Theta$, and $x \in \mathcal{X}_\theta$, one has

$$\langle \nabla_\theta h(\theta), \nabla_\theta f_i(x; \theta) \rangle = 0. \tag{2}$$

Since $h$ is defined solely on the parameter space $\Theta$, whereas Equation (2) depends explicitly on the input $x$, characterizing all such functions $h$ can be viewed as reducing Equation (2) to constraints on $h$ that are *independent of $x$*. These resulting constraints typically take the form of a partial differential equation (PDE) for $h$, whose solvability depends on the structural complexity of the underlying architecture.

In general, such PDE systems need not admit closed-form or tractable solutions. Therefore, we regard the characterization problem as successful if Equation (2) can be simplified into $x$-independent conditions that fully determine $h$. As we will show in the subsequent sections, for all architectures considered in this paper, this reduction leads to a particularly clean and explicit description of the conservation laws.

**A Toy Example.** Consider the model $f(x; a, b) = abx$, where $x \in \mathbb{R}$ and $(a, b) \in \mathbb{R}^2$ are the parameters. A conservation law $h \colon \mathbb{R}^2 \to \mathbb{R}$ for this model must satisfy Equation (2), which in this case becomes

$$\begin{aligned} 0 &= \big\langle (\partial_a h(a,b), \partial_b h(a,b)), (bx, ax) \big\rangle \\ &= \partial_a h(a,b) \cdot bx + \partial_b h(a,b) \cdot ax, \end{aligned}$$

Cancelling $x$ yields the $x$-independent constraint

$$0 = \partial_a h(a,b) \cdot b + \partial_b h(a,b) \cdot a, \text{ for all } (a,b) \in \mathbb{R}^2.$$

To solve this first-order PDE, consider the characteristic curve $\gamma(t) = (a(t), b(t))$ satisfying $\dot{a} = b$ and $\dot{b} = a$. By the chain rule, we obtain

$$\frac{d}{dt} h(\gamma(t)) = \partial_a h \cdot \dot{a} + \partial_b h \cdot \dot{b} = \partial_a h \cdot b + \partial_b h \cdot a = 0.$$

Therefore, $h(\gamma(t))$ remains constant along every characteristic curve $\gamma$. Moreover, along such a curve, we have

$$\frac{d}{dt}(a^2 - b^2) = 2a\dot{a} - 2b\dot{b} = 2ab - 2ba = 0.$$

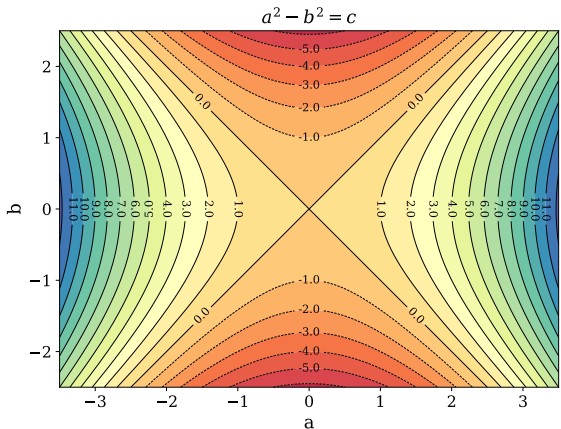

*Figure 1.* The disconnectedness of the level set $\{a^2 - b^2 = c\}$.

Thus, $a^2 - b^2$ is an invariant of the characteristic flow. It follows that $h$ must be constant on each connected component of the level sets $\{a^2 - b^2 = c\}$ for $c \in \mathbb{R}^2$.

**Remark 3.1.** For readers who may wonder, this constancy condition does not imply that $h$ can necessarily be expressed as a function of $a^2 - b^2$, even when $h$ is $\mathcal{C}^1$. The reason is that the level set $\{a^2 - b^2 = c\}$ is generally disconnected. Figure 1 illustrates this phenomenon.

Nevertheless, this example illustrates that identifying such a characteristic quantity–here $a^2 - b^2$–from the resulting $x$-independent constraints is often sufficient to capture the essential structure of $h$, without requiring a full analysis of finer details (such as constancy on individual connected components). In this sense, $a^2 - b^2$ can be viewed as a *characteristic invariant*, associated with the model $f$.

The solution strategy developed in this section–namely, reducing the conservation law condition to model-input-independent constraints on $h$ and identifying the resulting characteristic invariants–will serve as the core framework for our analysis in the subsequent sections.

# 4. Conservation Laws Characterization for Modern Neural Architectures

This section presents our main results on the complete characterization of conservation laws for several modern neural architectures. We begin with feedforward networks (FFNs), focusing on widely used activation functions such as GeLU, SiLU, and SwiGLU. We then investigate the attention mechanism, emphasizing the multihead setting and the influence of positional encodings. Finally, we study Mixture-of-Experts (MoE) models, covering both classical softmax gating and more recent alternatives, as well as both dense and sparse gating regimes.

## 4.1. Conservation Laws for Feedforward Networks

Since most FFN components used in practice employ only a single hidden layer, we restrict our analysis to this setting.

**Feedforward Networks with** GELU **and** SiLU **Activations.** An FFN with GELU activation is a map $f \colon \mathbb{R}^d \to \mathbb{R}^d$ parameterized by weight matrices $A \in \mathbb{R}^{d \times d_1}$ and $B \in \mathbb{R}^{d_1 \times d}$, defined by

$$f(x; A, B) = A \cdot \text{GELU}(Bx). \tag{3}$$

Here, the GELU activation is given by $\text{GELU}(z) = z \, \Phi(z)$, where $\Phi$ denotes the standard normal CDF,

$$\Phi(z) = \frac{1}{\sqrt{2\pi}} \int_{-\infty}^{z} e^{-t^2/2} \, dt.$$

Similarly, an FFN with SiLU activation is defined in the same form as Equation (3), with GELU replaced by the SiLU activation $\text{SiLU}(z) = z \, \sigma(z)$, where

$$\sigma(z) = \frac{1}{1 + e^{-z}} \quad \text{is the sigmoid function.}$$

The parameter spaces of these networks are given by

$$\Theta_{\text{GELU}} = \Theta_{\text{SiLU}} := \mathbb{R}^{d \times d_1} \times \mathbb{R}^{d_1 \times d}.$$

We now present our characterization of conservation laws for these architectures. The proof of the following Theorem 4.1 is provided in Appendices A.1 and A.2.

**Theorem 4.1** (Conservation laws for FFNs with GELU or SiLU)**.** *All conservation laws for FFNs with* GELU *or* SiLU *activations are constant functions.*

**Feedforward Networks with SwiGLU Activation.** An FFN with SwiGLU activation is a map $f \colon \mathbb{R}^d \to \mathbb{R}^d$ parameterized by $A \in \mathbb{R}^{d \times d_1}$ and $B, C \in \mathbb{R}^{d_1 \times d}$, given by

$$f(x; A, B, C) = A \cdot \big(\text{SiLU}(Bx) \odot (Cx)\big), \tag{4}$$

where $\odot$ denotes the Hadamard product. The associated parameter space of this architecture is

$$\Theta_{\text{SwiGLU}} := \mathbb{R}^{d \times d_1} \times \mathbb{R}^{d_1 \times d} \times \mathbb{R}^{d_1 \times d}.$$

The following result provides a complete characterization of conservation laws for SwiGLU feedforward networks. Unlike the GeLU and SiLU cases, the SwiGLU architecture exhibits a distinct structural form, in which the multiplicative interaction between the parameter matrices $A$ and $C$ gives rise to nontrivial conservation laws.

**Theorem 4.2** (Conservation laws for FFNs with SwiGLU)**.** *Let $h \colon \Theta_{\text{SwiGLU}} \to \mathbb{R}$ be a conservation law for FFNs with SwiGLU defined as in Equation* (4)*. Then, for all $i \in [d_1]$,*

$$\nabla_B h = 0, \; and \;\; \nabla_{A_{:,i}} h \cdot C_{i,:} + A_{:,i} \cdot \nabla_{C_{i,:}} h = 0.$$

*Consequently, $h$ is constant on each connected component of the level sets determined by the invariants $\|A_{:,i}\|^2 - \|C_{i,:}\|^2, i \in [d_1]$.*

The proof of Theorem 4.2 is provided in Appendix A.3.

## 4.2. Conservation Laws for Multihead Attention

This section provides a complete characterization of conservation laws for the multihead attention (MHA) mechanism, thereby resolving an open problem in the literature (Marcotte et al., 2025). We further extend the analysis to settings incorporating positional encodings (PEs). In particular, we show that sinusoidal PEs preserve the conservation structure of vanilla attention, whereas rotary positional encodings (RoPE) fundamentally modify the internal structure of attention, giving rise to a distinct class of conservation laws.

Let $L, n \in \mathbb{N}$ denote the sequence length and the number of attention heads, respectively. The space of all sequences of $d$-dimensional token embeddings is given by $\bigsqcup_{L=1}^{\infty} \mathbb{R}^{L \times d}$.

**Multihead Attention without Positional Encoding.** Let $d_h$ denote the head dimension (usually $d_h = d/n$). For each head $i \in [n]$, we consider projection matrices $Q_i, K_i, V_i, O_i \in \mathbb{R}^{d \times d_h}$. The multihead attention (MHA) transforms an input sequence $\mathbf{x} = (x_1, \ldots, x_L)^\top \in \mathbb{R}^{L \times d}$ into an output sequence in $\mathbb{R}^{L \times d}$ according to:

$$\text{MHA}\big(\mathbf{x}; \{Q_i, K_i, V_i, O_i\}_{i=1}^n\big) \qquad (5)$$
$$:= \sum_{i=1}^n \text{softmax}\left((\mathbf{x}Q_i)(\mathbf{x}K_i)^\top\right) \cdot \mathbf{x}V_i O_i^\top.$$

The parameter space of the MHA map is given by

$$\Theta_{\text{MHA}} := \left(\mathbb{R}^{d \times d_h}\right)^{4n}. \qquad (6)$$

We now present our characterization of conservation laws for multihead attention, thereby confirming the conjecture of Marcotte et al. (2025) that all such conservation laws are characterized by the quantities $Q_i^\top Q_i - K_i^\top K_i$ and $V_i^\top V_i - O_i^\top O_i$ for $i \in [n]$.

**Theorem 4.3** (Conservation laws for Multihead Attention). *Let $h \colon \Theta_{\text{MHA}} \to \mathbb{R}$ be a conservation laws for the MHA map defined as in Equation (5). Then, for all $i \in [n]$,*

$$0 = K_i \cdot (\nabla_{Q_i} h)^\top + (\nabla_{K_i} h) \cdot Q_i^\top, \text{ and}$$
$$0 = O_i \cdot (\nabla_{V_i} h)^\top + (\nabla_{O_i} h) \cdot V_i^\top.$$

*Consequently, $h$ is constant on each connected component of the level sets determined by the invariants $Q_i^\top Q_i - K_i^\top K_i$ and $V_i^\top V_i - O_i^\top O_i$ for $i \in [n]$.*

The proof of Theorem 4.3 is provided in Appendix B.1.

We next investigate how incorporating PEs alters the functional behavior of MHA, thereby inducing a corresponding change in the associated conservation laws.

**Sinusoidal Encoding.** Let $\mathbf{p} = \{p_i\}_{i=1}^{\infty} \subset \mathbb{R}^d$ denote the sequence of positional vectors used to encode positional information. In the original Transformer architecture (Vaswani et al., 2017), each $p_i$ is defined through a fixed combination of sine and cosine functions at different frequencies. For an input sequence $\mathbf{x} \in \mathcal{S}$ of length $L$, the positional encoding is incorporated by addition, namely

$$\mathbf{x} \mapsto \mathbf{x} + \mathbf{p} := (x_1 + p_1, \ldots, x_L + p_L)^\top.$$

This operation amounts to a fixed shift of the input sequence and does not alter the internal structure of the MHA map. Moreover, since this shift is bijective, the problem of characterizing conservation laws for MHA with sinusoidal positional encoding reduces directly to the vanilla setting.

**Rotary Encoding.** We next recall the Rotary Positional Encoding (RoPE) introduced by Su et al. (2024). Assume that the head dimension $d_h = 2m$ is even. Define the block-diagonal rotation matrix $R \in \mathbb{R}^{d_h \times d_h}$ by

$$R := \text{diag}\left(\begin{bmatrix} \cos(\varphi_i) & -\sin(\varphi_i) \\ \sin(\varphi_i) & \cos(\varphi_i) \end{bmatrix}_{i \in [m]}\right),$$

where the rotation frequencies are given by $\varphi_i = 10000^{-(i-1)/m}$ for each $i \in [m]$. For any integer position index $p$, we further define $R_p := R^p$. Multihead attention equipped with RoPE is then obtained by applying these position-dependent rotations to the query and key representations, as defined below.

$$\text{MHA}^{\text{RoPE}}\big(\mathbf{x}; \{Q_i, K_i, V_i, O_i\}_{i=1}^n\big) \qquad (7)$$
$$:= \sum_{i=1}^n \text{softmax}\left[x_p Q_i R_{p-q} K_i^\top x_q^\top\right]_{p,q \in [L]} \cdot \mathbf{x}V_i O_i^\top.$$

Although the parameter space of $\text{MHA}_{\text{RoPE}}$ coincides with that of the standard MHA map in Equation (6), its mechanism for incorporating positional information is fundamentally different. In RoPE, positional information is injected via the rotation matrix $R$ after projection into the head dimension, leading to a nontrivial departure from the case of absolute positional encodings. Consequently, the internal structure of the attention operator is no longer related to vanilla MHA by a simple input translation, as in the sinusoidal setting, but is instead substantially altered.

We now present our results on conservation laws for $\text{MHA}_{\text{RoPE}}$. To this end, we decompose the projection matrices $Q_i$ and $K_i$ into blocks of size $d \times 2$:

$$Q_i = \left[Q_i^{(1)}, \ldots, Q_i^{(m)}\right], \quad K_i = \left[K_i^{(1)}, \ldots, K_i^{(m)}\right],$$

where $Q_i^{(j)}, K_i^{(j)} \in \mathbb{R}^{d \times 2}$ consists of the $(2j-1)$-st and $(2j)$-th columns of $Q_i$ and $K_i$, respectively. Finally, we introduce the canonical complex structure matrix $J = \begin{bmatrix} 0 & -1 \\ 1 & 0 \end{bmatrix}$.

**Theorem 4.4** (Conservation laws for Multihead Attention with RoPE). *Let $h \colon \Theta_{\text{MHA}} \to \mathbb{R}$ be a conservation laws for the $\text{MHA}^{\text{RoPE}}$ map defined as in Equation (7). Then, for all*

$i \in [n]$ and $j \in [m]$,

$$0 = K_i^{(j)} \cdot \left(\nabla_{Q_i^{(j)}} h\right)^\top + \nabla_{K_i^{(j)}} h \cdot \left(Q_i^{(j)}\right)^\top,$$

$$0 = K_i^{(j)} \cdot J \cdot \left(\nabla_{Q_i^{(j)}} h\right)^\top + \nabla_{K_i^{(j)}} h \cdot J \cdot \left(Q_i^{(j)}\right)^\top;$$

and for all $i \in [n]$,

$$0 = O_i \cdot (\nabla_{V_i} h)^\top + (\nabla_{O_i} h) \cdot V_i^\top.$$

*Consequently, $h$ is constant on each connected component of level sets determined by the invariants $\|Q_i^{(j)}\|_F^2 - \|K_i^{(j)}\|_F^2$ for $i \in [n], j \in [m]$ and $V_i^\top V_i - O_i^\top O_i$ for $i \in [n]$.*

The proof of Theorem 4.4 is provided in Appendix B.2.

### 4.3. Conservation Laws for Mixture-of-Experts

A Mixture-of-Experts (MoE) model combines multiple expert networks via a gating mechanism that assigns input-dependent weights. Since most modern MoE implementations employ feedforward experts with SwiGLU activations, we restrict our analysis to this setting.

Let $n$ denoting the number of expert.

**Dense Mixture-of-Experts.** For each $i \in [n]$, let

$$\theta_i = \left(A^{(i)}, B^{(i)}, C^{(i)}\right) \in \Theta_{\text{SwiGLU}}$$

denote the parameters of the $i$-th expert. We define the corresponding expert network as the map $\text{E}(\cdot; \theta_i)$, which is a feedforward network with SwiGLU activation parameterized by $\theta_i$, as in Equation (4). Let $W \in \mathbb{R}^{n \times d}$ denote the gating parameter matrix. The *Mixture-of-Experts with dense gating* is defined as the function $\text{MoE}: \mathbb{R}^d \to \mathbb{R}^d$ given by

$$\text{MoE}\left(x; W, \{\theta_i\}_{i=1}^n\right) = \sum_{i=1}^n g_i(x; W) \cdot \text{E}(x; \theta_i), \quad (8)$$

where the gating weight $g_i(x; W)$ is defined by

$$g_i(x; W) := \text{softmax}_i(Wx) = e^{W_i x} / \sum_{p=1}^n e^{W_p x},$$

with $W_i := W_{i,:}$ denoting the $i$-th row of $W$. The score $g_i(x; W)$ specifies the relative contribution of the $i$-th expert to the final output. The parameter space of MoE is given by

$$\Theta_{\text{MoE}} = \mathbb{R}^{n \times d} \times (\Theta_{\text{SwiGLU}})^n.$$

The following theorem provides a characterization of conservation laws for MoE, showing that invariants are localized at the level of individual expert parameters, together with an additional invariant arising from the gating mechanism.

**Theorem 4.5** (Conservation laws for Dense Mixture-of-Experts). *Let $h: \Theta_{\text{MoE}} \to \mathbb{R}$ be a conservation laws for the MoE map defined as in Equation (8). Then*

*1. For $i \in [n]$, for the expert parameters $\theta_i$, $h$ satisfies the same constraints as in Theorem 4.2; and,*

*2. For the gating parameters $W$, $h$ satisfies the constraint*

$$\nabla_{W_1} h = \nabla_{W_2} h = \ldots = \nabla_{W_n} h.$$

*Consequently, $h$ is constant on each connected component of the level sets determined by the invariants $\|A_{:,j}^{(i)}\|^2 - \|C_{j,:}^{(i)}\|^2$ for $i \in [n], j \in [d_1]$, and $\sum_{i=1}^n W_i$.*

The proof of Theorem 4.5 is provided in Appendix C.2.

**Sparse Mixture-of-Experts.** Let $k \leq n$ be the number of activated experts. The Top-$k$ operator is defined by

$$\text{Top-}k(z) = \{i_1, \ldots, i_k\} \quad \text{for } z = (z_1, \ldots, z_n) \in \mathbb{R}^n,$$

where $i_1, \ldots, i_k$ are the indices of the $k$ largest entries of $z$. Ties are resolved by selecting indices in increasing order. The *Sparse Mixture-of-Experts* (SMoE) architecture is a variant of MoE in which, for each input $x$, only the $k$ experts with the highest gating scores are activated. The model shares the same parameter space $\Theta_{\text{MoE}}$ as the dense MoE, and is defined by

$$\text{SMoE}(x; \theta) := \sum_{i \in \text{Top-}k(Wx)} \bar{g}_i(x, W) \cdot \text{E}(x; \theta_i),$$

where $\bar{g}_i$ is obtained by normalizing the original softmax scores over the active experts:

$$\bar{g}_i(x; W) = g_i(x; W) / \sum_{i \in \text{Top-}k(Wx)} g_i(x; W).$$

In other words, the Top-$k$ selects the $k$ highest-scoring experts used to compute the output. The following result characterizes the conservation laws of SMoE for $k > 1$, and shows that they coincide with those of the dense MoE case.

**Theorem 4.6** (Conservation laws for Sparse Mixture-of-Experts). *Let $k > 1$. Every conservation law of the SMoE map is also a conservation law of the corresponding dense MoE map. In particular, such conservation laws satisfy exactly the same constraints as those given in Theorem 4.5.*

The proof of Theorem 4.6 is provided in Appendix C.3.

**Mixture-of-Experts with Sigmoid Gating.** In several recent variants of MoE, the standard softmax gating mechanism is replaced by a normalized sigmoid gating function. The resulting MoE map is defined as in Equation (8), except that the gating weights $g_i(x; W)$ are now given by

$$g_i(x; W) := \sigma(W^{(i)} x) / \sum_{p=1}^n \sigma(W^{(p)} x),$$

where $\sigma$ denotes the sigmoid function. The corresponding SMoE map is defined analogously. The following result shows that conservation laws under sigmoid gating coincide with those of the softmax-gated case.

**Theorem 4.7** (Conservation laws for sigmoid variants of Mixture-of-Experts). *Every conservation laws for sigmoid-gated* MoE *and* SMoE *maps coincides with that of the standard softmax-gated* MoE. *In particular, such conservation laws in the sigmoid-gated setting satisfy exactly the same constraints as those given in Theorem 4.5.*

The proof of Theorem 4.7 is provided in Appendix C.4.

## 5. Overview of the Proof Strategy

We now provide a high-level sketch of the main ideas underlying the proofs of the characterization results in Section 4. As discussed in Section 3, our strategy is to reduce the conservation-law condition to model-input-independent constraints on $h$, and then to identify the resulting characteristic invariants. This perspective is reflected in each theorem in Section 4, where the condition yields a system of PDE-type constraints on $h$, together with the corresponding invariants. Note that our proofs require no additional assumptions beyond those stated in Sections 2 and 3.

**Proof Sketch.** The conservation-law condition is stated in Equation (2), which we recall here for simplicity in the case $d_{\text{in}} = d_{\text{out}} = d$. For all $i \in [d]$, $\theta \in \Theta$, and $x \in \mathcal{X}_\theta$,

$$\langle \nabla_\theta h(\theta), \nabla_\theta f_i(x; \theta) \rangle = 0.$$

For fixed $\theta \in \Theta$ and $i \in [d]$, define

$$F_i(\cdot; \theta) := \langle \nabla_\theta h(\theta), \nabla_\theta f_i(\cdot; \theta) \rangle.$$

Whenever the operations appearing in $f_i(\cdot; \theta)$ admit complex-analytic extensions, we may view $F_i(\cdot; \theta)$ as a complex-valued function on $\mathbb{C}^d$. In the cases considered in this paper (FFNs, MHA, and dense MoE), $f_i(\cdot; \theta)$ is meromorphic, and hence so is $F_i$. We then restrict $F_i$ to $\mathbb{C}$, typically by considering inputs of the form $x = te_j$, where $t$ ranges over an open subset of $\mathbb{R}$ such that $te_j \in \mathcal{X}_\theta$. By the identity theorem from complex analysis, a meromorphic function that vanishes on a set with an accumulation point must vanish identically. This implies that $F_i(\cdot; \theta)$ is identically zero along these subspaces, and hence yields input-independent constraints on $\nabla_\theta h$.

The SMoE case requires additional care, since the Top-$k$ routing introduces discontinuities at the boundaries where the active expert set changes. However, a refined argument shows that conservation laws for SMoE reduce to those of the corresponding dense MoE model.

Once the identically-zero property is established, the PDE constraints on $h$ are obtained by analyzing the pole structure of the meromorphic components of $F_i(\cdot; \theta)$. A key tool is the following principle: if a linear combination of meromorphic functions vanishes identically, then comparing pole orders forces the coefficients associated with dominant

poles to vanish. This argument repeatedly yields explicit constraints on the gradients of $h$.

**Remark 5.1.** It is worth emphasizing that the characterization result for MHA$^{\text{RoPE}}$ is highly nontrivial and substantially more challenging than in the standard MHA setting. The main difficulty stems from the fact that the interaction term $Q_i R_{p-q} K_i^\top$ in the attention scores depends on token positions, whereas in vanilla MHA the corresponding quantity is position-independent. This dependence makes the pole analysis described above more delicate.

**Remark 5.2.** The additional condition $k > 1$ in Theorem 4.6 is used to connect the regions associated with different active expert sets. This connectivity, in turn, forces the gating gradients $\nabla_{W_i} h$ to coincide across all experts.

While the proposed framework is broadly applicable, a limitation is that it does not straightforwardly extend to certain additional components of deep models. For instance, stacking multiple layers may introduce functions such as $\sqrt{x}$ (through layer normalization) or $e^{1/x}$ (through compositions of attention layers). Such functions do not admit meromorphic extensions to the entire complex plane, as they involve branch points or essential singularities. Addressing these architectures would require a more refined analysis of singular behavior, for example by extending $F_i$ only to appropriate regions of $\mathbb{C}^d$ rather than the whole space. We leave this direction as a promising avenue for future work.

## 6. Empirical Evidence of Conservation Laws

We empirically examine the degree to which conservation laws that hold in the continuous-time gradient flow setting persist under discrete-time optimization. Our experiments are motivated by Proposition 5.1 of Marcotte et al. (2025), which characterizes the accumulation of conservation error under stochastic gradient descent (SGD). In particular, for a quantity $h(\theta)$ that is conserved along the gradient flow, and assuming bounded Hessians ($C_h$) and gradients in expectation ($C_L$), this result establishes that

$$\mathbb{E}\big|h(\theta_k) - h(\theta_0)\big| \leq \frac{C_h C_L}{2} \sum_{i=0}^{k-1} \tau_i^2. \tag{9}$$

As a result, when a constant step size $\tau_k = \tau$ is used, the conservation error grows linearly with the number of iterations, $|h(\theta_k) - h(\theta_0)| = \mathcal{O}(\tau^2 k)$. In contrast, when employing a decaying step size $\tau_k = \tau_0/(k+1)$, which ensures convergence of SGD, the deviation from conservation remains uniformly bounded, $|h(\theta_k) - h(\theta_0)| = \mathcal{O}(\tau_0^2)$.

**Datasets and Models.** We validate our theoretical finding across language modeling and computer vision domains. For language modeling, we employ Qwen-3 architectures (Yang et al., 2025) on WikiText-103 (Merity et al., 2017) and Penn Treebank (Marcus et al., 1993), incorporating RoPE

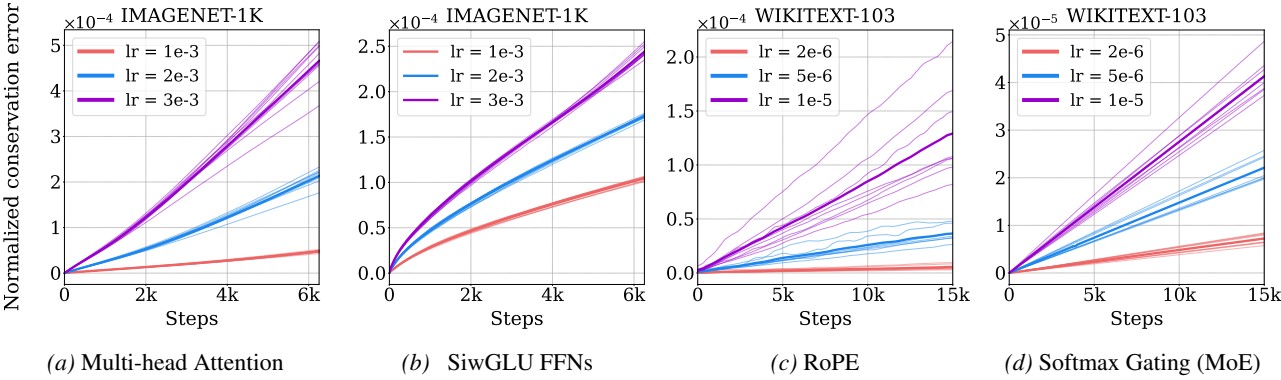

*Figure 2.* Conservation error scales with learning rate. (a-b) MHA and SwiGLU FFN conservation tracking on ImageNet-1K across three learning rates. (c-d) RoPE and MoE gating conservation on Wikitext-103.

and both Dense MoE and SMoE variants with softmax and normalized sigmoid gating. For computer vision, we utilize Vision Transformers (ViT) (Dosovitskiy et al., 2021) on CIFAR-10 (Krizhevsky et al., 2009) and ImageNet-1K (Deng et al., 2009), featuring absolute PEs and SwiGLU activations. Experimental details are detailed in Appendix D.

**Validation Protocol and Metrics.** We systematically vary optimization step size and random initialization stochasticity. For each configuration, we train 10 models with independent random seeds, monitoring training loss and conservation errors as functions of iteration $k$. For a network block with $N$ conservation laws $\{h_i\}_{i=1}^N$, we compute the block-level conservation error by averaging the relative deviations:

$$\epsilon_{\text{block}}(k) = \frac{1}{N} \sum_{i=1}^{N} \frac{|h_i(\theta_k) - h_i(\theta_0)|_2}{|h_i(\theta_0)|_2} \qquad (10)$$

We track these metrics across key architectural components: FFN, MHA, MHA with RoPE, and MoE gating.

**Non-conserved Quantities.** To provide a qualitative baseline, we also monitor representative non-conserved quantities. While lacking theoretical bounds for direct comparison, non-conserved quantities exhibit large deviations under small parameter changes, unlike the strictly bounded evolution of conservation laws. Detailed specifications of the non-conserved quantities tracked for each architectural component are provided in Appendix E.

**Learning Rate Scheduler.** We employ a linear learning rate scheduler to assess the long-term evolution of conservation laws under sustained optimization dynamics. Unlike schedulers that decay to zero, which yield trivial error bounds as parameter updates vanish, a linear schedule maintains active learning. This approach prevents the artificial saturation of conservation deviations, thereby providing a more rigorous validation of our theoretical predictions.

**Results.** We empirically validate conservation laws under two experimental regimes. First, to approximate continuous-

time gradient flow with minimal stochasticity, we employ full-batch gradient descent on small-scale datasets (CIFAR-10 and PTB). Second, we utilize mini-batch SGD on large-scale datasets (ImageNet-1K and WikiText-103), thereby testing the bounds of Equation (9) under practical training conditions. Our analysis assesses conservation laws across MHA, MHA with RoPE, SwiGLU FFNs, and MoE Gating (covering both softmax and normalized sigmoid mechanisms) within dense and spare MoE architectures. Figures 2 and 3 demonstrate that conservation laws maintain their bounded behavior regardless of training regimes, architectures, and datasets. In both settings, we observe consistent $O(\tau^2 k)$ scaling for multi-head attention (a), SwiGLU feedforward networks (b), RoPE (c), and MoE softmax gating (d), with higher learning rates inducing proportionally larger bounds as predicted by Equation (9). This consistency confirms that conservation laws are fundamental properties of the optimization geometry rather than artifacts of specific network designs or training procedures. Normalized sigmoid gating results are detailed in Appendix E.

## 7. Conclusion

This paper develops a theoretical framework and establishes complete characterizations of conservation laws for modern neural architectures widely used in deep learning, including FFNs with GeLU, SiLU, and SwiGLU activations, MHA with positional encodings, and MoE under diverse gating designs. We believe these results advance the theoretical understanding of conserved quantities in training dynamics. A limitation of our approach is that its reliance on complex-analytic techniques requires further refinement to handle additional mechanisms such as layer normalization or deep layer compositions, as discussed in Section 5. Moreover, the data-independence assumption, adopted from prior work (Marcotte et al., 2025), may be seen as restrictive; however, it is best viewed as a principled design choice that isolates the implicit bias induced solely by the architecture and optimization dynamics (see Remark 2.2). We leave these directions as promising avenues for future research.

## Acknowledgements

This research / project is supported by the National Research Foundation Singapore under the AI Singapore Programme (AISG Award No: AISG2-TC-2023-012-SGIL). This research / project is supported by the Ministry of Education, Singapore, under the Academic Research Fund Tier 1 (FY2023) (A-8002040-00-00, A-8002039-00-00). This research / project is also supported by the NUS Presidential Young Professorship Award (A-0009807-01-00), the NUS Artificial Intelligence Institute–Seed Funding (A-8003062-00-00), and the Cross Faculty Grant 2025, CFG25 - 012 (A-8004460-00-00).

Tuan Dam is funded by Vietnam National Foundation for Science and Technology Development (NAFOSTED) under grant number 102.01-2025.47.

## Impact Statement

This paper presents work whose goal is to advance the field of machine learning. There are many potential societal consequences of our work, none of which we feel must be specifically highlighted here.

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

# Appendix of "Conservation Laws for Modern Neural Architectures"

**Table of Contents**

## A. Theoretical Proofs of Conservation Laws for Feedforward Networks with Modern Activations

As noted in Section 3, it suffices to consider feedforward networks with real-valued (scalar) outputs.

### A.1. Conservation Laws for Feedforward Networks with GELU Activation – Theorem 4.1 (GELU part)

**Feedforward Networks with GELU Activation.** Recall that, an FFN with GELU is a function $f\colon \mathbb{R}^d \to \mathbb{R}$ of the form

$$f(x; A, B) = A \cdot \mathrm{GELU}(Bx), \tag{11}$$

where the GELU activation is defined by:

$$\mathrm{GELU}(z) = z\Phi(z), \quad \text{where } \Phi(z) = \frac{1}{\sqrt{2\pi}} \int_{-\infty}^{z} e^{-t^2/2} dt \text{ is the standard normal CDF.} \tag{12}$$

**Parameter Space.** The function $f$ is parameterized by:

$$\theta = (A, B) \in \Theta_{\mathrm{GELU}} := \mathbb{R}^{1 \times d_1} \times \mathbb{R}^{d_1 \times d}. \tag{13}$$

We now present the proof of the conservation laws for feedforward networks with GELU activation, which constitutes the first part of Theorem 4.1.

*Proof.* By definition, the $f$ is $\mathcal{C}^1$ with respect to $A, B$. The gradients of $f$ with respect to $A, B$ are given by:

$$\nabla_A f = \mathrm{GELU}(Bx)^\top, \qquad \nabla_B f = (A^\top \odot \mathrm{GELU}'(Bx))x^\top. \tag{14}$$

Let $h\colon \Theta_{\text{GELU}} \to \mathbb{R}$ be a conservation law for $f$. We have, for all $(A, B) \in \Theta_{\text{GELU}}$ and $x \in \mathbb{R}^d$,

$$0 = \langle \nabla f, \nabla h \rangle = \langle \nabla_A f, \nabla_A h \rangle + \langle \nabla_B f, \nabla_B h \rangle. \tag{15}$$

Substituting the gradients of $f$, we obtain

$$0 = \left\langle \text{GELU}(Bx)^\top, \nabla_A h \right\rangle + \left\langle \left( A^\top \odot \text{GELU}'(Bx) \right) x^\top, \nabla_B h \right\rangle. \tag{16}$$

For $i \in [d_1]$, let $b_i \in \mathbb{R}^d$ be the $i$-th row of $B$. Then Equation (16) is equivalent to:

$$\sum_{i=1}^{d_1} \text{GELU}(b_i x) \cdot \partial_{A_i} h + \sum_{i=1}^{d_1} \sum_{j=1}^{d} A_i \text{GELU}'(b_i x) x_j \cdot \partial_{B_{ij}} h = 0, \tag{17}$$

for all $x \in \mathbb{R}^d$. For each $j \in [d]$, set $x = t e_j$ where $t \in \mathbb{R}$ and $e_j \in \mathbb{R}^{d \times 1}$ is the $j$-th basis vector. Then $b_i x = B_{ij} t$, and Equation (17) becomes

$$\sum_{i=1}^{d_1} \text{GELU}(B_{ij}t) \cdot \partial_{A_i} h + \sum_{i=1}^{d_1} A_i t \text{GELU}'(B_{ij}t) \cdot \partial_{B_{ij}} h = 0, \tag{18}$$

for all $t \in \mathbb{R}$. Thus, for every $j \in [d]$, we obtain an analytic identity in one real variable $t$. We have the following lemma.

**Lemma A.1.** *Let $\alpha_1, \ldots, \alpha_h \in \mathbb{R}$ such that $\alpha_1^2, \ldots, \alpha_h^2$ are pairwise distinct and nonzero. The $2h$ functions*

$$t \mapsto \text{GELU}(\alpha_i t), \qquad t \mapsto t\text{GELU}'(\alpha_i t), \qquad for \ i \in [h], \tag{19}$$

*are linearly independent.*

*Proof.* Since GELU is analytic, we express $\sigma$ and $\sigma'$ by their power series, as follows:

$$\text{GELU}(z) = \frac{1}{2} z + \frac{1}{\sqrt{2\pi}} \sum_{n=0}^{\infty} \frac{(-1)^n}{2^n n!(2n+1)} z^{2n+2}, \tag{20}$$

$$\text{GELU}'(z) = \frac{1}{2} + \frac{1}{\sqrt{2\pi}} \sum_{n=0}^{\infty} \frac{(-1)^n (2n+2)}{2^n n!(2n+1)} z^{2n+1}. \tag{21}$$

For simplicity, set the coefficients of these two series as follows:

$$\text{GELU}(z) = \sum_{n=0}^{\infty} \beta_n z^n, \qquad \text{GELU}'(z) = \sum_{n=0}^{\infty} (n+1)\beta_{n+1} z^n. \tag{22}$$

From Equation (20), we have $\beta_1 \neq 0$, $\gamma_0 \neq 0$, and $\beta_{2m} \neq 0$, $\gamma_{2m-1} \neq 0$ for all $m \geq 1$. The rest are equal to 0. Now, assume that there exists real numbers $u_i$ and $v_i$ for $i \in [h]$ such that

$$\sum_{i=1}^{d_1} u_i \text{GELU}(\alpha_i t) + \sum_{i=1}^{d_1} v_i \, t\text{GELU}'(\alpha_i t) = 0. \tag{23}$$

This is an analytic function in $t$ that is identical to 0. Thus, for all $m \geq 0$, the coefficient of $t^m$ must vanish. In particular, for $m > 0$, the coefficient of $t^{2m}$ is equal to 0, i.e.

$$\beta_{2m} \sum_{i=1}^{h} u_i \alpha_i^{2m} + \beta_{2m} 2m \sum_{i=1}^{h} v_i \alpha_i^{2m-1} = 0. \tag{24}$$

Since $\beta_{2m} \neq 0$, by setting $w_i = 2v_i/\alpha_i$ and $\gamma_i = \alpha_i^2$, we have

$$\sum_{i=1}^{h} (u_i + m w_i) \gamma_i^m = 0. \tag{25}$$

Consider the following power series:

$$s(z) = \sum_{m=1}^{\infty} \left( \sum_{i=1}^{h} (u_i + m w_i) \gamma_i^m \right) z^m. \tag{26}$$

By Equation (25), we have $s$ is identical to 0. Moreover,

$$
\begin{aligned}
0 = s(z) &= \sum_{m=1}^{\infty} \left( \sum_{i=1}^{h} (u_i + m w_i) \gamma_i^m \right) z^m \\
&= \sum_{i=1}^{h} u_i \sum_{m=1}^{\infty} (\gamma_i z)^m + \sum_{i=1}^{h} w_i \sum_{m=1}^{\infty} m (\gamma_i z)^m = \sum_{i=1}^{h} u_i \frac{\gamma_i z}{1 - \gamma_i z} + \sum_{i=1}^{h} w_i \frac{\gamma_i z}{(1 - \gamma_i z)^2}.
\end{aligned} \tag{27}
$$

Note that $1/\gamma_i$ for $i \in [h]$ are $h$ distinct poles of the last expression of $s$ in Equation (27). Since $s$ is identically zero, its Laurent expansion at every pole must have vanishing principal part. Therefore both coefficients in the above expression must be zero, which implies $u_i = w_i = 0$. This leads to $u_i = v_i = 0$. We conclude that the $2h$ functions

$$t \mapsto \mathrm{GELU}(\alpha_i t), \qquad t \mapsto t\mathrm{GELU}'(\alpha_i t), \qquad \text{for } i \in [h], \tag{28}$$

are linearly independent. $\qquad\square$

Back to the problem. Define the following set

$$\mathbf{S}_1 := \{\theta = (A, B) : B_{ij}^2 \text{ is nonzero and pairwise distinct for all } i \in [d_1] \text{ and } j \in [d]\} \subset \Theta_{\mathrm{GELU}}. \tag{29}$$

From Equation (18), by applying Lemma A.1, we have:

$$\nabla_{A_i} h = A_i \cdot \nabla_{B_{ij}} h = 0, \tag{30}$$

for all $i \in [d_1]$, $j \in [d]$, and $\theta \in \mathbf{S}_1$. Define the following set

$$\mathbf{S}_2 := \{\theta = (A, B) : A_i \text{ is nonzero for all } i \in [d_1]\} \subset \Theta_{\mathrm{GELU}}. \tag{31}$$

From Equation (30), we have $\partial_{A_i} h = \partial_{B_{ij}} h = 0$ for all $i \in [d_1]$, $j \in [d]$ and $\theta \in \mathbf{S}_1 \cap \mathbf{S}_2$. Since $h$ is $\mathcal{C}^1$, and the set $\mathbf{S}_1 \cap \mathbf{S}_2$ is dense in $\Theta_{\mathrm{GELU}}$, we conclude that $\nabla_A h$ and $\nabla_B h$ are 0 on $\Theta_{\mathrm{GELU}}$, which means $h$ is a constant function. $\qquad\square$

## A.2. Conservation Laws for Feedforward Networks with SiLU Activation – Theorem 4.1 (SiLU part)

**Feedforward Networks with SiLU Activation.** Recall that, an FFN with SiLU is a function $f : \mathbb{R}^d \to \mathbb{R}$ of the form

$$f(x; A, B) = A \cdot \mathrm{SiLU}(Bx), \tag{32}$$

where the SiLU activation is defined by:

$$\mathrm{SiLU}(z) = z\sigma(z), \quad \text{where } \sigma(z) = \frac{1}{1 + e^{-z}} \text{ is the sigmoid function.} \tag{33}$$

**Parameter Space.** The function $f$ is parameterized by:

$$\theta = (A, B) \in \Theta_{\mathrm{SiLU}} := \mathbb{R}^{1 \times d_1} \times \mathbb{R}^{d_1 \times d}. \tag{34}$$

We now present the proof of the conservation laws for feedforward networks with SiLU, which constitutes the second part of Theorem 4.1.

*Proof.* Owing to the close similarity between SiLU and GeLU feedforward networks, the proof follows exactly the same steps. In particular, if $h : \Theta_{\mathrm{SiLU}} \to \mathbb{R}$ is a conservation law for $f$, then the same argument yields

$$\sum_{i=1}^{d_1} \mathrm{SiLU}(B_{ij}t) \cdot \partial_{A_i} h + \sum_{i=1}^{d_1} A_i t \mathrm{SiLU}'(B_{ij}t) \cdot \partial_{B_{ij}} h = 0, \tag{35}$$

for all $j \in [d]$ and $t \in \mathbb{R}$. Moreover, an analogue of Lemma A.1 holds for the SiLU activation.

**Lemma A.2.** *Let $\alpha_1, \ldots, \alpha_h \in \mathbb{R}$ such that $\alpha_1^2, \ldots, \alpha_h^2$ are pairwise distinct and nonzero. The $2h$ functions*

$$t \mapsto \text{SiLU}(\alpha_i t), \qquad t \mapsto t\text{SiLU}'(\alpha_i t), \qquad \text{for } i \in [h], \tag{36}$$

*are linearly independent.*

*Proof.* As in Lemma A.1, the proof of Lemma A.2 follows from analyzing the power series expansion of the SiLU activation. A key distinction, however, is that unlike GeLU, the SiLU function is not entire. However, it is holomorphic in the disk $\{z \in \mathbb{C} \colon |z| < \pi\}$, and within this disk we have

$$\text{SiLU}(z) = \frac{1}{2}z + \sum_{n=1}^{\infty} \frac{(2^{2n} - 1)B_{2n}}{(2n)!} z^{2n}. \tag{37}$$

Here, $B_{2n}$ denotes the $2n$-th Bernoulli numbers, which are well-known to be nonzero for all $n \geq 1$. $\qquad\square$

The remainder of the proof follows exactly as in the GeLU case; see Appendix A.1. $\qquad\square$

### A.3. Conservation Laws for Feedforward Networks with SwiGLU Activation – Theorem 4.2

**Feedforward Networks with SwiGLU Activation.** Recall that, an FFN with SwiGLU is a function $f \colon \mathbb{R}^d \to \mathbb{R}$ of the form

$$f(x; A, B, C) = A \cdot \big(\text{SiLU}(Bx) \odot (Cx)\big), \tag{38}$$

where $\odot$ denotes the Hadamard product.

**Parameter Space.** The function FFN $f$ is parameterized by:

$$\theta = (A, B, C) \in \Theta_{\text{SwiGLU}} := \mathbb{R}^{1 \times d_1} \times \mathbb{R}^{d_1 \times d} \times \mathbb{R}^{d_1 \times d}. \tag{39}$$

We now present the proof of the conservation laws for feedforward networks with SwiGLU, as stated in Theorem 4.2.

*Proof.* The SwiGLU FFN $f$ is $\mathcal{C}^1$ with respect to $A, B, C$. The gradients of $f$ with respect to $A, B, C$ are given by:

$$\nabla_A f = \big(\text{SiLU}(Bx) \odot (Cx)\big)^{\top}, \quad \nabla_B f = \big(A^{\top} \odot ((Cx) \odot \text{SiLU}'(Bx))\big)x^{\top}, \quad \nabla_C f = \big(A^{\top} \odot \text{SiLU}(Bx)\big)x^{\top}. \tag{40}$$

Let $h \colon \Theta_{\text{SwiGLU}} \to \mathbb{R}$ be a conservation law for $f$. We have, for all $(A, B, C) \in \Theta_{\text{SwiGLU}}$ and $x \in \mathbb{R}^d$,

$$0 = \langle \nabla f, \nabla h \rangle = \langle \nabla_A f, \nabla_A h \rangle + \langle \nabla_B f, \nabla_B h \rangle + \langle \nabla_C f, \nabla_C h \rangle. \tag{41}$$

Substituting the gradients of $f$, we obtain

$$0 = \Big\langle \big(\text{SiLU}(Bx) \odot (Cx)\big)^{\top}, \nabla_A h \Big\rangle$$
$$+ \Big\langle \big(A^{\top} \odot ((Cx) \odot \text{SiLU}'(Bx))\big)x^{\top}, \nabla_B h \Big\rangle + \Big\langle \big(A^{\top} \odot \text{SiLU}(Bx)\big)x^{\top}, \nabla_C h \Big\rangle. \tag{42}$$

For $i \in [d_1]$, let $b_i \in \mathbb{R}^d$ be the $i$-th row of $B$ and $c_i \in \mathbb{R}^d$ be the $i$-th row of $C$. Then Equation (42) is equivalent to

$$0 = \sum_{i=1}^{d_1} \Big(\text{SiLU}(b_i x)(c_i x)\Big) \cdot \partial_{A_i} h$$
$$+ \sum_{i=1}^{d_1} \sum_{j=1}^{d} \Big(A_i (c_i x)\text{SiLU}'(b_i x)x_j\Big) \cdot \partial_{B_{ij}} h + \sum_{i=1}^{d_1} \sum_{j=1}^{d} \Big(A_i \text{SiLU}(b_i x)x_j\Big) \cdot \partial_{C_{ij}} h. \tag{43}$$

For each $j \in [d]$, set $x = te_j$ where $t \in \mathbb{R}$ and $e_j \in \mathbb{R}^{d \times 1}$ is the $j$-th basis vector. Then $b_i x = B_{ij} t$ and $c_i x = C_{ij} t$, and Equation (43) becomes

$$
\begin{aligned}
0 &= \sum_{i=1}^{d_1} \Big( \text{SiLU}(B_{ij}t)(C_{ij}t) \Big) \cdot \partial_{A_i} h + \sum_{i=1}^{d_1} \Big( A_i (C_{ij}t) \text{SiLU}'(B_{ij}t)t \Big) \cdot \partial_{B_{ij}} h + \sum_{i=1}^{d_1} \Big( A_i \text{SiLU}(B_{ij}t)t \Big) \cdot \partial_{C_{ij}} h \\
&= t \Big( \sum_{i=1}^{d_1} (C_{ij} \cdot \partial_{A_i} h + A_i \cdot \partial_{C_{ij}} h) \cdot \text{SiLU}(B_{ij}t) + \sum_{i=1}^{d_1} A_i C_{ij} \cdot \partial_{B_{ij}} h \cdot t\text{SiLU}'(B_{ij}t) \Big).
\end{aligned}
\tag{44}
$$

By continuity on $t \in \mathbb{R}$, we obtain

$$
0 = \sum_{i=1}^{d_1} (C_{ij} \cdot \partial_{A_i} h + A_i \cdot \partial_{C_{ij}} h) \cdot \text{SiLU}(B_{ij}t) + \sum_{i=1}^{d_1} A_i C_{ij} \cdot \partial_{B_{ij}} h \cdot t\text{SiLU}'(B_{ij}t)
\tag{45}
$$

for all $t \in \mathbb{R}$. Define the following set

$$
\mathbf{S}_1 := \{ \theta = (A, B, C) : \ B_{ij}^2 \text{ is nonzero and pairwise distinct for all } i \in [d_1] \text{ and } j \in [d] \} \subset \Theta_{\text{SwiGLU}}.
\tag{46}
$$

From Equation (45), by applying Lemma A.2, we have

$$
C_{ij} \cdot \partial_{A_i} h + A_i \cdot \partial_{C_{ij}} h = A_i C_{ij} \cdot \partial_{B_{ij}} h = 0,
\tag{47}
$$

for all $i \in [d_1]$, $j \in [d]$, and $\theta \in \mathbf{S}_1$. Define the following set

$$
\mathbf{S}_2 := \{ \theta = (A, B) : \ A_i \text{ and } C_{ij} \text{ are nonzero for all } i \in [d_1] \text{ and } j \in [d] \} \subset \Theta_{\text{SwiGLU}}.
\tag{48}
$$

From Equation (47), we have

$$
C_{ij} \cdot \partial_{A_i} h + A_i \cdot \partial_{C_{ij}} h = \partial_{B_{ij}} h = 0,
\tag{49}
$$

for all $i \in [d_1]$, $j \in [d]$ and $\theta \in \mathbf{S}_1 \cap \mathbf{S}_2$. Since $h$ is $\mathcal{C}^1$, and the set $\mathbf{S}_1 \cap \mathbf{S}_2$ is dense in $\Theta_{\text{SwiGLU}}$, we conclude that Equation (49) holds for all $\theta \in \Theta_{\text{SwiGLU}}$.

Extending the argument to the vector-valued setting of $f$, Equation (49) yields the same constraints on $h$ as those stated in Theorem 4.2: For all $i \in [d_1]$,

$$
\nabla_B h = 0, \quad \text{and} \quad \nabla_{A_{:,i}} h \cdot C_{i,:} + A_{:,i} \cdot \nabla_{C_{i,:}} h = 0.
$$

Solving this system of PDEs leads to the invariants $\|A_{:,i}\|^2 - \|C_{i,:}\|^2$, $i \in [d_1]$. We conclude the proof. $\qquad \square$

## B. Theoretical Proofs of Conservation Laws for Multihead Attention

### B.1. Conservation Laws for Multihead Attention without Positional Encoding – Theorem 4.3

Recall the space of all sequences of $d$-dimensional tokens as $\mathcal{S} := \sqcup_{L=1}^{\infty} \mathbb{R}^{L \times d}$. Given a fixed head dimension $d_h$ (usually, $d_h = d/n$), consider $Q_i, K_i, V_i, O_i \in \mathbb{R}^{d \times d_h}$ for each $i \in [n]$. A Multihead Attention is a map $\text{MHA} : \mathcal{S} \to \mathcal{S}$ of the form

$$
\text{MHA}\big( \mathbf{x}; \{Q_i, K_i, V_i, O_i\}_{i=1}^n \big) = \sum_{i=1}^{n} \text{softmax} \Big( (\mathbf{x}Q_i)(\mathbf{x}K_i)^\top \Big) \cdot (\mathbf{x}V_i) O_i^\top.
\tag{50}
$$

By construction, we have $\text{MHA}(\mathbb{R}^{L \times d}) \subset \mathbb{R}^{L \times d}$.

**Parameter Space.** The map MHA is parameterized by

$$
\theta = (Q_i, K_i, V_i, O_i)_{i=1}^n \in \Theta_{\text{MHA}} := \big( \mathbb{R}^{d \times d_h} \big)^{4n}.
\tag{51}
$$

We now present the proof of the conservation laws for Multihead Attention without Positional Encoding.

*Proof.* For $i \in [h]$ and $k, p \in [L]$, define the similarity scores and the attention scores of $k$-th token to the $p$-th token at $i$-th head as follows:

$$s_{kp}^{(i)}(\mathbf{x}; \theta) := x_k Q_i K_i^\top x_p^\top, \quad \text{and} \quad a_{kp}^{(i)}(\mathbf{x}; \theta) := \text{softmax}_p \left[ \left( s_{kq}^{(i)}(\mathbf{x}; \theta) \right)_{q=1}^L \right] = \frac{e^{s_{kp}(\mathbf{x}; \theta)}}{\sum_{q=1}^L e^{s_{kq}(\mathbf{x}; \theta)}}. \tag{52}$$

For $k \in [L]$ and $\kappa \in [d]$, the $k$-th output token and its $\kappa$-th feature are:

$$\text{MHA}_k(\mathbf{x}; \theta) = \sum_{i=1}^n \left( \sum_{p=1}^L a_{kp}^{(i)} \cdot x_p V_i O_i^\top \right), \quad \text{and} \quad \text{MHA}_{k,\kappa}(\mathbf{x}; \theta) = \sum_{i=1}^n \left( \sum_{p=1}^L a_{kp}^{(i)} \cdot x_p (V_i O_i^\top)_{:,\kappa} \right). \tag{53}$$

**Step 1.** The function $\text{MHA}_{k,\kappa}$ is $\mathcal{C}^1$ with respect to $\theta$. We now derive its gradients.

*(i) The gradients of $\text{MHA}_{k,\kappa}$ with respect to $Q$ and $K$.*

The row-softmax derivative gives

$$\frac{\partial a_{kp}^{(i)}}{\partial s_{kq}^{(i)}} = \frac{\delta_{pq} e^{s_{kp}^{(i)}} \left( \sum_{q=1}^L e^{s_{kq}^{(i)}} \right) - e^{s_{kp}^{(i)}} e^{s_{kq}^{(i)}}}{\left( \sum_{q=1}^L e^{s_{kq}^{(i)}} \right)^2} = \delta_{pq} \frac{e^{s_{kp}^{(i)}}}{\sum_{q=1}^L e^{s_{kq}^{(i)}}} - \frac{e^{s_{kp}^{(i)}} e^{s_{kq}^{(i)}}}{\left( \sum_{q=1}^L e^{s_{kq}^{(i)}} \right)^2} = a_{kp}^{(i)}(\delta_{pq} - a_{kq}^{(i)}). \tag{54}$$

We have

$$\nabla_{Q_i} s_{kp}^{(i)} = x_k^\top x_p K_i, \quad \text{and} \quad \nabla_{K_i} s_{kp}^{(i)} = x_p^\top x_k Q_i. \tag{55}$$

Therefore,

$$\nabla_{Q_i} a_{kp}^{(i)} = \sum_{q=1}^L \frac{\partial a_{kp}^{(i)}}{\partial s_{kq}^{(i)}} \cdot \nabla_{Q_i} s_{kq}^{(i)} = \sum_{q=1}^L a_{kp}^{(i)}(\delta_{pq} - a_{kq}^{(i)}) x_k^\top x_q K_i = a_{kp}^{(i)} x_k^\top \left( x_p - \sum_{q=1}^L a_{kq}^{(i)} x_q \right) K_i, \tag{56}$$

$$\nabla_{K_i} a_{kp}^{(i)} = \sum_{q=1}^L \frac{\partial a_{kp}^{(i)}}{\partial s_{kq}^{(i)}} \cdot \nabla_{K_i} s_{kq}^{(i)} = \sum_{q=1}^L a_{kp}^{(i)}(\delta_{pq} - a_{kq}^{(i)}) x_q^\top x_k Q_i = a_{kp}^{(i)} \left( x_p^\top - \sum_{q=1}^L a_{kq}^{(i)} x_q^\top \right) x_k Q_i. \tag{57}$$

The gradients of $\text{MHA}_{k,\kappa}$ with respect to $Q_i, K_i$ are given by

$$\nabla_{Q_i} \text{MHA}_{k,\kappa}(\mathbf{x}; \theta) = \sum_{p=1}^L \nabla_{Q_i} a_{kp}^{(i)} \cdot x_p (V_i O_i^\top)_{:,\kappa} = \sum_{p=1}^L a_{kp}^{(i)} x_k^\top \left( x_p - \sum_{q=1}^L a_{kq}^{(i)} x_q \right) K_i \cdot x_p (V_i O_i^\top)_{:,\kappa}, \tag{58}$$

$$\nabla_{K_i} \text{MHA}_{k,\kappa}(\mathbf{x}; \theta) = \sum_{p=1}^L \nabla_{K_i} a_{kp}^{(i)} \cdot x_p (V_i O_i^\top)_{:,\kappa} = \sum_{p=1}^L a_{kp}^{(i)} \left( x_p^\top - \sum_{q=1}^L a_{kq}^{(i)} x_q^\top \right) x_k Q_i \cdot x_p (V_i O_i^\top)_{:,\kappa}. \tag{59}$$

*(ii) The gradients of $\text{MHA}_{k,\kappa}$ with respect to $V$ and $O$.*

For $\kappa \in [d]$, let $e_\kappa \in \mathbb{R}^d$ denote the $\kappa$-th column basis vector. Since

$$\nabla_{V_i} \left( x (V_i O_i^\top)_{:,\kappa} \right) = (e_\kappa x)^\top O_i, \quad \text{and} \quad \nabla_{O_i} \left( x (V_i O_i^\top)_{:,\kappa} \right) = (e_\kappa x) V_i, \tag{60}$$

the gradients of $\text{MHA}_{k,\kappa}$ with respect to $V_i, O_i$ are given by

$$\nabla_{V_i} \text{MHA}_{k,\kappa}(\mathbf{x}; \theta) = \sum_{p=1}^L a_{kp}^{(i)} \cdot \nabla_{V_i} \left( x_p (V_i O_i^\top)_{:,\kappa} \right) = \sum_{p=1}^L a_{kp}^{(i)} \cdot (e_\kappa x_p)^\top O_i, \tag{61}$$

$$\nabla_{O_i} \text{MHA}_{k,\kappa}(\mathbf{x}; \theta) = \sum_{p=1}^L a_{kp}^{(i)} \cdot \nabla_{O_i} \left( x_p (V_i O_i^\top)_{:,\kappa} \right) = \sum_{p=1}^L a_{kp}^{(i)} \cdot (e_\kappa x_p) V_i. \tag{62}$$

**Step 2.** Fix $k = 1$ and $\kappa \in [d]$. Consider an input $\mathbf{x} \in \mathbb{R}^{L \times d}$ of the form $\mathbf{x} = (x, y, y, \ldots, y)^\top$, where the first row is $x$ and the remaining $L - 1$ rows are all equal to $y$. We have

$$
\begin{aligned}
\mathrm{MHA}_{1,\kappa}(\mathbf{x}; \theta) &= \sum_{p=1}^{L} a_{1p}^{(i)} \cdot x_p (V_i O_i^\top)_{:,\kappa} = a_{11}^{(i)} \cdot x (V_i O_i^\top)_{:,\kappa} + \left( \sum_{p=2}^{L} a_{1p}^{(i)} \right) \cdot y (V_i O_i^\top)_{:,\kappa} \\
&= a_{11}^{(i)} \cdot x (V_i O_i^\top)_{:,\kappa} + \left( 1 - a_{11}^{(i)} \right) \cdot y (V_i O_i^\top)_{:,\kappa} = y (V_i O_i^\top)_{:,\kappa} + a_{11}^{(i)} \cdot (x - y)(V_i O_i^\top)_{:,\kappa}.
\end{aligned} \tag{63}
$$

From Equation (56), we have

$$
\begin{aligned}
\nabla_{Q_i} a_{11}^{(i)} &= a_{11}^{(i)} x^\top \left( x - \sum_{q=1}^{L} a_{1q}^{(i)} x_q \right) K_i \\
&= a_{11}^{(i)} x^\top \left( x - a_{11}^{(i)} x - \left( 1 - a_{11}^{(i)} \right) y \right) K_i = a_{11}^{(i)} \left( 1 - a_{11}^{(i)} \right) x^\top (x - y) K_i.
\end{aligned} \tag{64}
$$

Similarly, we have

$$
\nabla_{K_i} a_{11}^{(i)} = a_{11}^{(i)} \left( 1 - a_{11}^{(i)} \right) (x - y)^\top x Q_i. \tag{65}
$$

Thus, the gradient of $\mathrm{MHA}_{1,\kappa}$ are given by

$$
\nabla_{Q_i} \mathrm{MHA}_{1,\kappa}(\mathbf{x}; \theta) = a_{11}^{(i)} \left( 1 - a_{11}^{(i)} \right) (x - y)(V_i O_i^\top)_{:,\kappa} \cdot x^\top (x - y) K_i, \tag{66}
$$

$$
\nabla_{K_i} \mathrm{MHA}_{1,\kappa}(\mathbf{x}; \theta) = a_{11}^{(i)} \left( 1 - a_{11}^{(i)} \right) (x - y)(V_i O_i^\top)_{:,\kappa} \cdot (x - y)^\top x Q_i, \tag{67}
$$

$$
\nabla_{V_i} \mathrm{MHA}_{1,\kappa}(\mathbf{x}; \theta) = \left( y + a_{11}^{(i)} \cdot (x - y) \right)^\top e_\kappa^\top O_i, \tag{68}
$$

$$
\nabla_{O_i} \mathrm{MHA}_{1,\kappa}(\mathbf{x}; \theta) = e_\kappa \left( y + a_{11}^{(i)} \cdot (x - y) \right) V_i. \tag{69}
$$

Let $h \colon \Theta_{\mathrm{MHA}} \to \mathbb{R}$ be a conservation law for MHA. We have, for all $\theta \in \Theta_{\mathrm{MHA}}$, $L \in \mathbb{N}$, and $x, y \in \mathbb{R}^{1 \times d}$,

$$
\begin{aligned}
0 &= \sum_{i=1}^{n} \left( \langle \nabla_{Q_i} \mathrm{MHA}_{1,\kappa}, \nabla_{Q_i} h \rangle + \langle \nabla_{K_i} \mathrm{MHA}_{1,\kappa}, \nabla_{K_i} h \rangle + \langle \nabla_{V_i} \mathrm{MHA}_{1,\kappa}, \nabla_{Q_i} h \rangle + \langle \nabla_{O_i} \mathrm{MHA}_{1,\kappa}, \nabla_{O_i} h \rangle \right) \\
&= \sum_{i=1}^{n} \left( \left\langle a_{11}^{(i)} \left( 1 - a_{11}^{(i)} \right) (x - y)(V_i O_i^\top)_{:,\kappa} \cdot x^\top (x - y) K_i, \nabla_{Q_i} h \right\rangle \right. \\
&\qquad\qquad + \left\langle a_{11}^{(i)} \left( 1 - a_{11}^{(i)} \right) (x - y)(V_i O_i^\top)_{:,\kappa} \cdot (x - y)^\top x Q_i, \nabla_{K_i} h \right\rangle \\
&\qquad\qquad\qquad \left. + \left\langle \left( y + a_{11}^{(i)} \cdot (x - y) \right)^\top e_\kappa^\top O_i, \nabla_{V_i} h \right\rangle + \left\langle e_\kappa \left( y + a_{11}^{(i)} \cdot (x - y) \right) V_i, \nabla_{O_i} h \right\rangle \right) \\
&= \sum_{i=1}^{h} a_{11}^{(i)} \left( 1 - a_{11}^{(i)} \right) \cdot \left\langle (x - y)(V_i O_i^\top)_{:,\kappa} \cdot (x - y)^\top x, K_i (\nabla_{Q_i} h)^\top + (\nabla_{K_i} h) Q_i^\top \right\rangle \\
&\qquad + \sum_{i=1}^{n} a_{11}^{(i)} \cdot \left\langle e_\kappa (x - y), O_i (\nabla_{V_i} h)^\top + (\nabla_{O_i} h) V_i^\top \right\rangle + \sum_{i=1}^{h} \left\langle e_\kappa y, O_i (\nabla_{V_i} h)^\top + (\nabla_{O_i} h) V_i^\top \right\rangle.
\end{aligned} \tag{70}
$$

**Step 3.** For $i \in [n]$, define

$$
\mathfrak{a}_i := \left\langle (x - y)(V_i O_i^\top)_{:,\kappa} \cdot (x - y)^\top x, K_i (\nabla_{Q_i} h)^\top + (\nabla_{K_i} h) Q_i^\top \right\rangle, \tag{71}
$$

$$
\mathfrak{b}_i := \left\langle e_\kappa (x - y), O_i (\nabla_{V_i} h)^\top + (\nabla_{O_i} h) V_i^\top \right\rangle, \tag{72}
$$

$$
\mathfrak{c} := \sum_{i=1}^{n} \left\langle e_\kappa y, O_i (\nabla_{V_i} h)^\top + (\nabla_{O_i} h) V_i^\top \right\rangle. \tag{73}
$$

From Equation (70), we have

$$0 = \sum_{i=1}^{n} a_{11}^{(i)} \left(1 - a_{11}^{(i)}\right) \cdot \mathfrak{a}_i + \sum_{i=1}^{n} a_{11}^{(i)} \cdot \mathfrak{b}_i + \mathfrak{c}. \tag{74}$$

Here, $\mathfrak{a}_i, \mathfrak{b}_i, \mathfrak{c}$ are functions of $x, y, \theta$, while $a_{11}^{(i)}$ are functions of $x, y, \theta$ and $L$; during our argument, their input arguments are omitted for simplicity. We set $L$ to be $L + 1$, and express $a_{11}^{(i)}$ as follows:

$$a_{11}^{(i)} = a_{11}^{(i)} ((x, y, \ldots, y); \theta) = \frac{e^{xQ_i K_i^\top x^\top}}{e^{xQ_i K_i^\top x^\top} + L \cdot e^{xQ_i K_i^\top y^\top}} = \frac{1}{1 + L \cdot e^{xQ_i K_i^\top (y-x)^\top}}. \tag{75}$$

From Equation (74), we have

$$0 = \sum_{i=1}^{n} \frac{L \cdot e^{xQ_i K_i^\top (y-x)^\top}}{\left(1 + L \cdot e^{xQ_i K_i^\top (y-x)^\top}\right)^2} \cdot \mathfrak{a}_i + \sum_{i=1}^{n} \frac{1}{1 + L \cdot e^{xQ_i K_i^\top (y-x)^\top}} \cdot \mathfrak{b}_i + \mathfrak{c}. \tag{76}$$

For fixed $x, y, \theta$, the RHS of Equation (76) defines a rational function in the variable $L$. As this function is zero for all positive integers $L$, it follows that it is identically zero. Therefore, for all $t \in \mathbb{R}$, we have

$$0 = \sum_{i=1}^{n} \frac{t \cdot e^{xQ_i K_i^\top (y-x)^\top}}{\left(1 + t \cdot e^{xQ_i K_i^\top (y-x)^\top}\right)^2} \cdot \mathfrak{a}_i + \sum_{i=1}^{n} \frac{1}{1 + t \cdot e^{xQ_i K_i^\top (y-x)^\top}} \cdot \mathfrak{b}_i + \mathfrak{c}. \tag{77}$$

This motivates the following lemma.

**Lemma B.1.** *Let $\alpha_1, \ldots, \alpha_n$ be $n$ pairwise distinct nonzero numbers. The $2n + 1$ functions*

$$t \mapsto \frac{t\alpha_i}{(1 + t\alpha_i)^2}, \qquad t \mapsto \frac{1}{1 + t\alpha_i}, \qquad t \mapsto 1, \qquad for\ i \in [n] \tag{78}$$

*are linear independent.*

Lemma B.1 can be proved by analyzing the multiplicities of the poles at $t = -1/\alpha_i$ for each $i$. It suggests that $\mathfrak{a}_i = \mathfrak{b}_i = \mathfrak{c} = 0$ whenever $xQ_i K_i^\top (y - x)^\top, i \in [n]$ are pairwise distinct. Define the following set

$$\mathbf{S}_1 := \{\theta :\ Q_i K_i^\top \text{ are pairwise distinct for all } i \in [n]\}. \tag{79}$$

For each $\theta \in \mathbf{S}_1$, define the following set

$$\mathbf{S}_\theta := \{(x, y) \in \mathbb{R}^d \times \mathbb{R}^d :\ xQ_i K_i^\top (y - x)^\top \text{ are pairwise distinct for all } i \in [n]\}. \tag{80}$$

From Equation (77), for $\theta \in \mathbf{S}_1$ and $(x, y) \in \mathbf{S}_\theta$, by applying Lemma B.1, we have $\mathfrak{a}_i = \mathfrak{b}_i = \mathfrak{c} = 0$. Moreover, $\mathbf{S}_\theta$ is dense in $\mathbb{R}^d \times \mathbb{R}^d$, since $\mathbf{S}_\theta$ is the complement of the zero set of the following polynomial

$$(x, y) \mapsto \prod_{1 \leq i < j \leq n} \left(xQ_i K_i^\top (y - x)^\top - xQ_j K_j^\top (y - x)^\top\right), \tag{81}$$

which is not identical to 0 since $\theta \in \mathbf{S}_1$ (In other words, $\mathbf{S}_\theta$ is the complement of a proper real algebraic variety in $\mathbb{R}^d \times \mathbb{R}^d$, which is always dense). Moreover, $\mathbf{S}_1$ is also dense in $\Theta_{\text{MHA}}$. We conclude that $\mathfrak{a}_i = \mathfrak{b}_i = \mathfrak{c} = 0$ for all $x, y$ and $\theta$.

**Step 4.** Setting $z = y - x$ into the identity $\mathfrak{a}_i = \mathfrak{b}_i = 0$, we have

$$0 = \left\langle z(V_i O_i^\top)_{:,\kappa} \cdot z^\top x, K_i (\nabla_{Q_i} h)^\top + (\nabla_{K_i} h) Q_i^\top \right\rangle, \tag{82}$$

$$0 = \left\langle e_\kappa z, O_i (\nabla_{V_i} h)^\top + (\nabla_{O_i} h) V_i^\top \right\rangle. \tag{83}$$

Since the linear span of $\{e_\kappa z : \kappa \in [d], z \in \mathbb{R}^d\}$ is the whole space $\mathbb{R}^{d \times d}$, it implies that $0 = O_i(\nabla_{V_i} h)^\top + (\nabla_{O_i} h)V_i^\top$ for all $i \in [h]$. For the constraints on $Q_i, K_i$, define the following set

$$\mathbf{S}_2 := \{\theta : \text{ all entries of } V_i O_i^\top \text{ are nonzero for all } i \in [n]\}. \tag{84}$$

From Equation (82), we have: For $z \in \mathbb{R}^d$ such that $z(V_i O_i^\top)_{:,\kappa} \neq 0$, then

$$0 = \left\langle z^\top x, K_i(\nabla_{Q_i} h)^\top + (\nabla_{K_i} h)Q_i^\top \right\rangle. \tag{85}$$

However, for $\theta \in \mathbf{S}_2$, the set $\{z \in \mathbb{R}^d : z(V_i O_i^\top)_{:,\kappa} \neq 0\}$ is dense in $\mathbb{R}^d$ (since $(V_i O_i^\top)_{:,\kappa}$ is nonzero), by continuity, Equation (85) holds for all $z \in \mathbb{R}^d$. Moreover, the linear span of $\{z^\top x : x, z \in \mathbb{R}^d\}$ is the whole space $\mathbb{R}^{d \times d}$, it implies that $0 = K_i(\nabla_{Q_i} h)^\top + (\nabla_{K_i} h)Q_i^\top$. This holds for all $\theta \in \mathbf{S}_2$, and $\mathbf{S}_2$ is dense in $\Theta_{\text{MHA}}$. We conclude that $0 = K_i(\nabla_{Q_i} h)^\top + (\nabla_{K_i} h)Q_i^\top$ for all $i \in [n]$.

In summary, we derive the same set of constraints on $h$ as stated in Theorem 4.3: For all $i \in [n]$,

$$0 = K_i \cdot (\nabla_{Q_i} h)^\top + (\nabla_{K_i} h) \cdot Q_i^\top = O_i \cdot (\nabla_{V_i} h)^\top + (\nabla_{O_i} h) \cdot V_i^\top.$$

Solving these constraints leads to the invariants $Q_i^\top Q_i - K_i^\top K_i$ and $V_i^\top V_i - O_i^\top O_i$ for $i \in [n]$. □

## B.2. Conservation Laws for Multihead Attention with Rotary Positional Encoding – Theorem 4.4

Recall the $d_h \times d_h$ block-diagonal rotation matrix for a token at position $n$,

$$R_n = \begin{bmatrix} \cos(n\varphi_1) & -\sin(n\varphi_1) & 0 & \cdots & 0 & 0 \\ \sin(n\varphi_1) & \cos(n\varphi_1) & 0 & \cdots & 0 & 0 \\ 0 & 0 & \cos(n\varphi_2) & \cdots & 0 & 0 \\ \vdots & \vdots & \vdots & \ddots & \vdots & \vdots \\ 0 & 0 & 0 & \cdots & \cos(n\varphi_{d_h/2}) & -\sin(n\varphi_{d_h/2}) \\ 0 & 0 & 0 & \cdots & \sin(n\varphi_{d_h/2}) & \cos(n\varphi_{d_h/2}) \end{bmatrix}, \tag{86}$$

where $\varphi_i = 10000^{-2(i-1)/d_h}$ for $i \in [d_h/2]$. A Multihead Attention with Rotary Positional Encoding (RoPE) is a map $\text{MHA}^{\text{RoPE}} : \mathcal{S} \to \mathcal{S}$ of the form

$$\text{MHA}^{\text{RoPE}}(\mathbf{x}; \theta) = \sum_{i=1}^n \text{softmax}\left[ x_p(Q_i R_{p-q} K_i^\top)x_q^\top \right]_{p,q \in [L]} \cdot \mathbf{x}(V_i O_i^\top). \tag{87}$$

We now present the proof of the conservation laws for Multihead Attention with Rotary Psotional Encoding, which is Theorem 4.4.

*Proof.* For $i \in [h]$ and $k, p \in [L]$, define the similarity scores and the attention scores of $k$-th token to the $p$-th token at $i$-th head with positional encoding as follows:

$$s_{kp}^{(i)}(\mathbf{x}; \theta) := x_k Q_i R_{k-p} K_i^\top x_p^\top, \quad \text{and} \quad a_{kp}^{(i)}(\mathbf{x}; \theta) := \text{softmax}_p\left[ \left(s_{kq}^{(i)}(\mathbf{x}; \theta)\right)_{q=1}^L \right] = \frac{e^{s_{kp}(\mathbf{x};\theta)}}{\sum_{q=1}^L e^{s_{kq}(\mathbf{x};\theta)}}. \tag{88}$$

For $k \in [L]$ and $\kappa \in [d]$, the $k$-th output token and its $\kappa$-th feature are:

$$\text{MHA}_k^{\text{RoPE}}(\mathbf{x}; \theta) = \sum_{i=1}^n \left( \sum_{p=1}^L a_{kp}^{(i)} \cdot x_p V_i O_i^\top \right), \quad \text{and} \quad \text{MHA}_{k,\kappa}^{\text{RoPE}}(\mathbf{x}; \theta) = \sum_{i=1}^n \left( \sum_{p=1}^L a_{kp}^{(i)} \cdot x_p(V_i O_i^\top)_{:,\kappa} \right). \tag{89}$$

Fix $k = 1$ and $\kappa \in [d]$. We now consider the function $\text{MHA}_{1,\kappa}^{\text{RoPE}}(\mathbf{x}; \theta)$, is $\mathcal{C}^1$ with respect to $\theta$.

**Step 1.** Consider an input $\mathbf{x} \in \mathbb{R}^{L \times d}$ of the form $\mathbf{x} = (x, 0, \ldots, 0)^\top$, where the first row is $x$ and the remaining $L - 1$ rows are all equal to $0$. We have

$$\text{MHA}_{1,\kappa}^{\text{RoPE}}\left((x, 0, \ldots, 0); \theta\right) = \sum_{p=1}^L a_{1p}^{(i)} \cdot x_p(V_i O_i^\top)_{:,\kappa} = a_{11}^{(i)} \cdot x(V_i O_i^\top)_{:,\kappa}. \tag{90}$$

We now derive the gradients of $\mathrm{MHA}_{1,\kappa}^{\mathrm{RoPE}}$ at $\mathbf{x} = (x, 0, \ldots, 0)^\top$.

*(i) The gradients of $\mathrm{MHA}_{1,\kappa}^{\mathrm{RoPE}}$ with respect to $Q$ and $K$ at $\mathbf{x} = (x, 0, \ldots, 0)^\top$.*

We have

$$\nabla_{Q_i} a_{11}^{(i)} = \sum_{q=1}^L a_{11}^{(i)} (\delta_{1q} - a_{1q}^{(i)}) x_1^\top x_q K_i R_{q-1} = a_{11}^{(i)} \left(1 - a_{11}^{(i)}\right) x^\top x K_i, \tag{91}$$

$$\nabla_{K_i} a_{11}^{(i)} = \sum_{q=1}^L a_{11}^{(i)} (\delta_{1q} - a_{1q}^{(i)}) x_q^\top x_1 Q_i R_{1-q} = a_{11}^{(i)} \left(1 - a_{11}^{(i)}\right) x^\top x Q_i. \tag{92}$$

Therefore,

$$\nabla_{Q_i} \mathrm{MHA}_{1,\kappa}^{\mathrm{RoPE}}(\mathbf{x}; \theta) = a_{11}^{(i)} \left(1 - a_{11}^{(i)}\right) x^\top x K_i \cdot x(V_i O_i^\top)_{:,\kappa},$$

$$\nabla_{K_i} \mathrm{MHA}_{1,\kappa}^{\mathrm{RoPE}}(\mathbf{x}; \theta) = a_{11}^{(i)} \left(1 - a_{11}^{(i)}\right) x^\top x Q_i \cdot x(V_i O_i^\top)_{:,\kappa}. \tag{93}$$

*(ii) The gradients of $\mathrm{MHA}_{1,\kappa}^{\mathrm{RoPE}}$ with respect to $V$ and $O$ at $\mathbf{x} = (x, 0, \ldots, 0)^\top$.*

Similar to the previous section, we have

$$\nabla_{V_i} \mathrm{MHA}_{1,\kappa}^{\mathrm{RoPE}}(\mathbf{x}; \theta) = a_{11}^{(i)} \cdot x^\top e_\kappa^\top O_i, \quad \text{and} \quad \nabla_{O_i} \mathrm{MHA}_{1,\kappa}^{\mathrm{RoPE}}(\mathbf{x}; \theta) = a_{11}^{(i)} \cdot x e_\kappa V_i. \tag{94}$$

Let $h \colon \Theta_{\mathrm{MHA}} \to \mathbb{R}$ be a conservation law for $\mathrm{MHA}^{\mathrm{RoPE}}$. We have, for all $\theta \in \Theta_{\mathrm{MHA}}$, $L \in \mathbb{N}$, and $x, y \in \mathbb{R}^{1 \times d}$,

$$\begin{aligned}
0 &= \sum_{i=1}^n \left( \left\langle \nabla_{Q_i} \mathrm{MHA}_{1,\kappa}^{\mathrm{RoPE}}, \nabla_{Q_i} h \right\rangle + \left\langle \nabla_{K_i} \mathrm{MHA}_{1,\kappa}^{\mathrm{RoPE}}, \nabla_{K_i} h \right\rangle + \left\langle \nabla_{V_i} \mathrm{MHA}_{1,\kappa}^{\mathrm{RoPE}}, \nabla_{V_i} h \right\rangle + \left\langle \nabla_{O_i} \mathrm{MHA}_{1,\kappa}^{\mathrm{RoPE}}, \nabla_{O_i} h \right\rangle \right) \\
&= \sum_{i=1}^n \left\langle a_{11}^{(i)} \left(1 - a_{11}^{(i)}\right) x^\top x K_i \cdot x(V_i O_i^\top)_{:,\kappa}, \nabla_{Q_i} h \right\rangle + \sum_{i=1}^n \left\langle a_{11}^{(i)} \left(1 - a_{11}^{(i)}\right) x^\top x Q_i \cdot x(V_i O_i^\top)_{:,\kappa}, \nabla_{K_i} h \right\rangle \\
&\qquad + \sum_{i=1}^n \left\langle a_{11}^{(i)} \cdot x^\top e_\kappa^\top O_i, \nabla_{V_i} h \right\rangle + \sum_{i=1}^n \left\langle a_{11}^{(i)} \cdot x e_\kappa V_i, \nabla_{O_i} h \right\rangle \\
&= \sum_{i=1}^n a_{11}^{(i)} \left(1 - a_{11}^{(i)}\right) \cdot x(V_i O_i^\top)_{:,\kappa} \cdot \left[ \left\langle x^\top x K_i, \nabla_{Q_i} h \right\rangle + \left\langle x^\top x Q_i, \nabla_{K_i} h \right\rangle \right] \\
&\qquad\qquad\qquad\qquad + \sum_{i=1}^n a_{11}^{(i)} \cdot \left[ \left\langle x^\top e_\kappa^\top O_i, \nabla_{V_i} h \right\rangle + \left\langle x e_\kappa V_i, \nabla_{O_i} h \right\rangle \right].
\end{aligned} \tag{95}$$

For $i \in [n]$, define

$$\mathfrak{a}_i := x(V_i O_i^\top)_{:,\kappa} \cdot \left[ \left\langle x^\top x K_i, \nabla_{Q_i} h \right\rangle + \left\langle x^\top x Q_i, \nabla_{K_i} h \right\rangle \right], \tag{96}$$

$$\mathfrak{b}_i := \left[ \left\langle x^\top e_\kappa^\top O_i, \nabla_{V_i} h \right\rangle + \left\langle x e_\kappa V_i, \nabla_{O_i} h \right\rangle \right]. \tag{97}$$

From Equation (95), we have

$$0 = \sum_{i=1}^n a_{11}^{(i)} \cdot \left(1 - a_{11}^{(i)}\right) \cdot \mathfrak{a}_i + \sum_{i=1}^n a_{11}^{(i)} \cdot \mathfrak{b}_i. \tag{98}$$

Note that, $a_{11}^{(i)}$ is a function of $L, x$ and $\theta$, while $\mathfrak{a}_i, \mathfrak{b}_i$ are functions of $x$ and $\theta$; during our argument, their input arguments are omitted for simplicity. We set $L$ to be $L + 1$, and express $a_{11}^{(i)}$ as follows:

$$a_{11}^{(i)} = a_{11}^{(i)} ((x, 0, \ldots, 0); \theta) = \frac{e^{x Q_i K_i^\top x^\top}}{e^{x Q_i K_i^\top x^\top} + L} = \frac{1}{1 + L \cdot e^{-x Q_i K_i^\top x^\top}}. \tag{99}$$

From Equation (98), we have

$$0 = \sum_{i=1}^{n} \frac{L \cdot e^{-xQ_iK_i^\top x^\top}}{\left(1 + L \cdot e^{-xQ_iK_i^\top x^\top}\right)^2} \cdot \mathfrak{a}_i + \sum_{i=1}^{n} \frac{1}{1 + L \cdot e^{-xQ_iK_i^\top x^\top}} \cdot \mathfrak{b}_i. \tag{100}$$

By the same argument as in the proof of the classical MHA (which is related to Lemma B.1), we obtain that $\mathfrak{a}_i = \mathfrak{b}_i = 0$ whenever the $n$ scalars $xQ_iK_i^\top x^\top$, $i \in [n]$, are pairwise distinct. Define the following set

$$\mathbf{S}_1 := \{\theta : \ Q_iK_i^\top + K_iQ_i^\top \text{ are pairwise distinct for all } i \in [n]\}. \tag{101}$$

Observe that each symmetric matrix $A \in \mathbb{R}^{n \times n}$ uniquely determines the polynomial $xAx^\top$. Moreover, by the same follow-up density argument, the identity $\mathfrak{a}_i = \mathfrak{b}_i = 0$ extends to all $x$ and $\theta$. We then have, for all $i \in [n]$, $\mathfrak{b}_i = 0$ implies $0 = O_i(\nabla_{V_i}h)^\top + (\nabla_{O_i}h)V_i^\top$, and $\mathfrak{a}_i = 0$ implies

$$0 = \left\langle x^\top xK_i, \nabla_{Q_i}h \right\rangle + \left\langle x^\top xQ_i, \nabla_{K_i}h \right\rangle. \tag{102}$$

However, while the constraints on $h$ associated with $V$ and $O$ are obtained as expected, the constraints on $h$ associated with $Q$ and $K$ in Equation (102) are not sufficient to guarantee that $h$ is a conservation law of $\mathrm{MHA}_{1,\kappa}^{\mathrm{RoPE}}$. Before proceeding to the next step, we make an observation: since $0 = O_i(\nabla_{V_i}h)^\top + (\nabla_{O_i}h)V_i^\top$, the following identity always holds:

$$0 = \sum_{i=1}^{n} \left( \left\langle \nabla_{V_i}\mathrm{MHA}_{1,\kappa}^{\mathrm{RoPE}}, \nabla_{V_i}h \right\rangle + \left\langle \nabla_{O_i}\mathrm{MHA}_{1,\kappa}^{\mathrm{RoPE}}, \nabla_{O_i}h \right\rangle \right). \tag{103}$$

**Step 3.** Let $N$ be a positive integer. Consider an input $\mathbf{x} \in \mathbb{R}^{L \times d}$ of the form $\mathbf{x} = (x, 0, \ldots, 0, y, 0, \ldots, 0)^\top$, where there are $L - 2$ token zeros between $x$ and $y$, and an additional $N$ trailing zeros, We have

$$\mathrm{MHA}_{1,\kappa}\big((x, 0, \ldots, 0, y); \theta\big) = \sum_{p=1}^{L} a_{1p}^{(i)} \cdot x_p(V_iO_i^\top)_{:,\kappa} = a_{11}^{(i)} \cdot x(V_iO_i^\top)_{:,\kappa} + a_{1L}^{(i)} \cdot y(V_iO_i^\top)_{:,\kappa}. \tag{104}$$

We now derive the gradients of $\mathrm{MHA}_{1,\kappa}^{\mathrm{RoPE}}$ with respect to $Q$ and $K$ at $\mathbf{x} = (x, 0, \ldots, 0, y)^\top$.

$$\nabla_{Q_i}a_{11}^{(i)} = \sum_{q=1}^{L} a_{11}^{(i)}(\delta_{1q} - a_{1q}^{(i)})x_1^\top x_q K_i R_{q-1} = a_{11}^{(i)}\left(1 - a_{11}^{(i)}\right)x^\top xK_i - a_{11}^{(i)}a_{1L}^{(i)}x^\top yK_iR_{L-1}, \tag{105}$$

$$\nabla_{Q_i}a_{1L}^{(i)} = \sum_{q=1}^{L} a_{1L}^{(i)}(\delta_{Lq} - a_{1q}^{(i)})x_1^\top x_q K_i R_{q-1} = -a_{1L}^{(i)}a_{11}^{(i)}x^\top xK_i + a_{1L}^{(i)}\left(1 - a_{1L}^{(i)}\right)x^\top yK_iR_{L-1}. \tag{106}$$

Similarly,

$$\nabla_{K_i}a_{11}^{(i)} = a_{11}^{(i)}\left(1 - a_{11}^{(i)}\right)x^\top xQ_i - a_{11}^{(i)}a_{1L}^{(i)}y^\top xQ_iR_{1-L}, \tag{107}$$

$$\nabla_{K_i}a_{1L}^{(i)} = -a_{1L}^{(i)}a_{11}^{(i)}x^\top xQ_i + a_{1L}^{(i)}\left(1 - a_{1L}^{(i)}\right)y^\top xQ_iR_{1-L}. \tag{108}$$

Therefore,

$$\nabla_{Q_i}\mathrm{MHA}_{1,\kappa}^{\mathrm{RoPE}} = \left[a_{11}^{(i)}\left(1 - a_{11}^{(i)}\right)x^\top xK_i - a_{11}^{(i)}a_{1L}^{(i)}x^\top yK_iR_{L-1}\right] \cdot x(V_iO_i^\top)_{:,\kappa}$$
$$+ \left[-a_{1L}^{(i)}a_{11}^{(i)}x^\top xK_i + a_{1L}^{(i)}\left(1 - a_{1L}^{(i)}\right)x^\top yK_iR_{L-1}\right] \cdot y(V_iO_i^\top)_{:,\kappa}, \tag{109}$$

$$\nabla_{K_i}\mathrm{MHA}_{1,\kappa}^{\mathrm{RoPE}} = \left[a_{11}^{(i)}\left(1 - a_{11}^{(i)}\right)x^\top xQ_i - a_{11}^{(i)}a_{1L}^{(i)}y^\top xQ_iR_{1-L}\right] \cdot x(V_iO_i^\top)_{:,\kappa}$$
$$+ \left[-a_{1L}^{(i)}a_{11}^{(i)}x^\top xQ_i + a_{1L}^{(i)}\left(1 - a_{1L}^{(i)}\right)y^\top xQ_iR_{1-L}\right] \cdot y(V_iO_i^\top)_{:,\kappa}. \tag{110}$$

The constraints on $h$ associated with $Q$ and $K$ now becomes

$$
\begin{aligned}
0 =\ & \sum_{i=1}^{n} \left( \left\langle \nabla_{Q_i} \mathrm{MHA}_{1,\kappa}^{\mathrm{RoPE}}, \nabla_{Q_i} h \right\rangle + \left\langle \nabla_{K_i} \mathrm{MHA}_{1,\kappa}^{\mathrm{RoPE}}, \nabla_{K_i} h \right\rangle \right) \\
=\ & \sum_{i=1}^{n} \left\langle \left[ a_{11}^{(i)} \left( 1 - a_{11}^{(i)} \right) x^\top x K_i - a_{11}^{(i)} a_{1L}^{(i)} x^\top y K_i R_{L-1} \right] \cdot x (V_i O_i^\top)_{:,\kappa}, \nabla_{Q_i} h \right\rangle \\
& + \sum_{i=1}^{n} \left\langle \left[ -a_{1L}^{(i)} a_{11}^{(i)} x^\top x K_i + a_{1L}^{(i)} \left( 1 - a_{1L}^{(i)} \right) x^\top y K_i R_{L-1} \right] \cdot y (V_i O_i^\top)_{:,\kappa}, \nabla_{Q_i} h \right\rangle \\
& + \sum_{i=1}^{n} \left\langle \left[ a_{11}^{(i)} \left( 1 - a_{11}^{(i)} \right) x^\top x Q_i - a_{11}^{(i)} a_{1L}^{(i)} y^\top x Q_i R_{1-L} \right] \cdot x (V_i O_i^\top)_{:,\kappa}, \nabla_{K_i} h \right\rangle \\
& + \sum_{i=1}^{n} \left\langle \left[ -a_{1L}^{(i)} a_{11}^{(i)} x^\top x Q_i + a_{1L}^{(i)} \left( 1 - a_{1L}^{(i)} \right) y^\top x Q_i R_{1-L} \right] \cdot y (V_i O_i^\top)_{:,\kappa}, \nabla_{K_i} h \right\rangle \\
=\ & \sum_{i=1}^{n} a_{11}^{(i)} \left( 1 - a_{11}^{(i)} \right) \cdot x (V_i O_i^\top)_{:,\kappa} \cdot \left[ \left\langle x^\top x K_i, \nabla_{Q_i} h \right\rangle + \left\langle x^\top x Q_i, \nabla_{K_i} h \right\rangle \right] \\
& - \sum_{i=1}^{n} a_{1L}^{(i)} a_{11}^{(i)} \cdot y (V_i O_i^\top)_{:,\kappa} \cdot \left[ \left\langle x^\top x K_i, \nabla_{Q_i} h \right\rangle + \left\langle x^\top x Q_i, \nabla_{K_i} h \right\rangle \right] \\
& - \sum_{i=1}^{n} a_{1L}^{(i)} a_{11}^{(i)} \cdot x (V_i O_i^\top)_{:,\kappa} \cdot \left[ \left\langle x^\top y K_i R_{L-1}, \nabla_{Q_i} h \right\rangle + \left\langle y^\top x Q_i R_{1-L}, \nabla_{K_i} h \right\rangle \right] \\
& + \sum_{i=1}^{n} a_{1L}^{(i)} \left( 1 - a_{1L}^{(i)} \right) \cdot y (V_i O_i^\top)_{:,\kappa} \cdot \left[ \left\langle x^\top y K_i R_{L-1}, \nabla_{Q_i} h \right\rangle + \left\langle y^\top x Q_i R_{1-L}, \nabla_{K_i} h \right\rangle \right]. \quad (111)
\end{aligned}
$$

In this expression, the sum of the four terms is equal to $0$. However, from Equation (102) in **Step 2**, the first and second terms always vanish. We therefore obtain

$$
\begin{aligned}
0 =\ & -\sum_{i=1}^{n} a_{1L}^{(i)} a_{11}^{(i)} \cdot x (V_i O_i^\top)_{:,\kappa} \cdot \left[ \left\langle x^\top y K_i R_{L-1}, \nabla_{Q_i} h \right\rangle + \left\langle y^\top x Q_i R_{1-L}, \nabla_{K_i} h \right\rangle \right] \\
& + \sum_{i=1}^{n} a_{1L}^{(i)} \left( 1 - a_{1L}^{(i)} \right) \cdot y (V_i O_i^\top)_{:,\kappa} \cdot \left[ \left\langle x^\top y K_i R_{L-1}, \nabla_{Q_i} h \right\rangle + \left\langle y^\top x Q_i R_{1-L}, \nabla_{K_i} h \right\rangle \right]. \quad (112)
\end{aligned}
$$

For $i \in [h]$, define

$$
\mathfrak{c}_i := x (V_i O_i^\top)_{:,\kappa} \cdot \left[ \left\langle x^\top y K_i R_{L-1}, \nabla_{Q_i} h \right\rangle + \left\langle y^\top x Q_i R_{1-L}, \nabla_{K_i} h \right\rangle \right], \tag{113}
$$

$$
\mathfrak{d}_i := y (V_i O_i^\top)_{:,\kappa} \cdot \left[ \left\langle x^\top y K_i R_{L-1}, \nabla_{Q_i} h \right\rangle + \left\langle y^\top x Q_i R_{1-L}, \nabla_{K_i} h \right\rangle \right]. \tag{114}
$$

From Equation (112), we have

$$
0 = -\sum_{i=1}^{n} a_{1L}^{(i)} a_{11}^{(i)} \cdot \mathfrak{c}_i + \sum_{i=1}^{n} a_{1L}^{(i)} \left( 1 - a_{1L}^{(i)} \right) \cdot \mathfrak{d}_i. \tag{115}
$$

Note that, $a_{11}^{(i)}, a_{1L}^{(i)}$ are functions of $L, N, x, y$ and $\theta$, while $\mathfrak{c}_i, \mathfrak{d}_i$ are functions of $x$ and $\theta$; during our argument, their input arguments are omitted for simplicity. We express $a_{11}^{(i)}$ and $a_{1L}^{(i)}$ as follows:

$$
a_{11}^{(i)} = a_{11}^{(i)}\big((x,0,\ldots,0,y,0,\ldots,0);\theta\big) = \frac{e^{xQ_i K_i^\top x^\top}}{e^{xQ_i K_i^\top x^\top} + e^{xQ_i R_{L-1} K_i^\top y^\top} + (L-2) + N}, \tag{116}
$$

$$
a_{1L}^{(i)} = a_{1L}^{(i)}\big((x,0,\ldots,0,y,0,\ldots,0);\theta\big) = \frac{e^{xQ_i R_{L-1} K_i^\top y^\top}}{e^{xQ_i K_i^\top x^\top} + e^{xQ_i R_{L-1} K_i^\top y^\top} + (L-2) + N}. \tag{117}
$$

From Equation (115), we have

$$0 = \sum_{i=1}^{n} \frac{e^{xQ_i K_i^\top x^\top} \cdot e^{xQ_i R_{L-1} K_i^\top y^\top}}{\left(e^{xQ_i K_i^\top x^\top} + e^{xQ_i R_{L-1} K_i^\top y^\top} + (L-2) + N\right)^2} \cdot \mathfrak{c}_i + \sum_{i=1}^{n} \frac{e^{xQ_i R_{L-1} K_i^\top y^\top} \cdot \left(e^{xQ_i K_i^\top x^\top} + (L-2) + N\right)}{\left(e^{xQ_i K_i^\top x^\top} + e^{xQ_i R_{L-1} K_i^\top y^\top} + (L-2) + N\right)^2} \cdot \mathfrak{d}_i.$$
(118)

For fixed $x, y, \theta, L$, the RHS of Equation (118) defines a rational function in the variable $N$. As this function is zero for all integers $N \geq 0$, it follows that it is identically zero. Therefore, for all $t \in \mathbb{R}$, we have

$$0 = \sum_{i=1}^{n} \frac{e^{xQ_i K_i^\top x^\top} \cdot e^{xQ_i R_{L-1} K_i^\top y^\top}}{\left(e^{xQ_i K_i^\top x^\top} + e^{xQ_i R_{L-1} K_i^\top y^\top} + (L-2) + t\right)^2} \cdot \mathfrak{c}_i + \sum_{i=1}^{n} \frac{e^{xQ_i R_{L-1} K_i^\top y^\top} \cdot \left(e^{xQ_i K_i^\top x^\top} + (L-2) + t\right)}{\left(e^{xQ_i K_i^\top x^\top} + e^{xQ_i R_{L-1} K_i^\top y^\top} + (L-2) + t\right)^2} \cdot \mathfrak{d}_i.$$
(119)

This motivates the following lemma.

**Lemma B.2.** *Let* $\alpha_1, \ldots, \alpha_n$ *and* $\beta_1, \ldots, \beta_n$ *be* $2n$ *real positive numbers. Assume that* $a_i + b_i, i \in [n]$ *are pairwise distinct. The* $2n$ *functions*

$$t \mapsto \frac{\alpha_i \beta_i}{(\alpha_i + \beta_i + t)^2}, \qquad t \mapsto \frac{\beta_i(t + \alpha_i)}{(\alpha_i + \beta_i + t)^2}, \qquad \text{for } i \in [n]$$
(120)

*are linear independent.*

Lemma B.2 can be proved by analyzing the multiplicities of the poles at $t = -(\alpha_i + \beta_i)$ for each $i$. It suggests that $\mathfrak{c}_i = \mathfrak{d}_i = 0$ whenever $e^{xQ_i K_i^\top x^\top} + e^{xQ_i R_{L-1} K_i^\top y^\top}$, $i \in [n]$, are pairwise distinct. Recall the set $\mathbf{S}_1$ in **Step 2**, and for each $\theta \in \mathbf{S}_1$, define the following set

$$\mathbf{S}_\theta := \{(x, y) \in \mathbb{R}^d \times \mathbb{R}^d : e^{xQ_i K_i^\top x^\top} + e^{xQ_i R_{L-1} K_i^\top y^\top} \text{ are pairwise distinct for all } i \in [n]\}.$$
(121)

From Equation (119), for $\theta \in \mathbf{S}_1$ and $(x, y) \in \mathbf{S}_\theta$, by applying Lemma B.2, we have $\mathfrak{c}_i = \mathfrak{d}_i = 0$. Moreover, $\mathbf{S}_\theta$ is dense in $\mathbb{R}^d \times \mathbb{R}^d$, since $\mathbf{S}_\theta$ is the complement of the real zero set of the following holomorphic function

$$(x, y) \mapsto \prod_{1 \leq i < j \leq n} \left[ \left(e^{xQ_i K_i^\top x^\top} + e^{xQ_i R_{L-1} K_i^\top y^\top}\right) - \left(e^{xQ_j K_j^\top x^\top} + e^{xQ_j R_{L-1} K_j^\top y^\top}\right) \right],$$
(122)

which is not identical to 0 since $\theta \in \mathbf{S}_1$ (In other words, $\mathbf{S}_\theta$ is the complement of a proper real analytic variety in $\mathbb{R}^d \times \mathbb{R}^d$, which is always dense). Moreover, $\mathbf{S}_1$ is also dense in $\Theta_{\text{MHA}}$. We conclude that $\mathfrak{c}_i = \mathfrak{d}_i = 0$ for all $x, y$ and $\theta$. By the same argument as in the proof of the classical MHA (**Step 4**), we conclude that

$$0 = \left\langle x^\top y K_i R_{L-1}, \nabla_{Q_i} h \right\rangle + \left\langle y^\top x Q_i R_{1-L}, \nabla_{K_i} h \right\rangle.$$
(123)

**Step 4.** Equation (123) implies

$$0 = \left\langle y^\top x, K_i R_{L-1} (\nabla_{Q_i} h)^\top + (\nabla_{K_i} h) R_{L-1} Q_i^\top \right\rangle,$$
(124)

for all $x, y$, integer $L \geq 2$ and $\theta$. Since the linear span of $\{y^\top x : x, y \in \mathbb{R}^d\}$ is the whole space $\mathbb{R}^{d \times d}$, it implies that

$$0 = K_i R_{L-1} (\nabla_{Q_i} h)^\top + (\nabla_{K_i} h) R_{L-1} Q_i^\top.$$
(125)

Note that, although this identity is independent of $x$ and $y$, it still depends on the sequence length $L$. In particular, it still depends on the input $\mathbf{x}$ through its length. We further analyze this identity as follows. Equation (125) implies, for a matrix $A$ that belongs to the linear span of $\{R_{L-1} : L \geq 2\}$ in $\mathbb{R}^{d_h \times d_h}$, we have

$$0 = K_i A (\nabla_{Q_i} h)^\top + (\nabla_{K_i} h) A Q_i^\top.$$
(126)

This leads us to analyze the linear space linear span of $\{R_L : L \in \mathbb{N}\}$ in $\mathbb{R}^{d_h \times d_h}$. Let $d_h = 2m$.

**Lemma B.3.** *The linear span of* $\{R_L \colon L \in \mathbb{N}\}$ *is* $V := \{\operatorname{diag}(a_1 I + b_1 J, \ldots, a_m I + b_m J) \colon a_i, b_i \in \mathbb{R}\}$, *where*

$$I = \begin{pmatrix} 1 & 0 \\ 0 & 1 \end{pmatrix}, \qquad J = \begin{pmatrix} 0 & -1 \\ 1 & 0 \end{pmatrix}. \tag{127}$$

*Proof.* Clearly $R_L \in V$ for all $L$. Define a real-linear homomorphism $\Psi \colon V \to \mathbb{R}^{2m}$ by

$$\operatorname{diag}(a_1 I_2 + b_1 J, \ldots, a_m I_2 + b_m J) \mapsto (a_1, b_1, \ldots, a_m, b_m). \tag{128}$$

This defines an isomorphism of real vector spaces. We have

$$\Psi(R_L) = (\cos(L\varphi_1), \sin(L\varphi_1), \ldots, \cos(L\varphi_m), \sin(L\varphi_m))$$
$$= \left( \frac{e^{iL\varphi_1} + e^{-iL\varphi_1}}{2}, \frac{e^{iL\varphi_1} - e^{-iL\varphi_1}}{2i}, \ldots, \frac{e^{iL\varphi_m} + e^{-iL\varphi_m}}{2}, \frac{e^{iL\varphi_m} - e^{-iL\varphi_m}}{2i} \right). \tag{129}$$

Let $\lambda_i := e^{i\varphi_i}$. Consider the $2m \times 2m$ matrix

$$M = [\Psi(R_1), \Psi(R_2), \ldots, \Psi(R_{2m})]^{\top} = \begin{bmatrix} \dfrac{\lambda_1^1 + \lambda_1^{-1}}{2} & \dfrac{\lambda_1^1 - \lambda_1^{-1}}{2i} & \cdots & \dfrac{\lambda_m^1 + \lambda_m^{-1}}{2} & \dfrac{\lambda_m^1 - \lambda_m^{-1}}{2i} \\ \dfrac{\lambda_1^2 + \lambda_1^{-2}}{2} & \dfrac{\lambda_1^2 - \lambda_1^{-2}}{2i} & \cdots & \dfrac{\lambda_m^2 + \lambda_m^{-2}}{2} & \dfrac{\lambda_m^2 - \lambda_m^{-2}}{2i} \\ \vdots & \vdots & \ddots & \vdots & \vdots \\ \dfrac{\lambda_1^{2m} + \lambda_1^{-2m}}{2} & \dfrac{\lambda_1^{2m} - \lambda_1^{-2m}}{2i} & \cdots & \dfrac{\lambda_m^{2m} + \lambda_m^{-2m}}{2} & \dfrac{\lambda_m^{2m} - \lambda_m^{-2m}}{2i} \end{bmatrix} \tag{130}$$

Note that, $\lambda_i, i \in [2m]$ are nonzero and pairwise distinct by the design of $R_L$. The determinant of $M$ can be computed by transforming it to a nonzero scalar multiply with a Vandermonde matrix of $\lambda_i$. It follows that $M$ is invertible. Thus the $2m$ vectors $\Psi(R_i), i \in [2m]$ are linear independent in $\mathbb{R}^{2m}$. Since $\Psi$ is an isomorphism, we conclude that $\operatorname{span}_{\mathbb{R}}\{R_L \colon L \in [2m]\} = V$, which means $\operatorname{span}_{\mathbb{R}}\{R_L \colon L \in \mathbb{N}\} = V$. $\qquad\square$

Back to the problem. From Lemma B.3, it follows that

$$0 = K_i A (\nabla_{Q_i} h)^{\top} + (\nabla_{K_i} h) A Q_i^{\top}, \quad \text{for all } A \in V. \tag{131}$$

Write $Q_i, K_i$ as blocks of $d \times 2$ matrices as follows

$$Q_i = \left[ Q_i^{(1)}, \ldots, Q_i^{(m)} \right]^{\top} \quad \text{and} \quad K_i = \left[ K_i^{(1)}, \ldots, K_i^{(m)} \right]^{\top} \quad \text{for } Q_i^{(j)}, K_i^{(j)} \text{ are } d \times 2 \text{ matrices for } j \in [m]. \tag{132}$$

Therefore Equation (132) is equivalent to

$$0 = K_i^{(j)} (\nabla_{Q_i^{(j)}} h)^{\top} + (\nabla_{K_i^{(j)}} h)(Q_i^{(j)})^{\top} = K_i^{(j)} J (\nabla_{Q_i^{(j)}} h)^{\top} + (\nabla_{K_i^{(j)}} h) J (Q_i^{(j)})^{\top}, \quad \text{for all } j \in [m]. \tag{133}$$

This is exactly the set of constraints stated in Theorem 4.4. Solving these constraints leads to the invariants $\|Q_i^{(j)}\|_F^2 - \|K_i^{(j)}\|_F^2$ for $i \in [n], j \in [m]$. The same conclusion for the matrices $V_i, O_i$. This concludes the proof. $\qquad\square$

## C. Theoretical Proofs of Conservation Laws for Mixture-of-Experts

### C.1. Results on the Poles of SiLU Functions

Recall the SiLU function

$$\operatorname{SiLU}(z) = z \cdot \sigma(z), \quad \text{where } \sigma \text{ is the sigmoid function} \quad \sigma(z) = \frac{1}{1 + e^{-z}}. \tag{134}$$

SiLU can be considered as a meromorphic function, whose poles are exactly the zeros of $1 + e^{-z}$ with corresponding multiplicity. For $a$ in $\mathbb{R}$, we now analyze the poles of $\operatorname{SiLU}(az)$. If $a = 0$, then $\operatorname{SiLU}(az) = 0$ for all $z$. If $a \neq 0$, then the set of poles of $\operatorname{SiLU}(az) = 0$ is $\{i(2k+1)\pi/a \colon k \in \mathbb{Z}\}$. Each pole of $\operatorname{SiLU}(az)$ is a simple pole.

**Lemma C.1.** *Let $S$ be a countable subset of $\mathbb{C}$. For $a \in \mathbb{R}$, define $f \colon \mathbb{C} \to \mathbb{C}$ such that*

$$f_a(z) = 1 + e^{az}, \quad \text{for all } z \in \mathbb{C}. \tag{135}$$

*Consider the subset $A \subset \mathbb{R}$ consisting of all $a \in \mathbb{R}$ such that $f_a$ has no zeros in $S$, i.e.*

$$A := \{a \in \mathbb{R} : f_a(z) \neq 0 \text{ for all } z \in S\} \subset \mathbb{R}. \tag{136}$$

*Then $A$ is cocountable in $\mathbb{R}$.*

*Proof.* We have

$$\mathbb{R} \setminus A = \{a \in \mathbb{R} : f_a(z) = 0 \text{ for some } z \in S\} = \bigcup_{z \in S} \{a \in \mathbb{R} : f_a(z) = 0\}. \tag{137}$$

Note that, for any $z \in \mathbb{C}$, $f_a(z) = 0$ is equivalent to $az = i(2k+1)\pi$ for some $k \in \mathbb{Z}$. Hence, the set $\{a \in \mathbb{R} : f_a(z) = 0\}$ is either empty or countable. From Equation (137), since $S$ is countable, we have $\mathbb{R} \setminus A$ is countable. $\square$

## C.2. Conservation Laws for Dense Mixture-of-Experts with Softmax Gating – Theorem 4.5

The *Mixture-of-Experts with dense gating* is defined as the function $\mathrm{MoE} \colon \mathbb{R}^d \to \mathbb{R}^d$ given by

$$\mathrm{MoE}\big(x; W, \{\theta_i\}_{i=1}^n\big) = \sum_{i=1}^{n} g_i(x; W) \cdot \mathrm{E}(x; \theta_i), \tag{138}$$

where the gating weight $g_i(x; W)$ is defined by

$$g_i(x; W) := \mathrm{softmax}_i(Wx) = e^{W^{(i)}x} \Big/ \sum_{p=1}^{n} e^{W^{(p)}x},$$

with $W^{(i)} := W_{i,:}$ denoting the $i$-th row of $W$. The score $g_i(x; W)$ specifies the relative contribution of the $i$-th expert to the final output. The parameter space of MoE is given by

$$\Theta_{\mathrm{MoE}} = \mathbb{R}^{n \times d} \times (\Theta_{\mathrm{SwiGLU}})^n.$$

We now present the proof of the conservation laws for Mixture-of-Experts with softmax gating. Note that, compared to the notation in the main text, we write $W^{(i)}$ in place of $W_i$ in order to index the entries (rows) of the gating matrix $W$. Following the observation in Section 3, we restrict attention to the scalar-valued setting, so that the MoE map, and hence each expert network, is real-valued.

*Proof.* **Step 1.** The MoE with softmax dense gating is $\mathcal{C}^1$ on its parameter space. Its gradients are given as follows. For the parameters of Experts, we have

$$\nabla_{A^{(i)}} \mathrm{MoE} = g_i \cdot \nabla_A \mathrm{E}(x; A^{(i)}, B^{(i)}, C^{(i)}) = g_i \cdot \big(\mathrm{SiLU}(B^{(i)}x) \odot (C^{(i)}x)\big)^\top \tag{139}$$

$$\nabla_{B^{(i)}} \mathrm{MoE} = g_i \cdot \nabla_B \mathrm{E}(x; A^{(i)}, B^{(i)}, C^{(i)}) = g_i \cdot \big((A^{(i)})^\top \odot ((C^{(i)}x) \odot \mathrm{SiLU}'(B^{(i)}x))\big)x^\top \tag{140}$$

$$\nabla_{C^{(i)}} \mathrm{MoE} = g_i \cdot \nabla_C \mathrm{E}(x; A^{(i)}, B^{(i)}, C^{(i)}) = g_i \cdot \big((A^{(i)})^\top \odot \mathrm{SiLU}(B^{(i)}x)\big)x^\top \tag{141}$$

For the parameters of the gating, we have $\nabla_W g_i = g_i(e_i - g)x^\top$, which implies $\nabla_{W^{(i)}} g_j = g_j(\delta_{ij} - g_i)x^\top$. Thus,

$$\nabla_{W^{(i)}} \mathrm{MoE}(x) = \sum_{j=1}^{n} \nabla_{W^{(i)}} g_j \cdot \mathrm{E}_j = \sum_{j=1}^{n} g_j(\delta_{ij} - g_i)x^\top \cdot \mathrm{E}_j = \left(\sum_{j=1}^{n} g_j(\delta_{ij} - g_i) \cdot \mathrm{E}_j\right) x^\top$$

$$= \left(\sum_{j=1}^{n} g_j \delta_{ij} \cdot \mathrm{E}_j - \sum_{j=1}^{n} g_j g_i \cdot \mathrm{E}_j\right) x^\top = \left(g_i \cdot \mathrm{E}_i - g_i \cdot \sum_{j=1}^{n} g_j \cdot \mathrm{E}_j\right) x^\top = g_i \cdot (\mathrm{E}_i - \mathrm{MoE})x^\top. \tag{142}$$

Let $h\colon \Theta_{\mathrm{MoE}} \to \mathbb{R}$ be a conservation law for MoE. We have, for all $\theta \in \Theta_{\mathrm{MoE}}$ and $x \in \mathbb{R}^d$,

$$
\begin{aligned}
0 &= \sum_{i=1}^{n} \left( \left\langle \nabla_{W^{(i)}} \mathrm{MoE}, \nabla_{W^{(i)}} h \right\rangle + \left\langle \nabla_{A^{(i)}} \mathrm{MoE}, \nabla_{A^{(i)}} h \right\rangle + \left\langle \nabla_{B^{(i)}} \mathrm{MoE}, \nabla_{B^{(i)}} h \right\rangle + \left\langle \nabla_{C^{(i)}} \mathrm{MoE}, \nabla_{C^{(i)}} h \right\rangle \right) \\
&= \sum_{i=1}^{n} \left( \left\langle g_i \cdot (\mathrm{E}_i - \mathrm{MoE}) x^\top, \nabla_{W^{(i)}} h \right\rangle + \left\langle g_i \cdot \left( \mathrm{SiLU}(B^{(i)} x) \odot (C^{(i)} x) \right)^\top, \nabla_{A^{(i)}} h \right\rangle \right. \\
&\qquad \left. + \left\langle g_i \cdot \left( (A^{(i)})^\top \odot \left( (C^{(i)} x) \odot \mathrm{SiLU}'(B^{(i)} x) \right) \right) x^\top, \nabla_{B^{(i)}} h \right\rangle + \left\langle g_i \cdot \left( (A^{(i)})^\top \odot \mathrm{SiLU}(B^{(i)} x) \right) x^\top, \nabla_{C^{(i)}} h \right\rangle \right) \\
&= \sum_{i=1}^{n} g_i \left( \left\langle (\mathrm{E}_i - \mathrm{MoE}) x^\top, \nabla_{W^{(i)}} h \right\rangle + \left\langle \left( \mathrm{SiLU}(B^{(i)} x) \odot (C^{(i)} x) \right)^\top, \nabla_{A^{(i)}} h \right\rangle \right. \\
&\qquad \left. + \left\langle \left( (A^{(i)})^\top \odot \left( (C^{(i)} x) \odot \mathrm{SiLU}'(B^{(i)} x) \right) \right) x^\top, \nabla_{B^{(i)}} h \right\rangle + \left\langle \left( (A^{(i)})^\top \odot \mathrm{SiLU}(B^{(i)} x) \right) x^\top, \nabla_{C^{(i)}} h \right\rangle \right). \quad (143)
\end{aligned}
$$

Fixed $j \in [d]$. Set $x = t e_j$ where $t \in \mathbb{R}$ and $e_j \in \mathbb{R}^{d \times 1}$ is the $j$-th basis vector. Then Equation (143) becomes

$$
\begin{aligned}
0 &= \sum_{i=1}^{n} g_i \left( (\mathrm{E}_i - \mathrm{MoE}) t \cdot \partial_{W_j^{(i)}} h + \sum_{k=1}^{d_1} \left( \mathrm{SiLU}(B_{kj}^{(i)} t)(C_{kj}^{(i)} t) \right) \cdot \partial_{A_k^{(i)}} h \right. \\
&\qquad \left. + \sum_{k=1}^{d_1} \left( A_k^{(i)} (C_{kj}^{(i)} t) \mathrm{SiLU}'(B_{kj}^{(i)} t) t \right) \cdot \partial_{B_{kj}^{(i)}} h + \sum_{k=1}^{d_1} \left( A_k^{(i)} \mathrm{SiLU}(B_{kj}^{(i)} t) t \right) \cdot \partial_{C_{kj}^{(i)}} h \right) \\
&= \sum_{i=1}^{n} g_i t \left( (\mathrm{E}_i - \mathrm{MoE}) \cdot \partial_{W_j^{(i)}} h + \sum_{k=1}^{d_1} \left( \mathrm{SiLU}(B_{kj}^{(i)} t) C_{kj}^{(i)} \right) \cdot \partial_{A_k^{(i)}} h \right. \\
&\qquad \left. + \sum_{k=1}^{d_1} \left( A_k^{(i)} (C_{kj}^{(i)} t) \mathrm{SiLU}'(B_{kj}^{(i)} t) \right) \cdot \partial_{B_{kj}^{(i)}} h + \sum_{k=1}^{d_1} \left( A_k^{(i)} \mathrm{SiLU}(B_{kj}^{(i)} t) \right) \cdot \partial_{C_{kj}^{(i)}} h \right) \\
&= \sum_{i=1}^{n} g_i t \left( (\mathrm{E}_i - \mathrm{MoE}) \cdot \partial_{W_j^{(i)}} h + \sum_{k=1}^{d_1} \left( A_k^{(i)} C_{kj}^{(i)} \cdot \partial_{B_{kj}^{(i)}} h \cdot t \mathrm{SiLU}'(B_{kj}^{(i)} t) \right) \right. \\
&\qquad \left. + \sum_{k=1}^{d_1} \left( C_{kj}^{(i)} \cdot \partial_{A_k^{(i)}} h + A_k^{(i)} \cdot \partial_{C_{kj}^{(i)}} h \right) \cdot \mathrm{SiLU}(B_{kj}^{(i)} t) \right). \quad (144)
\end{aligned}
$$

By continuity of $t \in \mathbb{R}$, we have

$$
\begin{aligned}
0 &= \sum_{i=1}^{n} g_i \left( (\mathrm{E}_i - \mathrm{MoE}) \cdot \partial_{W_j^{(i)}} h + \sum_{k=1}^{d_1} \left( A_k^{(i)} C_{kj}^{(i)} \cdot \partial_{B_{kj}^{(i)}} h \cdot t \mathrm{SiLU}'(B_{kj}^{(i)} t) \right) \right. \\
&\qquad \left. + \sum_{k=1}^{d_1} \left( C_{kj}^{(i)} \cdot \partial_{A_k^{(i)}} h + A_k^{(i)} \cdot \partial_{C_{kj}^{(i)}} h \right) \cdot \mathrm{SiLU}(B_{kj}^{(i)} t) \right). \quad (145)
\end{aligned}
$$

for all $t \in \mathbb{R}$. Note that, in Equation (145), $g_i, \mathrm{E}_i, \mathrm{MoE}_i$ are functions of $x = t e_j$; during our argument, their input arguments are omitted for simplicity.

**Step 2.** For $i \in [d]$ and $k \in [d_1]$, define the following functions:

$$
\mathfrak{a}^{(i)}(t) := g_i \cdot (\mathrm{E}_i - \mathrm{MoE}) \cdot \partial_{W_j^{(i)}} h, \quad (146)
$$

$$
\mathfrak{b}_k^{(i)}(t) := g_i \cdot A_k^{(i)} C_{kj}^{(i)} \cdot \partial_{B_{kj}^{(i)}} h \cdot t \mathrm{SiLU}'(B_{kj}^{(i)} t), \quad (147)
$$

$$
\mathfrak{c}_k^{(i)}(t) := g_i \cdot \left( C_{kj}^{(i)} \cdot \partial_{A_k^{(i)}} h + A_k^{(i)} \cdot \partial_{C_{kj}^{(i)}} h \right) \cdot \mathrm{SiLU}(B_{kj}^{(i)} t). \quad (148)
$$

From Equation (145), the sum of these functions is zero, that is,

$$0 = \sum_{i=1}^{n} \mathfrak{a}^{(i)}(t) + \sum_{i=1}^{n} \sum_{k=1}^{d_1} \mathfrak{b}_k^{(i)}(t) + \sum_{i=1}^{n} \sum_{k=1}^{d_1} \mathfrak{c}_k^{(i)}(t). \tag{149}$$

We now consider $\mathfrak{a}^{(i)}, \mathfrak{b}_k^{(i)}, \mathfrak{c}_k^{(i)}$ as complex functions $\mathfrak{a}_i(z), \mathfrak{b}_i(z), \mathfrak{c}_i(z)$ for $z \in \mathbb{C}$. By design, they are meromorphic functions on $\mathbb{C}$, since they are given as quotients of holomorphic functions whose denominators are nonzero for all $\theta$. The poles of these functions belongs to the poles of $g_i(ze_j)$ and $\mathrm{SiLU}(B_{kj}^{(i)}z)$ for all $i \in [n]$ and $k \in [d_1]$. Note that, the poles of $\mathrm{SiLU}'(B_{kj}^{(i)}z)$ are exactly the poles of $\mathrm{SiLU}(B_{kj}^{(i)}z)$. Define the following sets:

$$\mathbf{S}_1 := \{\theta : \text{ for each } i \in [n] \text{ and } k \in [d_1], \mathrm{SiLU}(B_{kj}^{(i)}z) \text{ has at least one pole}\}, \tag{150}$$

$$\mathbf{S}_2 := \{\theta : \mathrm{SiLU}(B_{kj}^{(i)}z) \text{ and } \mathrm{SiLU}(B_{k'j}^{(i')}z) \text{ do not have common poles for all } (i,k) \neq (i',k')\}, \tag{151}$$

$$\mathbf{S}_3 := \{\theta : \mathrm{SiLU}(B_{kj}^{(i)}z) \text{ and } g_{i'}(ze_j) \text{ do not have common poles for all } i, i' \in [n] \text{ and } k \in [d_1]\}. \tag{152}$$

Let $\theta \in \mathbf{S}_1 \cap \mathbf{S}_2 \cap \mathbf{S}_3$. In Equation (149), for each $i \in [n], k \in [d_1]$, consider a pole $z_0$ of $\mathrm{SiLU}(B_{kj}^{(i)}z)$ (which exists by $\mathbf{S}_1$). Note that, $g_i$ does not have zero, and $z_0$ is nonzero (as in Appendix C.1). Thus, $z_0$ is a double pole of $\mathfrak{b}_k^{(i)}$ (if $\mathfrak{b}_k^{(i)}$ is not identical to 0), while it is at most a simple pole, or not a pole at all, of any other function among $\mathfrak{a}, \mathfrak{b}, \mathfrak{c}$ (by $\mathbf{S}_2, \mathbf{S}_3$). We conclude that $\mathfrak{b}_k^{(i)} \equiv 0$. Define the following set

$$\mathbf{S}_4 = \{\theta : A_k^{(i)} \text{ and } C_{jk}^{(i)} \text{ are nonzero for all } i \in [n] \text{ and } k \in [d_1]\}. \tag{153}$$

Then $\mathfrak{b}_k^{(i)} \equiv 0$ implies $\partial_{B_{kj}^{(i)}} h = 0$ for all $\theta \in \mathbf{S}_1 \cap \mathbf{S}_2 \cap \mathbf{S}_3 \cap \mathbf{S}_4$.

We have the following observations on the topology of $\mathbf{S}_1, \mathbf{S}_2, \mathbf{S}_3, \mathbf{S}_4$:

- From Appendix C.1, the set $\mathbf{S}_1$ and $\mathbf{S}_2$ are simply

$$\mathbf{S}_1 = \{\theta : B_{kj}^{(i)} \text{ are nonzero for all } i \in [n] \text{ and } k \in [d_1]\}, \tag{154}$$

$$\mathbf{S}_2 = \{\theta : (B_{kj}^{(i)})^2 \text{ are pairwise distinct for all } i \in [n] \text{ and } k \in [d_1]\}. \tag{155}$$

  It implies $\mathbf{S}_1, \mathbf{S}_2$, and also $\mathbf{S}_4$, are open and dense in $\Theta_{\mathrm{MoE}}$.

- Fix $W \in \mathbb{R}^{n \times d}$. Since the map $z \mapsto \prod_{i=1}^{n} g_i(ze_j; W)$ is meromorphic, its set of poles is countable. The set $\mathbf{S}_3$ is defined to consist of all parameters $\theta$ such that none of the functions $\mathrm{SiLU}(B_{kj}^{(i)}z)$ has a pole in this countable set. By Lemma C.1, it follows that $\mathbf{S}_3$ is dense in $\Theta_{\mathrm{MoE}}$.

The intersection $\mathbf{S}_1 \cap \mathbf{S}_2 \cap \mathbf{S}_3 \cap \mathbf{S}_4$ is dense, since it is the intersection of three open dense sets and one dense set. Moreover, because $h$ is $\mathcal{C}^1$ and $\partial_{B_{kj}^{(i)}} h = 0$ for all $\theta \in \mathbf{S}_1 \cap \mathbf{S}_2 \cap \mathbf{S}_3 \cap \mathbf{S}_4$, this identity extends to all of $\Theta_{\mathrm{MoE}}$. As this holds for arbitrary $i, j, k$, we conclude that $\nabla_{B^{(i)}} h = 0$ for all $i \in [n]$.

**Step 3.** It follows that Equation (145) reduces to

$$0 = \sum_{i=1}^{n} g_i \left( (\mathrm{E}_i - \mathrm{MoE}) \cdot \partial_{W_j^{(i)}} h + \sum_{k=1}^{d_1} \left( C_{kj}^{(i)} \cdot \partial_{A_k^{(i)}} h + A_k^{(i)} \cdot \partial_{C_{kj}^{(i)}} h \right) \cdot \mathrm{SiLU}(B_{kj}^{(i)}t) \right). \tag{156}$$

Evaluating the value of $\mathrm{E}_i$ at $x = te_j$. We have

$$\mathrm{E}_i(te_j) = \sum_{k=1}^{d_1} A_k^{(i)} \mathrm{SiLU}(B_{kj}^{(i)}t)(C_{kj}^{(i)}t). \tag{157}$$

Substituting this expression in Equation ([156](#)), we have

$$0 = \sum_{i=1}^{n} g_i \left[ \left( \sum_{k=1}^{d_1} A_k^{(i)} \mathrm{SiLU}(B_{kj}^{(i)}t)(C_{kj}^{(i)}t) - \sum_{i'=1}^{n} \left( g_{i'} \sum_{k=1}^{d_1} A_k^{(i')} \mathrm{SiLU}(B_{kj}^{(i')}t)(C_{kj}^{(i')}t) \right) \right) \cdot \partial_{W_j^{(i)}} h \right.$$

$$\left. + \sum_{k=1}^{d_1} \left( C_{kj}^{(i)} \cdot \partial_{A_k^{(i)}} h + A_k^{(i)} \cdot \partial_{C_{kj}^{(i)}} h \right) \cdot \mathrm{SiLU}(B_{kj}^{(i)}t) \right]$$

$$= \sum_{i=1}^{n} \sum_{k=1}^{d_1} \mathrm{SiLU}(B_{kj}^{(i)}t) \cdot$$

$$\left[ g_i \left( C_{kj}^{(i)} \cdot \partial_{A_k^{(i)}} h + A_k^{(i)} \cdot \partial_{C_{kj}^{(i)}} h \right) + g_i A_k^{(i)} C_{kj}^{(i)} t \cdot \partial_{W_j^{(i)}} h - g_i A_k^{(i)} C_{kj}^{(i)} t \cdot \left( \sum_{i'=1}^{n} g_{i'} \partial_{W_j^{(i')}} h \right) \right]$$

$$= \sum_{i=1}^{n} \sum_{k=1}^{d_1} \mathrm{SiLU}(B_{kj}^{(i)}t) \cdot \mathfrak{d}_k^{(i)}(t), \tag{158}$$

where

$$\mathfrak{d}_k^{(i)}(t) \coloneqq g_i \left( C_{kj}^{(i)} \cdot \partial_{A_k^{(i)}} h + A_k^{(i)} \cdot \partial_{C_{kj}^{(i)}} h \right) + g_i A_k^{(i)} C_{kj}^{(i)} t \cdot \partial_{W_j^{(i)}} h - g_i A_k^{(i)} C_{kj}^{(i)} t \cdot \left( \sum_{i'=1}^{n} g_{i'} \partial_{W_j^{(i')}} h \right). \tag{159}$$

Note that, these $\mathfrak{d}$ functions do not depend on $B^{(i)}$ for any $i$, since $\nabla_{B^{(i)}} h = 0$ everywhere. We now consider $\mathrm{SiLU}(B_{kj}^{(i)}t)$ and $\mathfrak{d}_k^{(i)}$ as complex functions $\mathrm{SiLU}(B_{kj}^{(i)}z)$ and $\mathfrak{d}_k^{(i)}(z)$. By design, they are meromorphic functions on $\mathbb{C}$, since they are given as quotients of holomorphic functions whose denominators are nonzero for all $\theta$. The poles of these functions belongs to the poles of $g_i(ze_j)$ and $\mathrm{SiLU}(B_{kj}^{(i)}t)$ for all $i \in [n]$ and $k \in [d_1]$. Note that, the poles of $\mathrm{SiLU}'(B_{kj}^{(i)}t)$ are exactly the poles of $\mathrm{SiLU}(B_{kj}^{(i)}t)$. Recall the sets $\mathbf{S}_1, \mathbf{S}_2, \mathbf{S}_3$ from **Step 2**, and define the following set

$$\mathbf{S}_5 \coloneqq \{ \theta : \text{ for each } i \in [n] \text{ and } k \in [d_1], \text{ we have either } \mathfrak{d}_k^{(i)} \equiv 0, \text{ or}$$

$$\text{the zeros of } \mathfrak{d}_k^{(i)}(z) \text{ does not contain any pole of } \mathrm{SiLU}(B_{kj}^{(i)}z) \}. \tag{160}$$

Let $\theta \in \mathbf{S}_1 \cap \mathbf{S}_2 \cap \mathbf{S}_3 \cap \mathbf{S}_5$ and fix a pair $i \in [n], k \in [d_1]$. If $\mathfrak{d}_k^{(i)} \not\equiv 0$, in Equation ([158](#)), consider a pole $z_0$ of $\mathrm{SiLU}(B_{kj}^{(i)}z)$ (which exists by $\mathbf{S}_1$). Thus, $z_0$ is a single pole of $\mathrm{SiLU}(B_{kj}^{(i)}z) \cdot \mathfrak{d}_k^{(i)}(z)$ (by $\mathbf{S}_5$), while it is not a pole of any $\mathrm{SiLU}(B_{k'j}^{(i')}z) \cdot \mathfrak{d}_{k'}^{(i')}(z)$ for $(i', k') \neq (i, k)$ (by $\mathbf{S}_2, \mathbf{S}_3$). This is a contradiction since the sum of those functions is $0$. We conclude that $\mathfrak{d}_k^{(i)} \equiv 0$. Thus, for all $i \in [n]$, $k \in [d_1]$, and $\theta \in \mathbf{S}_1 \cap \mathbf{S}_2 \cap \mathbf{S}_3 \cap \mathbf{S}_5$, we have

$$0 = g_i \left( C_{kj}^{(i)} \cdot \partial_{A_k^{(i)}} h + A_k^{(i)} \cdot \partial_{C_{kj}^{(i)}} h \right) + g_i A_k^{(i)} C_{kj}^{(i)} t \cdot \partial_{W_j^{(i)}} h - g_i A_k^{(i)} C_{kj}^{(i)} t \cdot \left( \sum_{i'=1}^{n} g_{i'} \partial_{W_j^{(i')}} h \right). \tag{161}$$

Recall the set $\mathbf{S}_4$ and consider $\theta \in \mathbf{S}_1 \cap \mathbf{S}_2 \cap \mathbf{S}_3 \cap \mathbf{S}_4 \cap \mathbf{S}_5$. By substituting $t = 0$ and dividing for non-zero terms in Equation ([161](#)), we obtain

$$0 = C_{kj}^{(i)} \cdot \partial_{A_k^{(i)}} h + A_k^{(i)} \cdot \partial_{C_{kj}^{(i)}} h = \partial_{W_j^{(i)}} h - \sum_{i'=1}^{n} g_{i'} \partial_{W_j^{(i')}} h. \tag{162}$$

We now make several observations regarding the topology of these intersections. By the preceding arguments, the sets $\mathbf{S}_1$, $\mathbf{S}_2$, and $\mathbf{S}_4$ are open and dense. In contrast, the openness of $\mathbf{S}_3$ and $\mathbf{S}_5$ is not guaranteed. Fix $W$ and matrices $A^{(i)}, C^{(i)}$ for all $i \in [n]$. Then:

- In order for $\theta$ to belong to $\mathbf{S}_3$, each function $\mathrm{SiLU}(B_{kj}^{(i)}z)$ must avoid having poles in a certain countable set, namely the set of zeros of the meromorphic function $z \mapsto \prod_{i=1}^{n} g_i(ze_j; W)$.

- Similarly, for $\theta$ to belong to $\mathbf{S}_5$, for each $i \in [n]$ and $k \in [d_1]$ with $\mathfrak{d}_k^{(i)} \not\equiv 0$, the function $\mathrm{SiLU}(B_{kj}^{(i)} z)$ must avoid poles in the countable set of zeros of $z \mapsto \mathfrak{d}_k^{(i)}$.

- Consequently, membership in $\mathbf{S}_3 \cap \mathbf{S}_5$ requires $\mathrm{SiLU}(B_{kj}^{(i)} z)$ to avoid poles in a countable set. By Lemma C.1, it follows that $\mathbf{S}_3 \cap \mathbf{S}_5$ is dense in $\Theta_{\mathrm{MoE}}$.

The intersection $\mathbf{S}_1 \cap \mathbf{S}_2 \cap \mathbf{S}_4 \cap (\mathbf{S}_3 \cap \mathbf{S}_5)$ is the intersection of three open dense sets and one dense set, and is therefore dense. We may thus conclude that, for all $\theta \in \Theta_{\mathrm{MoE}}$,

$$0 = C_{kj}^{(i)} \cdot \partial_{A_k^{(i)}} h + A_k^{(i)} \cdot \partial_{C_{kj}^{(i)}} h = \partial_{W_j^{(i)}} h - \sum_{i'=1}^{n} g_{i'} \partial_{W_j^{(i')}} h. \tag{163}$$

The first implies $0 = C_{k,:}^{(i)} \cdot \partial_{A_k^{(i)}} h + A_k^{(i)} \cdot \nabla_{C_{k,:}^{(i)}} h$ for all $k \in [d_1]$ and $i \in [n]$, while the latter implies $\partial_{W_j^{(1)}} h = \ldots = \partial_{W_j^{(n)}} h$ for all $j \in [d]$, which is equivalent to $\nabla_{W^{(1)}} h = \ldots = \nabla_{W^{(n)}} h$. Solving the constraints for $h$ with respect to the expert parameters is exactly the same as in Theorem 4.2, while solving the constraints with respect to the gating parameter leads to the invariant $\sum_{i=1}^{n} W^{(i)}$. $\square$

## C.3. Conservation Laws for Sparse Mixture-of-Experts with Softmax Gating – Theorem 4.6

We now present the proof of the conservation laws for Sparse Mixture-of-Experts with softmax gating. The notations are kept the same as in the case of MoE.

*Proof.* Let $h \colon \Theta_{\mathrm{MoE}} \to \mathbb{R}$ be a conservation law for SMoE.

**Step 1.** Fix a subset $T \subset [n]$ of $k$ elements, and define the vector $e_T := \sum_{i \in T} e_i \in \mathbb{R}^n$. In other words, the $i$-th entry of $e_T$ is 1 if $i \in T$ and 0 if $i \notin T$. Define following the real open $n$-box of $\mathbb{R}^n$, centered at $e_T$

$$\Omega(T) := \{ e_T + v \ : \ v \in \mathbb{R}^n \text{ such that } |v_i| < 1/3 \text{ for all } i \in [n] \} \subset \mathbb{R}^n. \tag{164}$$

It is clear that $\mathrm{Top}\text{-}k(s) = T$ for all $s \in \Omega_T$. Now, define the linear map $\mathcal{F} \colon \mathbb{R}^d \times \Theta_{\mathrm{MoE}} \to \mathbb{R}^n$ such that $(x, \theta) \mapsto Wx$, and denote the inverse image of $\Omega(T)$ via $\mathcal{F}$ as $\Lambda(T)$. By the construction, it follows that for all $(x, \theta) \in \Lambda(T)$, we have $\mathrm{Top}\text{-}k(Wx) = T$, and

$$\mathrm{SMoE}(x; \theta) = \sum_{i \in T} \mathrm{softmax}_i(W^{(i)} x \ : \ i \in T) \cdot \mathrm{E}_i(x). \tag{165}$$

We have the following observations on the set $\Lambda(T)$.

*(i)* $\Lambda(T)$ is nonempty and open in $\mathbb{R}^d \times \Theta_{\mathrm{MoE}}$. Moreover, the set

$$\Theta_{\mathrm{MoE}}(T) := \{ \theta \in \Theta_{\mathrm{MoE}} \ : \ \text{there exists } x \in \mathbb{R}^d \text{ such that } (x, \theta) \in \Lambda(T) \} \tag{166}$$

is dense in $\Theta_{\mathrm{MoE}}$.

Indeed, since $\mathcal{F}$ is continuous and $\Omega(T)$ is open in $\mathbb{R}^n$, $\Lambda(T)$ open in $\mathbb{R}^d \times \Theta_{\mathrm{MoE}}$. Define the following set

$$\mathbf{S}_1 := \{ \theta \in \Theta_{\mathrm{MoE}} \ : \ \text{the set } \{ W^{(i)} \colon i \in [n] \} \text{ is linear independent in } \mathbb{R}^d \}. \tag{167}$$

By the assumption $d > n$, we have $\mathbf{S}_1$ is dense in $\Theta_{\mathrm{MoE}}$. Moreover, the linear dependency implies that for any $\theta \in \mathbf{S}_1$, there exists $x \in \mathbb{R}^d$ such that $e_T = Wx$. In particular, $\theta \in \Theta_{\mathrm{MoE}}(T)$. Therefore, $\mathbf{S}_1 \subset \Theta_{\mathrm{MoE}}(T)$. Since $\mathbf{S}_1$ is dense, $\Theta_{\mathrm{MoE}}(T)$ is dense.

*(ii)* For all $(x, \theta) \in \Lambda(T)$, $\mathrm{SMoE}(x; \cdot)$ is $\mathcal{C}^2$-differentiable in a neighborhood of $\theta$.

For $(x, \theta) \in \Lambda(T)$, since $\Lambda(T)$ is open, the Top-$k$ output remains $T$ in a neighborhood of $(x, \theta)$. As in Equation (165), $\mathrm{SMoE}(x, \cdot)$ is $\mathcal{C}^2$-differentiable in a neighborhood of $\theta$.

Define

$$f_T \colon \mathbb{R}^d \times \Theta_{\mathrm{MoE}} \to \mathbb{R}^d, \qquad f_T(x; \theta) := \sum_{i \in T} \mathrm{softmax}_i(W^{(i)}x \ : \ i \in T) \cdot \mathrm{E}_i(x). \tag{168}$$

Note that $f_T$ is a Mixture-of-Experts of $k$ experts, and is $\mathcal{C}^1$. We have $f_T \equiv \mathrm{SMoE}$ on $\Lambda(T)$. Let $h \colon \Theta_{\mathrm{MoE}} \to \mathbb{R}$ be a conservation law for SMoE. For all $(x, \theta) \in \Lambda(T)$, we have

$$0 = \langle \nabla_\theta \mathrm{SMoE}, \nabla_\theta h \rangle, \quad \text{which means} \quad 0 = \langle \nabla_\theta f_T, \nabla_\theta h \rangle. \tag{169}$$

Note that this holds only on $\Lambda(T)$. Observe that for each $\theta \in \Theta_{\mathrm{MoE}}(T)$, since $\Lambda(T)$ is open, there exists an open set $U$ of $\mathbb{R}^d$ such that $U \times \{\theta\} \subset \Lambda(T)$. In particular, $\langle \nabla_\theta f_T(z; \theta), \nabla_\theta h(\theta) \rangle = 0$ for all $z \in U$. Moreover, $z \mapsto \langle \nabla_\theta f_T(z; \theta), \nabla_\theta h(\theta) \rangle$ defines a meromorphic function. It follows that this meromorphic function is identical to 0. We conclude that,

$$0 = \langle \nabla_\theta f_T(x, \theta), \nabla_\theta h(\theta) \rangle, \text{ for all } \theta \in \Theta_{\mathrm{MoE}}(T) \text{ and } x \in \mathbb{R}^d. \tag{170}$$

The RHS of Equation (170) defines a continuous function since $h$ is $\mathcal{C}^1$. Since $\Theta_{\mathrm{MoE}}(T)$ is dense in $\Theta_{\mathrm{MoE}}$, Equation (170) holds for all $\theta \in \Theta_{\mathrm{MoE}}$ and $x \in \mathbb{R}^d$. In other words, by ignoring the gating weight and expert weights of the $i$-th experts for all $i \notin T$, $h$ can be considered as a conservation laws for $f_T$. By the result on conservation laws for MoE, we have

$$0 = C_{k,:}^{(i)} \cdot \partial_{A_k^{(i)}} h + A_k^{(i)} \cdot \nabla_{C_{k,:}^{(i)}} h \text{ for all } k \in [d_1] \text{ and } i \in T, \text{ and } \nabla_{W^{(i)}} h = \nabla_{W^{(j)}} h \text{ for all } i, j \in T. \tag{171}$$

This holds for any choice of $T \subset [n]$ of $k$ elements. Since $k \geq 2$, varying $T$ leads to

$$0 = C_{k,:}^{(i)} \cdot \partial_{A_k^{(i)}} h + A_k^{(i)} \cdot \nabla_{C_{k,:}^{(i)}} h \text{ for all } k \in [d_1] \text{ and } i \in [n], \text{ and } \nabla_{W^{(1)}} h = \ldots = \nabla_{W^{(j)}} h. \tag{172}$$

This concludes the proof. $\qquad\qquad\qquad\qquad\qquad\qquad\qquad\qquad\qquad\qquad\qquad\qquad\qquad\qquad\qquad\qquad\quad \square$

### C.4. Conservation Laws for Mixture-of-Experts with Sigmoid Gating – Theorem 4.7

The proof for MoE with sigmoid gating follows the same overall structure as in the softmax-gated case. Although the gating weights take a different form, the resulting identities can be treated in an analogous manner. In particular, the core relation used in the proof of the softmax MoE case, given in Equation (145), now becomes

$$0 = \sum_{i=1}^n g_i \Bigg( \left( 1 - \sigma\left( tW_j^{(i)} \right) \right) (\mathrm{E}_i - \mathrm{MoE}) \cdot \partial_{W_j^{(i)}} h + \sum_{k=1}^{d_1} \left( A_k^{(i)} C_{kj}^{(i)} \cdot \partial_{B_{kj}^{(i)}} h \cdot t\mathrm{SiLU}'(B_{kj}^{(i)}t) \right)$$
$$+ \sum_{k=1}^{d_1} \left( C_{kj}^{(i)} \cdot \partial_{A_k^{(i)}} h + A_k^{(i)} \cdot \partial_{C_{kj}^{(i)}} h \right) \cdot \mathrm{SiLU}(B_{kj}^{(i)}t) \Bigg). \tag{173}$$

Using this identity of meromorphic functions, the remainder of the proof proceeds exactly as in the softmax-gated MoE case in Appendix C.2. The sparse-gating case can be reduced to the dense-gating case via the density argument in Appendix C.3.

## D. Experimental Configuration

**CIFAR-10.** For CIFAR-10, we employ a Vision Transformer composed of 6 Transformer layers with a hidden size of 256 and an intermediate (MLP) dimension of 1024. The model uses 4 attention heads and a patch size of 4. Training is performed using stochastic gradient descent (SGD) with a batch size of 5000, gradient accumulation steps of 10, and no weight decay. We train for 300 epochs with learning rates of $\{1 \times 10^{-4}, 4 \times 10^{-4}, 7 \times 10^{-4}\}$ to examine the effect of step size on conservation law adherence. A linear learning rate scheduler is applied throughout training.

**ImageNet-1K.** For ImageNet-1k, we use a Vision Transformer with 12 Transformer layers, a hidden size of 192, and an intermediate dimension of 768. The model uses 4 attention heads, and images are tokenized using a patch size of 16. The model is trained with SGD using a batch size of 1024 and zero weight decay. We train for 5 epochs with learning rates of $\{1 \times 10^{-3}, 2 \times 10^{-3}, 3 \times 10^{-3}\}$ to examine the effect of step size on conservation law adherence. A linear learning rate scheduler is applied throughout training.

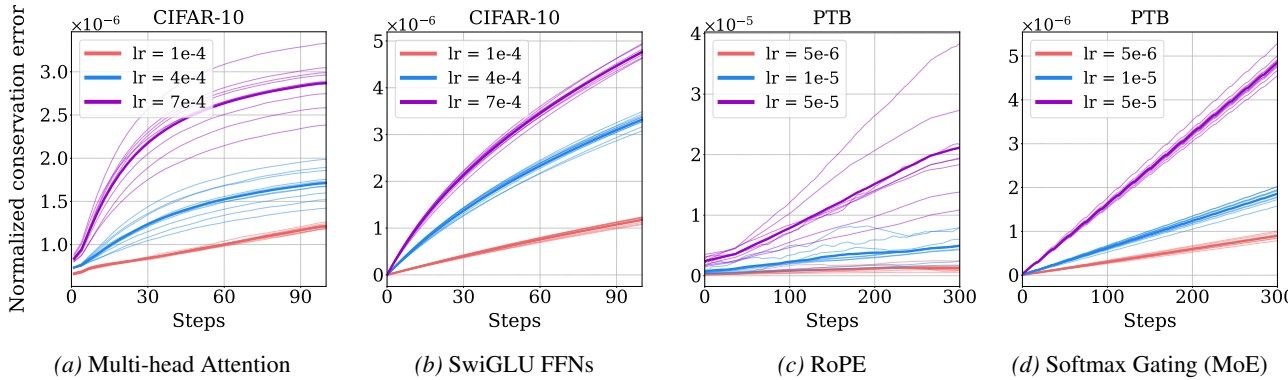

*Figure 3.* Conservation error tracking during training. (a-b) Per-step conservation metrics for multi-head attention and SwiGLU FFNs on CIFAR. (c-d) Average conservation errors for RoPE attention blocks and MoE softmax gating on PTB, computed as mean relative L2 deviations from initialization across all tracked quantities.

**Penn Treebank.** For the Penn Treebank language modeling task, we adopt a Transformer architecture with 12 layers, a hidden size of 192, and an intermediate size of 768. The model employs 3 attention heads and a mixture-of-experts (MoE) configuration with 4 experts. The maximum sequence length is set to 256 tokens. Training is conducted using SGD with a batch size of 192, gradient accumulation steps of 18, and zero weight decay. We train for 300 epochs with learning rates of $5 \times 10^{-6}$, $1 \times 10^{-5}$, and $5 \times 10^{-5}$ to examine the effect of step size on conservation law adherence. A linear learning rate scheduler is applied throughout training.

**WikiText103.** For WikiText103, we use a Transformer-based language model with 12 layers, hidden size 192, and intermediate size 768. The attention module consists of 3 heads, and the model is configured with 4 experts in the MoE layers. The maximum sequence length is 256 tokens. Optimization is performed using SGD with a batch size of 48 and no weight decay. We train for 15,000 steps with learning rates of $\{2 \times 10^{-6}, 5 \times 10^{-6}, 1 \times 10^{-5}\}$ to examine the effect of step size on conservation law adherence. A linear learning rate scheduler is applied throughout training.

**Runtime Environment.** All experiments are implemented in PyTorch 2.9.1 with CUDA 12.8 and conducted on a single NVIDIA H100 GPU equipped with 80GB of high-bandwidth memory. We utilize 12 parallel data loading workers to ensure efficient data pipeline throughput and minimize I/O bottlenecks during training. Across all experimental configurations, peak GPU memory consumption remains below 70GB, leaving sufficient headroom for system stability. Training time varies by dataset and model size but does not exceed 4 hours per individual configuration, enabling rapid iteration and extensive hyperparameter exploration.

## E. Experimental Results

### E.1. Normalized Sigmoid Gating

We extend our analysis to MoE architectures with normalized sigmoid gating, defined as $\text{sigmoid}(Wx)/\|\text{sigmoid}(Wx)\|_1$, to validate conservation law generality beyond softmax mechanisms. We evaluate on Penn Treebank (PTB) with full-batch gradient descent and WikiText-103 with mini-batch SGD using learning rates $\{0.01, 0.02, 0.05\}$.

Figure 4 confirms that normalized sigmoid gating exhibits the same bounded linear behavior as softmax gating. Conservation errors maintain $O(\tau^2 k)$ scaling on both datasets, with error magnitude increasing proportionally with learning rate. This consistency across gating mechanisms validates that conservation laws arise from optimization geometry rather than specific activation functions.

### E.2. Non-conservation Quantities

To provide qualitative context for our conservation results, we also track representative non-conserved quantities within each architectural component. Unlike conservation laws, which exhibit bounded $O(\tau^2 k)$ evolution, non-conserved quantities can experience arbitrarily large deviations during training–small parameter changes may induce unbounded growth without theoretical guarantees. Direct quantitative comparison between conserved and non-conserved quantities is inherently

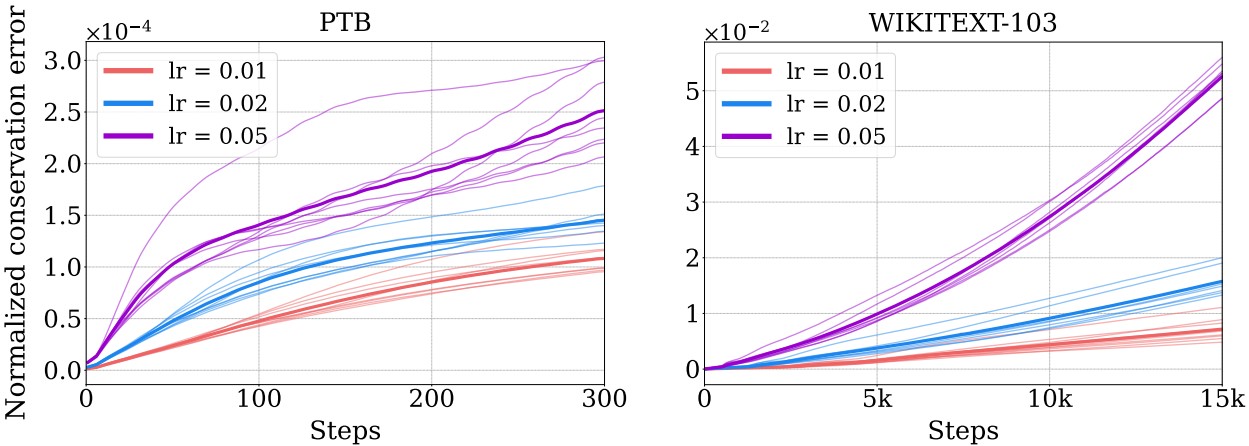

*Figure 4.* Normalized sigmoid gating conservation errors on Penn Treebank (left, full-batch) and WikiText-103 (right, mini-batch SGD). Conservation errors exhibit linear $O(\tau^2 k)$ scaling with learning rate-dependent bounds. Thin lines: individual layers; thick lines: averages.

problematic: non-conserved quantities lack theoretical bounds on their evolution, and any bounded linear combination of conservation laws would itself be conserved. Nevertheless, tracking these quantities alongside conservation laws illustrates the fundamental distinction between the two classes and demonstrates the special structure of conserved quantities.

For each architectural component, we monitor the following non-conserved quantities:

- **Multi-head Attention.** For each head $i \in [h]$, we track the sums $Q_i^\top Q_i + K_i^\top K_i$, $\quad V_i^\top V_i + O_i^\top O_i$, which differ from the conserved differences $Q_i^\top Q_i - K_i^\top K_i$ and $V_i^\top V_i - O_i^\top O_i$.

- **Multi-head Attention with RoPE.** For each head $i \in [h]$ and block $j \in [m]$, where $Q_i^{(j)}, K_i^{(j)} \in \mathbb{R}^{d \times 2}$ denote the $(j)$-th blocks of projection matrices, we track $\|Q_{i\ :,0}^{(j)}\|_2^2 + \|K_{i\ :,0}^{(j)}\|_2^2 - \|Q_{i\ :,1}^{(j)}\|_2^2 - \|K_{i\ :,1}^{(j)}\|_2^2$, where $Q_{i\ :,\ell}^{(j)}$ denotes the $\ell$-th column. This contrasts with the conserved block-wise Frobenius norm differences.

- **SwiGLU FFN.** For each pair of indices $(i, j)$, we track element-wise differences $A_{ij}^2 - C_{ij}^2$, which lack the column-wise or row-wise structure that produces conservation in $\|A_{:,i}\|^2 - \|C_{i,:}\|^2$.

- **MoE Gating.** We track the first row $W_{0,:}$ (gating of first experts) in contrast to the conserved row sums $\sum_{j=1}^{d} W_{:,j}$ (sum over dimensions for each expert).

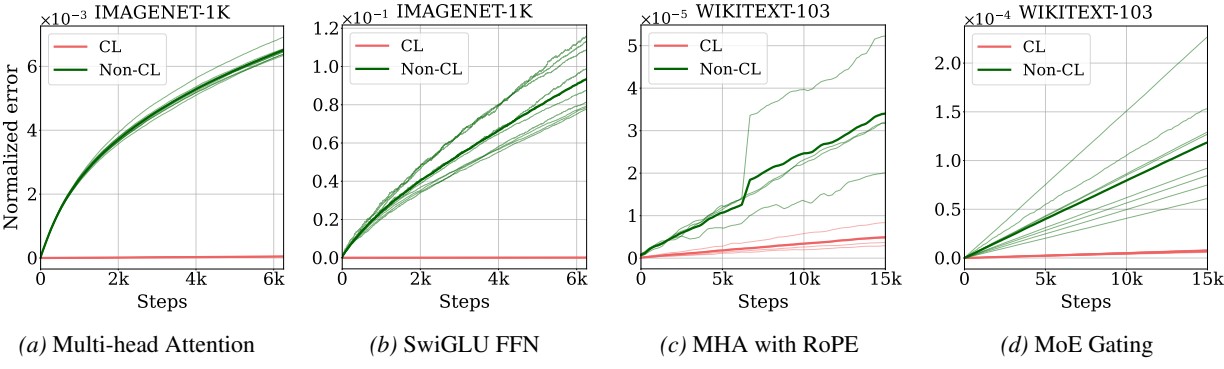

*(a)* Multi-head Attention      *(b)* SwiGLU FFN      *(c)* MHA with RoPE      *(d)* MoE Gating

*Figure 5.* Normalized errors of conserved (CL) and non-conserved quantities (Non-CL) on ImageNet-1K and Wikitext-103.

**Results.** Figure 5 reveals a stark contrast between conserved and non-conserved quantities. Conservation laws (red lines) remain tightly bounded near zero throughout training, while non-conserved quantities (green lines) exhibit larger deviations.

This substantial disparity demonstrates that conservation is a fundamental structural property arising from the optimization geometry, not merely slow parameter drift.

