# OpenReview forum: "Conservation Laws for Modern Neural Architectures"
_ICML.cc/2026/Conference — ICML 2026 spotlight_

### Official Review · Reviewer_21Wg · 2026-02-21

**Soundness:** 4
**Presentation:** 3
**Significance:** 4
**Originality:** 4
**Overall Recommendation:** 6
**Confidence:** 2

**Summary:**

This paper focuses on the study of conservation laws in the gradient flow of neural networks. To address the core issue that existing mainstream deep learning architectures lack a precise characterization of conservation laws, it develops a unified theoretical analysis framework. The framework fully characterizes the conservation laws for feed-forward networks with GELU, SiLU, and SwiGLU activations, multi-head attention with sinusoidal / rotary position encoding, and Mixture-of-Experts architectures covering dense / sparse / various gating designs. It resolves an open problem in the literature regarding the complete characterization of conservation laws for multi-head attention.

**Compliance With Llm Reviewing Policy:**

Affirmed.

**Final Justification:**

Thank you for your response. Most of my comments have been addressed. Considering the overall quality of the paper, I'll keep my original score.

**Key Questions For Authors:**

see weakness. I would say this paper is really interesting.

**Limitations:**

yes

**Strengths And Weaknesses:**

- To be honest, I am not able to fully understand all the details, as I am not an expert in this field. However, I must say that I am genuinely impressed by the conclusions presented in this paper. They are indeed very interesting.

- Line 164: c \in R^2 --> c \in R?

- Can a single architecture possess multiple conservation laws?

- What common features do all conservation laws share? Is there something like a “master conservation law” from which others can be derived, or that could even be used to generate certain network structures? In other words, is there a unified set of equations, similar to Maxwell’s equations, that governs these phenomena? If there were a one to one correspondence between structures and conservation laws, would it be possible to systematically generate architectures and then simply select among them? That said, this raises another question: what makes a conservation law good? For example, in large model settings, why does a SwiGLU based FFN often perform better than the original FFN?

- In the numerical validation, what are the main reasons behind the seemingly non conservative phenomena observed in some cases? In addition, I believe that more extensive and diverse numerical experiments would further strengthen the conclusions of the paper.

- Equation 5 does not include the common scaling factor 1 over square root of d. Would this omission affect the conclusions?

- Does the conservation law in parameters imply that there may also be certain conservation laws in the features of neural networks?
- I know it may be too early to discuss the practical applications of these findings, but I would also like to hear the authors' thoughts on the potential future applications of the discovered conservation laws.

---

> ### Author Rebuttal · Authors · 2026-03-31
>
> We thank the Reviewer for the feedback and address the concerns as follows.
>
> ---
>
> **W2.** This is a typo; it should indeed be $c \in \mathbb{R}$. We thank the Reviewer for pointing this out.
>
> **W3.** Based on our analysis, a single architecture can admit multiple conservation laws (CLs). For example, in Multihead Attention, the CLs are constant on each connected component of the level sets determined by the invariants $Q_i^\top Q_i - K_i^\top K_i$ and $V_i^\top V_i - O_i^\top O_i$ for $i \in [n]$.
>
> In particular, for each $i \in [n]$, the quantity $Q_i^\top Q_i - K_i^\top K_i$ itself defines a CL for MHA. More generally, any function of these invariants is also a CL for MHA.
>
> **W4.** The conservation laws (CLs)—especially the data-independent ones studied in our submission—are intrinsic to the architecture. It is well known that in parameterizations involving a product $AB^\top$ with $A,B \in \mathbb{R}^{m \times n}$, the quantity $A^\top A - B^\top B$ is conserved under gradient flow. However, our focus is not merely to identify examples of conserved quantities, but to characterize all CLs admitted by a given architecture—i.e., to establish an “if and only if” result.
>
> While CLs provide an interesting lens on training dynamics, we believe that the empirical performance of an architecture depends on many additional factors, including hardware efficiency, numerical stability, and compatibility with commonly used optimization methods.
>
> Moreover, since each architecture has distinct structural properties, it is unlikely that a single unified equation can capture conservation laws across all architectures. For example, SwiGLU-based FFNs often outperform standard FFNs, likely due to a combination of factors: they typically have more parameters at comparable widths while maintaining a favorable computation–memory trade-off. Together with strong empirical performance, this has led to their widespread adoption.
>
> **W5.** We track the evolution during training of several non-conserved quantities and compare them with the identified CLs. As expected, the variation in non-conserved quantities is significantly larger, while the CLs remain nearly constant. For example, in Figure 5, the CL error stays close to zero, whereas the error for non-CL quantities increases rapidly.
>
> **W6.** We omit the scalar factor $1/d_h$ (where $d_h$ is the dimension of each attention head) for notational simplicity. This does not affect the final conclusions.
>
> **W7.** This is a very interesting question from the Reviewer, and one we also considered during the preparation of this submission.
>
> Concretely, suppose a token (or feature) is represented as a vector in $\mathbb{R}^d$. In analogy with the parameter-space setting, one may define a “conservation law” as a function $h : \mathbb{R}^d \to \mathbb{R}$ that remains constant along the training dynamics of the token. We investigated this notion across several architectures; however, in all cases we examined, the only such conservation laws were constant functions. In other words, we did not observe any nontrivial CLs in token space.
>
> If nontrivial conservation laws of this kind were to exist, they could provide useful insights into token dynamics, potentially allowing certain behaviors to be identified early in training. Unfortunately, our analysis suggests that this is not the case under the current formulation.
>
> This indicates that alternative definitions or perspectives on conservation laws—tailored specifically to token or feature dynamics—may be necessary. We believe this is a promising direction for future work.
>
> **W8.**  Implicit bias in machine learning refers to the phenomenon that an optimization method (e.g., gradient descent or Adam) does not converge to an arbitrary solution, but instead favors certain types of solutions—even without explicit regularization. This bias depends on several factors, including the optimization algorithm, initialization, parameterization, and architecture.
>
> CLs provide a useful lens for understanding implicit bias, as they reveal structural properties of the solutions selected by the optimizer that are already determined at initialization. For instance, in MHA, if we track the norms of the $Q$ and $K$ matrices within each attention head during training, we observe that they tend to match in magnitude at convergence (e.g., in one head, norms of 510.2 and 508.5; in another, 24.9 and 25.4). This behavior is consistent with the conservation structure.
>
> That said, this direction remains in its early stages, and many open questions remain. In particular, most existing analyses of CLs rely on gradient flow, which is rarely used in practice. Modern optimizers such as Adam induce more complex dynamics, and each may lead to different conservation structures. Extending CL analysis to these settings is significantly more challenging due to the complexity of the optimization dynamics, but represents an important direction for future work.

---

> > ### Author Rebuttal · Reviewer_21Wg · 2026-04-02
> >
> > Thank you for your response. Most of my comments have been addressed. Considering the overall quality of the paper, I'll keep my original score.

---

### Official Review · Reviewer_FS8d · 2026-03-09

**Soundness:** 3
**Presentation:** 4
**Significance:** 2
**Originality:** 3
**Overall Recommendation:** 4
**Confidence:** 4

**Summary:**

This paper shows that the conservation laws in the gradient flow training of neural networks must held constant on the level sets of their respective characteristic invariants, and identifies such characteristic invariants for modern neural network architectures, including those with GELU, SiLU, and SwiGLU activations, MHA and MoE.

**Compliance With Llm Reviewing Policy:**

Affirmed.

**Final Justification:**

The authors have addressed my technical questions. Regarding the limitations, author discusses some implications of the conservation law, most of which have appeared in earlier works from for example Marcotte et al. Thus, I stand my point that there is solid technical contributions, but not much new high-level insights.

**Key Questions For Authors:**

1. How should we view these results? On the one hand, new conservation laws are discovered for modern architectures, which is good; On the other hand, the invariance identified there is not much different than those in "old" architectures, the conserved quantities are either a matrix of the form $AA^\top-B^\top B$, or the diagonal of it. They are as weak as the old ones: the dimension of the invariant manifolds is way smaller than the ambient dimension, thus there is not much gain from knowing this invariance, if the parameters are in billions. I think the writings should discuss such limitations.
2. About the exact correspondence: If one knows the characteristic invariants, how do you determine the conservation law $h$? All theorems only show the other direction.

**Limitations:**

See questions.

**Strengths And Weaknesses:**

Strength:
1. The paper is well written. The writing and the discussion clearly distinguish this work from those by Marcotte et al., where the latter identifies conservation laws using a Lie-theoretic framework.
2. The conservation laws are identified for several modern architectures, making the analysis more practical relevant.

Weaknesses:
1. The identified conservation laws are generally weak in the sense that it only defines invariant manifolds of dimensions slightly smaller than the ambient dimension in the parameter space.
2. The exact correspondence between the conservation laws and their characteristic invariants is not clear (See question).

---

> ### Author Rebuttal · Authors · 2026-03-31
>
> Since the weaknesses are reflected in the questions, we focus on addressing them directly. We group the concerns into two parts and answer each.
>
> ---
>
> **Part 1. About the exact correspondence: ... All theorems only show the other direction.**
>
> **The dimension of the invariant manifolds is way smaller ... the writings should discuss such limitations.**
>
> **Answer.** We believe the Reviewer’s confusion stems from our use of characteristic invariance (i.e., connected components of level sets) to describe conservation laws (CLs). We emphasize that this is not merely a stylistic choice, but the most general and mathematically precise formulation.
>
> To clarify, consider CLs for Multihead Attention (MA). For simplicity, we focus on a single head. The result states that a function $h$ is a CL if and only if it is constant on each connected component of the level sets defined by the invariant $Q^\top Q - K^\top K$, for $Q, K \in \mathbb{R}^{d \times d_h}$.
>
> Concretely, this means the following: If $(Q,K)$ and $(\bar Q, \bar K)$ can be connected by a continuous path $(Q(a),K(a)), a \in [0,1]$ in the parameter space, i.e.
>
> $(Q,K) = (Q(0),K(0))$ and $(\bar Q, \bar K) = (Q(1),K(1)),$
>
> such that $Q(a)^\top Q(a) - K(a)^\top K(a)$ remains constant along this path, i.e.
>
> $Q^\top Q - K^\top K = \bar{Q}^\top \bar{Q} - \bar{K}^\top \bar{K} = Q(a)^\top Q(a) - K(a)^\top K(a)$ for all $a \in [0,1]$,
>
> then $h(Q,K) = h(Q',K')$.
>
> This characterization is both:
>
> - **Necessary:** every CL must be constant along such invariant-preserving paths;
> - **Sufficient:** any function satisfying this property is a CL.
>
> Hence, this invariance property provides a **necessary and sufficient condition** for CLs.
>
> **Why not write $h(Q,K) = g(Q^\top Q - K^\top K)$?**
>
> A natural question is whether this condition can be simplified by expressing $h$ as a function of the invariant, i.e., $h(Q,K) = g(Q^\top Q - K^\top K)$ for some function $g$.
>
> In general, this is not valid.
>
> The issue is that the level sets $\{(Q,K) : Q^\top Q - K^\top K = C\}$ are, in general, not connected. Therefore, two points may share the same invariant value but lie in different connected components, and a conservation law is only required to be constant within each connected component, not across different ones.
>
> **Illustrative example**
>
> This point is already carefully explained in our paper (Section 3). In particular, we provide a simple example where a CL $h : \mathbb{R}^2 \to \mathbb{R}$ is constant on the level sets  $\{a^2 - b^2 = c\}$ for $c \in \mathbb{R}$. However, one cannot conclude that there exists a function $g : \mathbb{R} \to \mathbb{R}$ such that $h(a,b) = g(a^2-b^2)$.
>
> The reason is that, for some values of $c$, the level set $\{a^2 - b^2 = c\}$ for $c \in \mathbb{R}$ is disconnected. For example, when $c = 1$, the constraint $a^2 = 1+b^2$ implies $a \ge 1$ or $a \le -1$, so the level set consists of two disjoint components.
>
> In particular, although $1^2-0^2 = (-1)^2 - 0^2$ there is no continuous path in $\mathbb{R}^2$ connecting $(1,0)$ and $(-1,0)$ along which $a^2-b^2$ remains constant. Consequently, a CL may take different values at these two points. We also provide a visualization in Figure 1 of the paper.
>
> More generally, describing functions via invariance on connected components of level sets is standard in invariance theory. It captures the exact geometry induced by the invariants and avoids issues from disconnected level sets, making it the most precise characterization of CLs.
>
> **Part 2. How should we view these results? ... the form $AA^T-B^TB$, or the diagonal of it.**
>
> **Answer.** The main and most significant contribution of our work is a framework that characterizes all CLs for a given architecture.
> - Verifying that a specific function $h$ is a CL is relatively straightforward: one checks that $\frac{d}{dt} h(\theta(t)) = 0$ for all initializations $\theta(0)$ and all $t > 0$ along gradient flow.
> - In contrast, characterizing all such functions $h$ is highly nontrivial—this is precisely our focus. We provide a complete description for the architectures considered.
>
> To the best of our knowledge, only a few prior works (notably Marcotte et al.) address this converse direction.
>
> Moreover, as noted in [1], the case of MA remains an open problem. While it is relatively straightforward to verify that quantities such as $Q_i^\top Q_i - K_i^\top K_i$ and $V_i^\top V_i - O_i^\top O_i$ are conserved along gradient flow, it is substantially more difficult to establish that these invariants exhaust all CLs.
>
> In addition, there is currently no prior work that characterizes CLs for Mixture-of-Experts.
>
> [1] Sibylle Marcotte et al., Transformative or Conservative? Conservation laws for ResNets and Transformers
>
> ---
> We thank the Reviewer for the feedback. If our responses address the concerns, we kindly hope the evaluation may be updated accordingly, and we remain open to further discussion.

---

> > ### Author Rebuttal · Reviewer_FS8d · 2026-03-31
> >
> > I appreciate the technical discussions. I would like the authors to kindly share their thoughts on "They are as weak as the old ones: the dimension of the invariant manifolds is way smaller than the ambient dimension, thus there is not much gain from knowing this invariance, if the parameters are in billions" from a more practical perspective.

---

> > > ### Author Response · Authors · 2026-04-01
> > >
> > > >They are as weak as the old ones: the dimension of the invariant manifolds is way smaller than the ambient dimension, thus there is not much gain from knowing this invariance, if the parameters are in billions
> > >
> > > We address the concern step by step as follows.
> > >
> > > **Layer-level vs. model-level conservation laws.**
> > >
> > > First, we emphasize that conservation laws (CLs) are inherently component-wise and are preserved under composition. Specifically, consider a model given by a composition $f = a \circ b \circ c$. If $h$ is a CL associated with any individual component (e.g., $a$, $b$, or $c$), then along gradient flow $\theta(t)$ of the full model, one can verify via a direct application of the chain rule that $\frac{d}{dt} h(\theta(t)) = 0$. Hence, CLs identified at the level of individual components remain valid for the full network.
> > >
> > > As a consequence, the full model inherits CLs from each of its constituent layers. Therefore, *the total set of CLs scales with the number of layers*, not as a single global object independent of architecture.
> > >
> > > **On dimensional comparison.**
> > >
> > > Given this structure, comparing layer-level CLs to the total parameter count of the entire model is not meaningful, as CLs act locally within each layer.
> > >
> > > A more appropriate comparison is *layer-wise*: the parameter dimension of a layer should be compared with the dimension of its associated CLs. From this perspective, these constraints are nontrivial and structurally significant.
> > >
> > > **An example: Multihead Attention.**
> > >
> > > Consider the Multihead Attention (MHA) module. It has four weight matrices $Q, K, V, O \in \mathbb{R}^{d \times d}$, which are partitioned into $n$ heads. Each head has parameters $Q_i,K_i,V_i,O_i\in\mathbb{R}^{d \times d_h}$ where $d_h = d/n$.
> > >
> > > The $2n$ invariants identified in Theorem 4.3 are $Q_i^\top Q_i - K_i^\top K_i$ and $V_i^\top V_i - O_i^\top O_i$ for $i\in[n]$, each of which is a matrix in $\mathbb{R}^{d_h \times d_h}$.
> > >
> > > From a dimensional perspective, each projection matrix has size $d \times d_h$, while each invariant lies in $\mathbb{R}^{d_h \times d_h}$. Thus, at the level of a single head, CLs impose structured constraints of smaller but comparable dimension.
> > >
> > > These CLs have direct implications for training dynamics, particularly through implicit bias, as we discuss next.
> > >
> > > **What do these conservation laws tell us about training dynamics?**
> > >
> > > A key consequence of CLs is their connection to *implicit bias*. Implicit bias refers to the tendency of optimization methods (e.g., gradient descent or Adam) to favor specific solutions, even in the absence of explicit regularization, depending on factors such as initialization, parameterization, and architecture.
> > >
> > > CLs provide a principled explanation of this phenomenon: they impose structural constraints on training trajectories that are fixed at initialization and preserved throughout optimization.
> > >
> > > Returning to the MHA setting, consider the invariant $Q_i^\top Q_i - K_i^\top K_i = C_i$, where $C_i$ is fixed by initialization. Taking diagonal entries yields, for each column index $j \in [d_h]$, we have $\|q_{i,j}\|^2 - \|k_{i,j}\|^2 = (C_i)_{jj}.$
> > >
> > > Under standard initialization schemes such as Xavier initialization, the entries of $Q_i$ and $K_i$ are drawn i.i.d. from the same zero-mean distribution with identical variance. Consequently, $\mathbb{E}[Q_i^\top Q_i] = \mathbb{E}[K_i^\top K_i]$, which implies $\mathbb{E}[C_i] = 0$. Moreover, by concentration of measure, each entry of $C_i$ is small with high probability, so that $C_i \approx 0$ at initialization.
> > >
> > > The CL then enforces that this relation is preserved along training: $Q_i^\top Q_i - K_i^\top K_i = C_i \approx 0$ for all $t$, then $\|q_{i,j}(t)\|^2 \approx \|k_{i,j}(t)\|^2$ for all $t$, i.e., the norms remain close up to a fixed offset determined at initialization.
> > >
> > > **Empirical evidence.**
> > >
> > > This behavior is also observed in trained models. Tracking the norms of $Q_i$ and $K_i$ within each attention head during training shows that they tend to closely match at convergence. Typical observations include
> > >
> > > $\|Q_i\| \approx 510.2, \quad \|K_i\| \approx 508.5$, and in another head, $\|Q_i\| \approx 24.9, \quad \|K_i\| \approx 25.4$.
> > >
> > > Such near-matching follows from the CL: since the difference in squared norms is fixed (and typically small), optimization favors solutions where $Q_i$ and $K_i$ are metrically aligned. Consequently, although $(Q_i, K_i)$ and $(c Q_i, c^{-1} K_i)$ are functionally equivalent for any $c>0$, the constraint that $|Q_i|$ and $|K_i|$ remain close prevents large $c$.
> > >
> > > Thus, *while functionally equivalent minima exist in the loss landscape, the optimizer is constrained to a subset of them and may not reach others.*
> > >
> > > ---
> > >
> > > We thank the Reviewer for the feedback. We hope our responses have addressed the concerns. If so, we kindly request that the evaluation be updated accordingly. In accordance with the policy, this is our final response.
> > >
> > > Best regards,
> > >
> > > Authors

---

### Official Review · Reviewer_r3uS · 2026-03-13

**Soundness:** 3
**Presentation:** 3
**Significance:** 3
**Originality:** 3
**Overall Recommendation:** 4
**Confidence:** 2

**Summary:**

This paper studies conservation laws under gradient flow for several modern neural architecture modules, including GELU/SiLU/SwiGLU feedforward networks, multi-head attention with sinusoidal or rotary positional encoding, and mixture-of-experts.

**Compliance With Llm Reviewing Policy:**

Affirmed.

**Final Justification:**

The rebuttal solved my problem, and I'm inclined to accept.

**Key Questions For Authors:**

1. Can these conservation laws translate into stronger conclusions about training behavior?

2. Under discrete settings, can the author provide an upper bound on the changes in the conservation laws?

**Limitations:**

The authors discuss the paper's main limitations, including the limited applicability of their method to more complex singular structures and the fact that the theoretical analysis remains largely confined to the continuous-time setting.

**Strengths And Weaknesses:**

Strengths

1. The paper covers a relatively broad range of modern architectures.

2. The results aim at a complete C1 characterization

Weaknesses

1. Most of the nontrivial conservation laws still take the form of norm differences or Gram-matrix differences.

2. The paper mainly establishes the existence of these conservation structures but does not go much further in explaining how they translate into a stronger understanding of modern architecture.

3. The experiments mainly verify that the predicted invariants are approximately preserved during training, but they do not clearly demonstrate how these conservation laws provide practical theoretical insights into modern neural network behavior.

---

> ### Author Rebuttal · Authors · 2026-03-31
>
> We thank the Reviewer for the response and address the concerns below. For clarity and coherence, we found it appropriate to merge certain related weaknesses and questions and provide unified answers.
>
> ---
>
> **W1.** Conservation laws in our work are intrinsic to the model architecture: they are determined entirely by how the layers and mechanisms are constructed, and cannot be introduced or altered post hoc. In particular, they are not design choices but structural properties that are already present in the architecture.
>
> Our main contribution is to provide a complete characterization of these conservation laws. Specifically, for each architecture we study, we identify a condition X such that a function $h$ is a conservation law if and only if it satisfies X. That is:
>
> - **Necessity:** Every conservation law must satisfy condition X.
> - **Sufficiency:** Any function satisfying X is a conservation law.
>
> Therefore, the conditions we derive are both necessary and sufficient, and collectively describe all conservation laws of the given architecture.
>
> **W2+W3+Q1.** One important aspect of conservation laws (CLs) is their connection to implicit bias. Implicit bias refers to the phenomenon whereby an optimization method (e.g., gradient descent or Adam) does not converge to an arbitrary solution, but instead favors particular solutions—even in the absence of explicit regularization. This bias depends on several factors, including the optimization algorithm, initialization, parameterization, and architecture.
>
> CLs provide a useful lens for understanding implicit bias, as they reveal structural constraints on the solutions selected by the optimizer that are already fixed at initialization. For instance, in multi-head attention (MHA), tracking the norms of the $Q_i$ and $K_i$ matrices within each head during training shows that they tend to match in magnitude at convergence (e.g., 510.2 vs. 508.5 in one head, 24.9 vs. 25.4 in another), consistent with the predicted conservation structure.
>
> CLs also impose strict coupling between parameter matrices, constraining the entire training trajectory rather than only its asymptotic outcome [2]. This coupling rules out stable stagnation near small initialization and guarantees finite-time escape from the small-scale regime (Theorem 1). It also forces all diverging parameter directions to align, making condensation a necessary consequence of the dynamics rather than an incidental phenomenon (Theorem 2). Moreover, the conserved quantities provide two-sided control of energy growth, enabling us to characterize not only whether condensation occurs, but also when it completes. Together, these results yield stronger, phase-level insights into training behavior.
>
> **The application of CLs** remains at an early stage, with many open questions. Most existing CL analyses rely on gradient flow, which is rarely used in practice. Modern optimizers such as Adam induce more complex dynamics, and may lead to different conservation structures. Extending CL analysis to such settings is significantly more challenging, but represents an important direction for future work.
>
> **Our experiments** aim to empirically validate the theoretical conservation laws (CLs), consistent with the experimental protocol in [1]. We track the evolution of both conserved and non-conserved quantities during training. As expected, non-conserved quantities exhibit substantial variation, while the identified CLs remain nearly constant. For example, in Figure 5, the CL error stays close to zero, whereas the error for non-CL quantities increases rapidly.
>
> **Q2.** An upper bound on the variation of the conservation laws is provided at the beginning of Section 6, following the approach in [1]. Specifically, this bound grows linearly with the number of iterations when a constant step size is used, while it remains uniformly bounded under a decaying step size of the form $\tau_k = \tau_0/(k+1)$.
>
> ---
>
> **Reference**
>
> [1] Sibylle Marcotte et al., Transformative or Conservative? Conservation laws for ResNets and Transformers
>
> [2] Chen, Z.-A. & Luo, T. (2025). *From condensation to rank collapse: A two-stage analysis of transformer training dynamics*. arXiv:2510.06954.
>
> ---
>
> We thank the Reviewer for the constructive feedback and thoughtful suggestions. If our responses adequately address the concerns, we kindly hope that the evaluation may be adjusted to reflect this. We remain open to further discussion during the next stage of discussion.

---

> > ### Author Rebuttal · Reviewer_r3uS · 2026-04-03
> >
> > Thank you for the clarifications. The rebuttal adequately addressed my main questions.

---

> > > ### Author Response · Authors · 2026-04-05
> > >
> > > We are pleased that our responses have addressed the concerns you raised. As this is our final reply in accordance with the venue policy, we kindly hope that, if no further concerns remain, the evaluation can be adjusted to reflect this.
> > >
> > > We once again thank the Reviewer for the constructive feedback, thoughtful suggestions, and their endorsement.
> > >
> > > Best regards,
> > >
> > > Authors

---

### Official Review · Reviewer_r78Q · 2026-03-13

**Soundness:** 3
**Presentation:** 2
**Significance:** 3
**Originality:** 3
**Overall Recommendation:** 5
**Confidence:** 4

**Summary:**

The preprint derives conservation laws for neural network architecture blocks for which they were previously not known: GeLU, SiLU, and SwiGLU activation functions; multi-head attention; and Mixture-of-Experts models. To do so, the proofs use the standard characterization of conserved quantities, which leads to solving a system of partial differential equations. Finally, the submission experimentally confirms the conservation.

**Compliance With Llm Reviewing Policy:**

Affirmed.

**Final Justification:**

The work is original in deriving the complete characterization of conservation laws for architectural blocks for which they were previously unknown. The writing is overall clear and easy to follow, and the proofs looked convincing to me, though I did not read them in detail. Initially, I had concerns about the work's positioning in the related work and several technical questions. The authors provided insightful, detailed responses to all my questions, and I came to appreciate the problem's complexity more. Hence, I raised my score to Accept.

**Key Questions For Authors:**

**Crucial** (Questions affecting the score)

Q1. Could the preprint expand on the references related to the conservation laws to better position it within the related work? Currently, previous works on the characterization of conservation laws only mention Marcotte et al. (2023) and Marcotte et al. (2025). Several suggestions:

- Kunin et al. (2021) propose a unifying framework for conservation laws for any dataset;

- Abbe et al. (2023) utilize conservation laws for transformers with diagonal weights;

- Zhang et al. (2025) consider conservation laws in linear attention.

Q2. Could the submission provide more details on which aspects of the proposed approach make it a distinct framework rather than an application of dynamical systems methods? How does it differ from related work, especially Kunin et al. (2021)?

**Minor** (Questions to confirm my understanding and minor errors)

Q3. Based on the results, does a linear combination of architectural blocks, such as multi-head attention or a Mixture-of-Experts, result in a set of conservation laws for each component of the combination? Or is behavior more complicated?

Q4. Is there any limitation that prevents considering simple deep networks, where, for instance, GeLU layers are stacked one after another without any additional techniques such as normalization?

Q5. The title at the top of the submission has not been changed from the template and still shows ‘Submission and Formatting Instructions for ICML 2026’.

*References*

(Kunin et al., 2021) Kunin, D., Sagastuy-Brena, J., Ganguli, S., Yamins, D. L., and Tanaka, H. (2021). Neural mechanics: Symmetry and broken conservation laws in deep learning dynamics. In International Conference on Learning Representations.

(Abbe et al., 2023) Abbe, E., Bengio, S., Boix-Adsera, E., Littwin, E., and Susskind, J. M. (2023). Transformers learn through gradual rank increase. In Thirty-seventh Conference on Neural Information Processing Systems.

(Zhang et al., 2025) Zhang, Y., Singh, A. K., Latham, P. E., and Saxe, A. M. (2025). Training dynamics of in-context learning in linear attention. In Forty-second International Conference on Machine Learning.

**Limitations:**

yes

**Strengths And Weaknesses:**

**Strengths**

S1. The work is original in deriving conservation laws for architectural blocks for which they were previously unknown. While conservation laws for versions of attention have appeared in other works, the rest of the results is new. The results are significant for the neural network training dynamics analysis, for which the conservation laws have been used before.

S2. The writing is overall clear and easy to follow, and the proofs appear convincing. However, please note that I have not examined the proofs in detail.

**Weaknesses**

W1. The presentation could be improved by providing a broader review of prior work on conservation laws. More details are in Key Questions For Authors.

W2. Presenting the results as a new framework is confusing. As I understand it, the results are based on the standard approach from the dynamical systems theory. Additionally, a similar approach, though for simpler architectures, has also appeared before in Kunin et al. (2021).

---

> ### Author Rebuttal · Authors · 2026-03-31
>
> We thank the Reviewer for the response and address the concerns below. For clarity and coherence, we found it appropriate to merge certain related weaknesses and questions and provide unified answers, as follows:
>
> - **Part 1 (W1+W2+Q1+Q2).** The distinction between our framework for conservation laws (CLs) and prior work, including the cited references, and the significance of our contribution.
>
> - **Part 2 (Q3+Q4).** The behavior of conservation laws derived at the component level when extended to the full model.
>
> - **Writing and presentation.** We thank the Reviewer for identifying issues in writing and formatting, and will revise the manuscript accordingly.
>
> ---
>
> **Answer Part 1.** Conservation laws (CLs) are defined with respect to a dynamical system—here, the training dynamics. This is the notion considered both in the references cited by the Reviewer and in our work. However, our focus is fundamentally different.
>
> - The referenced works study the same notion of CLs, but primarily identify and verify specific conserved quantities during training.
> - In contrast, our work—along with Marcotte et al. (2025)—addresses the converse problem of characterizing all CLs. Our goal is **to develop a theoretical framework that fully determines all conserved quantities of a given architecture**. While verifying that a function $h$ is a CL is straightforward (by checking that $\frac{d}{dt} h(\theta(t)) = 0$ along gradient flow), characterizing all such functions $h$ is highly nontrivial. This is precisely our contribution.
>
> For example, in Kunin et al. (2021), CLs are derived from architectural symmetries such as translation, scaling, and rescaling. However, such approaches do not establish whether the resulting CLs are exhaustive.
>
> Our framework directly addresses this gap by aiming to characterize *all* possible CLs. To the best of our knowledge, only a few prior works (notably Marcotte et al.) consider this direction. In particular, as noted in [1], the case of Multihead Attention remains open: while it is straightforward to verify that quantities such as $Q_i^\top Q_i - K_i^\top K_i$ and $V_i^\top V_i - O_i^\top O_i$ are conserved, proving that these exhaust all CLs is significantly more challenging. Moreover, there is currently no prior work characterizing CLs for Mixture-of-Experts.
>
> We believe this distinction underscores the novelty and significance of our proposed framework. We sincerely thank the Reviewer for suggesting the three references and will incorporate them into the related work section of the manuscript.
>
> ---
>
> **Answer Part 2.** The behavior of CLs from individual components in the full model is as follows:
>
> - **Inheritance.** Any CL associated with a component remains a CL of the composed network. For a composition $a \circ b \circ c$, if $h$ is a CL for $a,b$ or $c$, then along gradient flow it satisfies $\frac{d}{dt} h(\theta(t)) = 0$, which follows by a direct algebraic computation.
>
> - **Emergent cross-layer CLs.** However, stacking some components can create non-trivial conservation laws across layer.  While any CL identified for $a,b$ or $c$ remains a CL for the full model, the converse is not necessarily true. The composed architecture may admit additional conservation laws that cannot be decomposed into CLs of the individual components. In particular, such CLs may involve nontrivial coupling across parameters from multiple layers simultaneously.
>
>
> In summary, **CLs of individual components remain valid for the full model**. However, they may not be exhaustive. Our current framework operates at the level of individual components and does not yet capture emergent cross-layer conservation structures. To the best of our knowledge, no existing framework provides a complete characterization of all CLs for fully stacked architectures.
>
> We hope this clarifies the concern.
>
> ---
>
> **Reference.**
>
> [1] Sibylle Marcotte et al., Transformative or Conservative? Conservation laws for ResNets and Transformers
>
> ---
>
> We thank the Reviewer for the constructive feedback and thoughtful suggestions. If our responses adequately address the concerns, we kindly hope that the evaluation may be adjusted to reflect this. We remain open to further discussion during the next stage of discussion.

---

> > ### Author Rebuttal · Reviewer_r78Q · 2026-04-04
> >
> > I thank the authors for the insightful and detailed response. All my questions have been answered, and the contribution of the paper is much clearer to me now. I also appreciate its significance more. I would propose mentioning in the Abstract that the submission provides a complete characterization of the conservation laws. I will raise my score to Accept.

---

> > > ### Author Response · Authors · 2026-04-04
> > >
> > > We thank the Reviewer for their feedback and endorsement. We agree that explicitly stating this complete characterization in the abstract would better highlight our contribution, and we will revise accordingly.
> > >
> > > Best regards,
> > >
> > > Authors

---

### Official Review · Reviewer_bMZz · 2026-03-17

**Soundness:** 4
**Presentation:** 4
**Significance:** 2
**Originality:** 2
**Overall Recommendation:** 5
**Confidence:** 4

**Summary:**

The paper derives conservation laws for several components of model neural networks including 2-layer feedforward networks with various activations, multi-headed attention layers and mixture of experts. Importantly, the paper characterizes *all* possible conservation laws for these components. The conservation laws are derived by first expressing the set of all functions of network parameters that are orthogonal to possible gradients of the output with respect to parameters. This set depends on the input to the network: the next step is to remove this dependence which can be done for the network components studied by the paper. This yields a PDE on the conversation law which can be solved to reveal invariants of the parameters of network components. The authors then empirically demonstrate that the conversation laws approximately hold under SGD, with error growing over training time.

**Compliance With Llm Reviewing Policy:**

Affirmed.

**Final Justification:**

The rebuttal fully addressed my concerns. Thus, I am in favor of acceptance.

**Key Questions For Authors:**

1. What is the practical significance of conversation laws in general?
2. What is the practical significance of the conversation laws derived in this paper?
3. Are there any straightforward ways to compose conversation laws for combinations of network components (e.g. 2 FF layers)?

**Limitations:**

Yes, limitations are adequately discussed.

**Strengths And Weaknesses:**

**Soundness**

The theory presented in the paper is sound. All theoretical results are correct as far as I can tell and the assumptions are relatively mild. The experiments are also well-confucted, with experiments on both relatively-large scale vision and language tasks (ImageNet and WikiText). Overall, the paper has little to no technical flaws.

**Presentation**

The presentation is really the highlight of the paper: all components are well-explained and the paper is overall nicely structed. In particular, the use of the toy example in section 3 and setting aside an entire section (section 5) to overview the proof strategy set a great example for theory papers. The related work is also summarized at the appropriate level of depth.

Minor comments:
- The first equation in the toy example in Section 3 has an extra parenthesis
- It's unclear just from the caption of Figure 2 what the different lines for each learning rate mean (different trials? different hyperparameters?)
- It would be helpful to have a short note on why the orthogonality characterization of conversation laws is intuitively reasonable (lines 124-127)

**Significance**

The significance of the paper is unfortunatly a bit unclear. Of course, conversation laws are an interesting topic, but the practical significance is left to be filled in by the reader. This is true at two levels: first of all, the authors could further expand on how conversation laws in general are important: for instance, adding more discussion on the connection between conversation laws and implicit bias, or how conservation laws can yield better optimizers.

More importantly, the *specific* conversation laws found by the authors are not interpreted at all. For instance, the authors find that FFNs with GELU or SiLU have no non-trivial conversation laws, but those with SwiGLU do. How can we interpret this? Does this mean that SwiGLU layers are somehow less flexible, or that they carry greater implit bias? What does this mean for practioners? Perhaps the initialization of SwiGLU layers must be more carefully chosen then?

As another example, the positional encoding discussion after Theorem 4.3 implies that sinusoidal encodings don't affect the MHA conversation law while RoPE does. Are the resulting conversation laws weaker or stronger, and what does that imply about the flexibility of MHA with RoPE?

Another important question that is not adequately discussed is how to build conservation laws for compositions of the network components studied in the paper. The discussion at the end of Section 5 implies that stacking multiple layers breaks the proof methodology introducted in the paper. To what extent do the conversation laws hold for practical architectures then (from a theoretical perspective- the empirical question is addressed in Section 6)? Are there any layer compositions for which conservation laws can be straightforwardly derived from component-wise conservation laws (for instance, stacking two FF layers without normalization in between)?

**Originality**

At first glance, the paper seems to be fairly incremental: it introduces conservation laws for some network components not previously explored in the literature, but it doesn't make fundamentally new statements about conservation laws. Thus, the direct insights from the paper about conversation laws are somewhat limited. However, importantly, the paper introduces a fairly clean, easy-to-use methodology for characterizing conversation laws for many different neural network components. This methodology, even if overlapping with prior work, has not been fleshed out fully before. Thus, the theoretical novelty is non-trivial.

---

> ### Author Rebuttal · Authors · 2026-03-31
>
> We appreciate the Reviewer’s comments on typos and minor writing issues and will revise the manuscript accordingly. We address the remaining concerns below.
>
> ---
>
> **It's unclear just from the caption of Figure 2 ... (different trials? different hyperparameters?)**
>
> **Answer.** The multiple faint lines for each learning rate represent $10$ independent training runs initialized with different random seeds. The thicker line for each learning rate represents the mean of these 10 runs. We will update the caption of Figure 2 to make this explicit.
>
> **Q1+Q2 (...on the practical significance...)**
>
> **More importantly, ... more carefully chosen then?**
>
> **As another example, ... MHA with RoPE?**
>
> **Answers.** Some key consequences of CLs are as follows:
> - CLs provide a lens for understanding implicit bias, i.e., the tendency of optimization methods to favor specific solutions even without explicit regularization. Since CLs impose structural constraints fixed at initialization, they restrict the set of reachable solutions. For example, in multi-head attention (MHA), the norms of certain parameter matrices within each head tend to match at convergence, consistent with the predicted conservation structure.
> - CLs couple parameter matrices and constrain the entire training trajectory, not just its limit [2]. This prevents stagnation near small initialization, ensures finite-time escape from the small-scale regime (Theorem 1), and forces alignment of diverging directions, making condensation inevitable (Theorem 2). Conserved quantities also provide two-sided control of energy growth, yielding phase-level insight into when condensation occurs.
> - However, we do not believe any single theoretical notion—including CLs—fully explains the success of modern architectures. Empirical performance also depends on factors such as hardware efficiency, numerical stability, and optimization compatibility. For example, SwiGLU-based FFNs often outperform standard FFNs due to increased parameterization and favorable compute–memory trade-offs, contributing to their widespread adoption.
>
> At the time of this submission, our primary goal is to develop a theoretical framework for fully characterizing all CLs of a given architecture.
> - Verifying that a specific function $h$ is a CL is relatively straightforward: one checks that $\frac{d}{dt} h(\theta(t)) = 0$ for all initializations $\theta(0)$ and all $t > 0$ along gradient flow.
> - In contrast, characterizing all such functions $h$ is highly nontrivial—this is precisely our focus. We provide a complete description for the architectures considered.
>
> Only a limited number of prior works (notably those of Marcotte et al.) address this converse direction in any generality. In particular, cases such as multi-head attention have remained open, and are resolved in our work.
>
> For this reason, we focus primarily on the theoretical development and do not emphasize applications. However, we agree with the reviewers that discussing consequences and practical implications is *important and valuable*. We will revise the manuscript accordingly to better highlight these aspects.
>
> **Q3. Are there any straightforward ... network components?**
>
> **Answer.** Any CL associated with an individual component remains a CL of the full network. Indeed, for a composition $a \circ b \circ c$, one can directly verify that if $h$ is a CL for $a,b$ or $c$, then along gradient flow it satisfies $\frac{d}{dt} h(\theta(t)) = 0$, which follows by a direct algebraic computation. We will include this clarification in the revised manuscript.
>
> **Why stacking multiple layers breaks the proof methodology introduced in the paper.**  Consider a composition $a \circ b \circ c$. While any CL identified for $a,b$ or $c$ remains a CL for the full model, the converse is not necessarily true. The composed architecture may admit additional conservation laws that cannot be decomposed into CLs of the individual components. In particular, such CLs may involve nontrivial coupling across parameters from multiple layers simultaneously.
>
> Since our goal is to fully characterize all CLs, this distinction is essential. Our current framework operates at the level of individual components and does not yet capture emergent cross-layer conservation structures. To the best of our knowledge, no existing framework provides a complete characterization of all CLs for fully stacked architectures.
>
> In summary, while CLs of individual components remain valid for the full model (a mathematically precise statement), they may not be exhaustive. We hope this clarifies the concern.
>
> ---
>
> **Reference.** Kindly refer to the references in our response to Reviewer r3uS.
>
> ---
>
> We thank the Reviewer for the constructive feedback. If our responses address the concerns, we kindly hope the evaluation can be updated accordingly. We remain open to further discussion.

---

> > ### Author Rebuttal · Reviewer_bMZz · 2026-03-31
> >
> > Thank you for the clarifications regarding the practical significance of the results and the clarification on compositions of CLs. My concerns have been fully addressed and I have increased my rating accordingly.

---

> > > ### Author Response · Authors · 2026-04-01
> > >
> > > We once again thank the Reviewer for their feedback and endorsement. We will ensure that all points raised during the rebuttal phase are incorporated into the revised manuscript.
> > >
> > > Best regards,
> > >
> > > Authors

---

### Official Review · Reviewer_TeQZ · 2026-03-23

**Soundness:** 2
**Presentation:** 2
**Significance:** 2
**Originality:** 2
**Overall Recommendation:** 5
**Confidence:** 3

**Summary:**

This paper studies conservation laws for modern neural architectures under the dataset-independent gradient-flow framework introduced in prior work. The main claimed contributions are complete characterizations for FFNs with GELU/SiLU/SwiGLU, multi-head attention with and without RoPE, and several MoE variants, together with experiments tracking the proposed invariants during optimization. The paper also positions the multi-head attention result as resolving an open problem left by prior work.

**Compliance With Llm Reviewing Policy:**

Affirmed.

**Final Justification:**

My concerns have been adequately addressed.
Overall, I am very happy with the answers provided by the authors in both the rebuttal to my review and to the other reviewers' reviews

**Key Questions For Authors:**

- Theorem 4.6 is stated only for $k>1$, but the appendix proof appears to additionally require $d>n$ to show density of $\Theta_{\mathrm{MoE}}(T)$. Is $d>n$ actually necessary? If yes, the theorem statement should be revised; if not, the proof should explain how to remove this assumption.

- In Appendix B.2, could the authors clarify the step asserting that the RHS of Equation (118) is a rational function of $L$? As written, this is not obvious to me because the expression contains terms involving $R^{L-1}$. A clearer justification would help resolve my main concern about Theorem 4.4.

- Since Section 5 explicitly notes that the framework does not straightforwardly cover layer normalization or deep layer compositions, can the authors add experiments on architectures that more faithfully match the theorem assumptions? This would make the empirical support substantially more convincing. In addition, I would appreciate a somewhat clearer discussion of the paper’s novelty and expected impact relative to prior work. Beyond extending the existing framework to additional modules, what do the authors see as the main conceptual advance, and how should readers think about its significance given that the empirical validation is relatively focused and does not fully match the theoretical setting?

**Limitations:**

yes

**Strengths And Weaknesses:**

Before the following, I would like to clarify to the authors that my own research is somewhat less theoretical than the scope of this paper. I nevertheless tried to read the submission carefully, including the appendix in some detail, but some of the weaknesses I identify may reflect limits in my familiarity with the specific interests of this subcommunity. I have reflected this in my confidence score of 3. That said, I would be genuinely interested in clarifications from the authors during the discussion phase and would be open to revising my score if they address the concerns below.

---

## Strengths
- **S1.** I find the topic relevant and potentially interesting. Extending conservation-law characterizations beyond the linear / ReLU settings studied in prior work to modules such as multi-head attention, RoPE, and MoE is a natural question, and the multi-head attention case in particular is presented as resolving an open problem left by Marcotte et al.

- **S2.** The paper is reasonably easy to follow at a high level. The theorem-by-theorem structure is clear, the paper states the claimed invariants explicitly, and Section 5 gives a useful overview of the intended proof strategy. However I have some questions regarding the formulation of some proofs (see **W3,4,5**), but they might be answered during the discussion phase.

- **S3.** Although I do not find the empirical section fully convincing overall (see **W2.**), it still has some positive aspects: the authors test both full-batch GD and mini-batch SGD, vary the learning rate, etc. This makes the empirical section more than a single anecdotal plot.

---

## Weaknesses
- **W1.** My first concern is that the contribution feels somewhat incremental. The paper explicitly adopts the dataset-independent conservation-law framework of Marcotte et al., builds on the same notion of characterization, and reuses a very similar proof template based on reducing the conservation condition to PDE-like constraints and then exploiting complex-analytic / meromorphic structure. The multi-head case may be nontrivial, but beyond that, much of the paper reads more like a technically involved extension of an existing program than a genuinely new conceptual advance.

- **W2.** The experimental setup is not fully convincing relative to the scope of the claims. The paper explicitly acknowledges that the framework does not straightforwardly extend to additional components such as layer normalization or deep layer compositions, yet the empirical validation is performed on ViT and Qwen-3 style architectures, which are settings that typically include exactly those ingredients. This creates a mismatch between the theorem assumptions and the systems used for validation. More generally, the experiments mainly track block-level conservation-error curves under SGD with linear schedules and zero weight decay, rather than providing sharper quantitative tests on stripped-down architectures that exactly match the theoretical setting.

- **W3.** I am not fully convinced that all the main theorems are rigorously established as written. The clearest issue is *Theorem 4.6*: in the main text it is stated only under the assumption $k>1$, and Remark 5.2 explains $k>1$ as a connectivity condition between active expert sets. However, the appendix proof additionally invokes "By the assumption $d>n$" to show density of $\Theta_{\mathrm{MoE}}(T)$. This looks like an extra assumption used by the proof that is not reflected in the theorem statement.

- **W4.** I also found *Theorem 4.7* insufficiently proved. The theorem states that sigmoid-gated MoE and SMoE have exactly the same conservation laws as the softmax-gated case, but in Appendix C.4 the proof gives one modified identity and then says that the remainder proceeds exactly as in the softmax case, omitting the details. I might be a bit picky here, but I would have liked this part to be more detailed.

- **W5.** I also have a substantive concern with the *RoPE proof (Theorem 4.4)*. In Appendix B.2, the proof states that for fixed $x,y,\theta$, the RHS of Equation (118) is "a rational function in the variable $L$,'' and then concludes that it vanishes identically. However, the displayed expression also contains terms such as $e^{xQ_i R^{L-1} K_i^\top y^\top}$, so the dependence on $L$ is not obviously rational in the ordinary sense. I am therefore not convinced that this continuation step is adequately justified as written. This does not show that the theorem is false, but it does lower my confidence in the rigor of the writing (together with **W6**).

- **W6.** The paper is under-polished and does not look fully submission-ready. The running header still says "Submission and Formatting Instructions for ICML 2026" throughout the main paper and appendix, and there are visible proofreading issues such as "SiwGLU" (caption (b) Figure 2), "spare MoE architectures" (l.409 right column), and minor grammatical mistakes like "Let $n$ denoting the number of expert." (l.298). These issues are not decisive individually, but together they weaken confidence in the care taken with the submission.

---

> ### Author Rebuttal · Authors · 2026-03-31
>
> For clarity, we merge related points and provide unified answers.
>
> ---
> **W1**. While our work shares some similarities in notation and perspective with that of Marcotte et al. (which we cite accordingly), there are significant differences:
> - **Methodological distinction.** As noted at the beginning of Section 3, the framework of Marcotte et al. is based on Lie-theoretic tools, whereas our approach directly analyzes the condition $\frac{d}{dt} h(\theta(t)) = 0$. Although our formulation appears simpler, it does not impose additional assumptions beyond those in Marcotte et al.
> - **Scope of results.** Our work analyzes and fully characterizes CLs for previously unstudied architectures. In particular, the case of multi-head attention remains open, which we address. Moreover, to the best of our knowledge, there is no prior work characterizing CLs for MoE.
>
> We hope this clarifies the distinction and highlights the novelty of our contributions.
>
> **Q1+W3**. Both assumptions $k>1$ and $d>n$ are used in the analysis of MoE; however, the condition $d>n$ was not explicitly stated. This is our mistake, and we will revise the manuscript accordingly.
>
> **On the assumption $d>n$.** Here, $d$ denotes the token dimension, and $n$ the number of experts. In practice, this condition is typically satisfied in MoE-based models:
> - The token dimension $d$ is commonly on the order of $512$–$1024$ for medium models, and significantly larger for large-scale models (e.g., LLaMA 65B has $d=8192$).
> - The number of experts $n$ is typically between $8$-$64$. In some very large-scale models, it can be as high as $256$. However, such increases in $n$ are typically accompanied by a larger $d$.
>
> Therefore, $d>n$ is standard in practical settings and constitutes a mild and reasonable assumption.
>
> **Q2+W5**. We thank the Reviewer for the careful inspection of our proof. We acknowledge an error in Eq. (118): the RHS is not a rational function of $L$. However, the argument can be corrected as follows.
>
> This issue arises in Step 3 of Theorem 4.4 (MHA + RoPE). Instead of taking $\mathbf{x} = (x, 0, \ldots, 0, y)$, we consider $\mathbf{x} = (x,0, \ldots, 0, y, 0, \ldots, 0)$ where there are $L-2$ token $0$ between $x$ and $y$, and an additional $N$ trailing zeros, for arbitrary $N>0$.
>
> The subsequent computation remains unchanged, since zero tokens do not contribute to the MHA output. The attention weights become
>
> $a^{(i)}_{11} = \frac{e^{x Q_i K\_i^\top x^\top}}{ e^{x Q_i K\_i^\top x^\top} + e^{x Q_i R\_{L-1}K\_i^\top y^\top} + (L-2)+N}$
>
> and
>
> $a^{(i)}_{1L} = \frac{e^{x Q_i R\_{L-1}K\_i^\top y^\top}}{ e^{x Q_i K\_i^\top x^\top} + e^{x Q_i R\_{L-1}K\_i^\top y^\top} + (L-2)+N}$
>
> We then substitute these expressions into Eq. (115) to obtain a new equation, replacing Eq. (118). In this new equation, for fixed $x,y,\theta,L$, the RHS is a rational function in $N$. The remainder of the proof proceeds identically, as Lemma B.2 still applies, yielding $\mathfrak{c}_i = \mathfrak{d}_i = 0$.
>
> In summary, the core argument remains valid, but the technical execution in this step requires correction. We will revise the manuscript accordingly. We strive to maintain full rigor in our work, and sincerely thank the Reviewer for the careful and detailed reading of the technical aspects.
>
> **Q3+W2**. We emphasize that our experiments are consistent with theoretical results. This follows from the behavior of CLs under composition: any CL associated with an individual component remains a CL of the full network. For a composition $a \circ b \circ c$, if $h$ is a CL for $a,b$ or $c$, then along gradient flow it satisfies $\frac{d}{dt} h(\theta(t)) = 0$, which follows by a direct algebraic computation.
>
> In summary, **CLs of individual components remain valid for the full model**.
>
> **On theoretical extension to full model.** While our framework applies at the level of individual components, stacking layers may introduce additional, nontrivial CLs that cannot be decomposed into those of the components. These may involve coupling across parameters from multiple layers. Thus, although component-wise CLs remain valid, they need not be exhaustive. To the best of our knowledge, no existing framework fully characterizes all CLs for stacked architectures.
>
> We hope this clarifies the concern.
>
> **W4**. We thank the Reviewer for this suggestion. The proofs for both cases are essentially identical and require no significant modification. However, to ensure completeness, we have updated the manuscript accordingly.
>
> Due to space constraints, we are unable to include the full details here; however, we will ensure they are presented in the camera-ready version if the paper is accepted.
>
> **W6**. We thank the Reviewer for the suggestions regarding typos and writing format, and have revised the manuscript accordingly.
>
> ---
> We thank the Reviewer for the feedback. If our responses address the concerns, we kindly hope the evaluation can be updated accordingly. We remain open to further discussion.

---

> > ### Author Rebuttal · Reviewer_TeQZ · 2026-04-01
> >
> > I would like to thank the authors for the insightful rebuttal they provided.
> >
> > - I agree with their answer to **W1** and I would like to precise that it was not a major weakness in my view.
> > - Answer to **Q1/W3** is convincing, particularly with the examples of standard $d$ and $n$ values.
> > - Answer to **Q2/W5** seems correct. I believe this fixes my concern.
> > - Answer to **Q3/W2** is coherent and credible.
> >
> > Overall, I am very happy with the answers provided by the authors in both the rebuttal to my review and to the other reviewers' reviews. I am therefore moving from 3 to 5 as I believe the authors will make the final version a technically solid, worthy of acceptance paper.

---

> > > ### Author Response · Authors · 2026-04-01
> > >
> > > We once again thank the Reviewer for their feedback and endorsement. We will ensure that all points raised during the rebuttal phase are incorporated into the revised manuscript.
> > >
> > > Best regards,
> > >
> > > Authors

---

### Decision · Program_Chairs · 2026-04-30

**Decision:**

Accept (spotlight)

**Comment:**

The paper studies conservation laws in gradient flow for modern architectures. All reviewers participated in the rebuttal and recommended acceptance. The reviewers agree that the problem is interesting, the results are significant, and the paper is well-written.
I recommend acceptance, and encourage the authors to incorporate in the final version the issues discussed in the rebuttal.